# Modelling polar marine ecosystem functions guided by bacterial physiological and taxonomic traits

Hyewon Heather Kim[1, 2, *], Jeff S. Bowman[3], Ya-Wei Luo[4], Hugh W. Ducklow[5], Oscar M. Schofield[6], Deborah K. Steinberg[7], Scott C. Doney[2]

[1]Woods Hole Oceanographic Institution, Woods Hole, MA 02543, USA
[2]University of Virginia, Charlottesville 22904, USA
[3]Scripps Institution of Oceanography, UC San Diego, La Jolla, CA, USA
[4]Xiamen University, Xiamen, China
[5]Lamont-Doherty Earth Observatory of Columbia University, Palisades, NY 10964, USA
[6]Rutgers University, New Brunswick, NJ 80901, USA
[7]Virginia Institute of Marine Science, William & Mary, Gloucester Point, VA 23062, USA

*Correspondence to*: Hyewon Heather Kim (hkim@whoi.edu)

## Abstract

Heterotrophic marine bacteria utilize organic carbon for growth and biomass synthesis. Thus, their physiological variability is key to the balance between the production and consumption of organic matter and ultimately particle export in the ocean. Here we investigated a potential link between bacterial traits and ecosystem functions in a rapidly changing polar marine ecosystem based on a bacteria-oriented ecosystem model. Using a data-assimilation scheme we utilized the observations of bacterial groups with different physiological states to constrain the group-specific bacterial ecosystem functions. We then examined the association of the modelled bacterial and other key ecosystem functions with eight recurrent modes representative of different bacterial taxonomic traits. High nucleic acid (HNA) bacteria showed relatively high cell-specific productivity, respiration, and utilization of the semi-labile dissolved organic carbon pool compared to their low nucleic acid (LNA) bacteria counterparts. Both taxonomy and physiology traits reflected the variability of bacterial carbon demand, net primary production, and particle sinking flux. Numerical experiments under perturbed climate conditions showed a potential shift from LNA- to HNA-dominated bacterial communities in the warming WAP. Our study suggests that bacterial diversity via different taxonomic and physiological traits can guide the modelling of the WAP ecosystem, providing insights into key bacterial and ecosystem functions under climate change.

## 1 Introduction

Microbes regulate many key ecosystem functions in the marine food web. Unicellular primary producers fix organic carbon (i.e., an ecosystem function termed primary production), while heterotrophic marine bacteria and archaea (hereafter bacteria) utilize the fixed organic carbon for growth and biomass synthesis (i.e., an ecosystem function termed bacterial production) (Azam et al. 1983). Thus, the variability in the abundance and activity of bacteria is central to understanding the balance between production and consumption of organic matter and ultimately particle export. In flow cytometric analyses, bacteria cluster into two groups of cells with different nucleic acid content, including high nucleic acid (HNA) and low nucleic acid (LNA) cells (Bouvier et al. 2007; Gasol et al. 1999). These two groups are suggested to represent lineages (Schattenhofer et al. 2011; Vila-Costa et al. 2012) or physiological states (Bowman et al. 2017), and HNA cells are generally larger in both

cell and genome size compared to LNA cells (Bouvier et al. 2007; Calvo-Díaz and Morán 2006). The significance of HNA versus LNA cells in determining distinct ecosystem states and functions has been investigated, but much is still unknown. In a recent study along the West Antarctic Peninsula (WAP), the high dimensionality of the bacterial community structure data was reduced via emergent self-organizing maps and subdivided into a small number of bacterial modes associated with specific taxonomic and functional traits (Bowman et al. 2017). Bowman et al. (2017) demonstrated that a combination of taxonomy, physiological structure (i.e., HNA and LNA cells), and abundance of bacterial communities explained up to 73% of the variance in bulk bacterial production. Their findings imply that bacterial physiological and taxonomic traits could inform a predictive ecosystem model to further explore ecologically important questions such as: *Can bacterial traits reflect key ecosystem functions like primary production and particle sinking flux? If so, what are the underlying mechanisms driving such bacterial trait-ecosystem function relationships? And how will these relationships be impacted by climate change?*

The WAP is a rapidly warming marine ecosystem, with resulting changes in physical, ecological, and biogeochemical processes (Clarke et al. 2009; Cook et al. 2005; Ducklow et al. 2007; King 1994; Meredith and King 2005; Stammerjohn et al. 2008; Vaughan et al. 2003; Vaughan 2006; Whitehouse et al. 2008). Routine monitoring through the Palmer Long-Term Ecological Research project (Palmer LTER; since 1991) has revealed climate-driven variations in seasonal phytoplankton accumulation (Saba et al. 2014; Schofield et al. 2017), bacterial dynamics (Bowman and Ducklow 2015; Ducklow et al. 2012a; Kim and Ducklow 2016; Luria et al. 2017; Luria et al. 2014), nutrient drawdown (Kim et al. 2016), and micro- and macrozooplankton dynamics (Garzio and Steinberg 2013; Steinberg et al. 2015; Thibodeau et al. 2019). The wealth of Palmer LTER observations enabled the construction of a numerical marine ecosystem model for the coastal WAP region (i.e., the WAP-1D-VAR model; Kim et al. 2021), based on existing regional test-bed models of other ocean basins (Friedrichs 2001; Friedrichs et al. 2006, 2007; Luo et al. 2010, 2012). The WAP-1D-VAR model was compared against roughly bi-weekly time-series data over the growth season (October - March) near Palmer Station (64.77°S, 64.05°W; the mean depth of ~65 m) that recorded seasonal variations in ecological processes modulated by variations in surface light, mixed layer depth, and surface sea-ice cover. The WAP-1D-VAR model utilized a data assimilation scheme to minimize the misfits between model results and observational data via a variational adjoint method (Lawson et al. 1995), by assimilating the available Palme LTER observations. Serving as a mechanistic model, assimilation of the Palmer LTER observations constrained poorly measured bacterial processes (e.g., respiration, viral and grazing mortality, growth efficiency, carbon demand, and utilization of dissolved organic matter with varying lability) and enabled model predictions of the microbial system state in changing environments. Yet, incorporating molecular observations into an ecosystem model is still a challenge due to differences in how levels of biological organization are treated in observations and models (Hellweger 2020) and the high dimensionality of microbial molecular observations. One argument is that molecular-level changes may not directly translate into a clear picture of changes in community structure or resulting changes to bulk ecosystem functions.

In this study, we explored a potential link between bacterial traits and ecosystem functions in the warming coastal WAP, using a bacteria-oriented ecosystem model originally derived and modified from the WAP-1D-VAR model (Kim et al. 2021). The bacterial traits examined in this study included both physiological and taxonomic traits. For physiological traits, the model explicitly simulated the dynamics of the two ubiquitous bacterial groups of differing nucleic acid content, a HNA group and a LNA group, by directly assimilating the group-specific biomass observations. For taxonomic traits, taxonomic modes derived from bacterial 16S rRNA gene sequence data (calculated by Bowman et al. 2017) were compared to model output values at the corresponding time points, with the assumption that microbial taxonomy would provide information about microbial ecosystem process and structure. In contrast to genome-scale, metabolic flux, or gene-centric models (Coles et al. 2017; Feist et al. 2009; Reed et al. 2014), this study combines statistical products from genomic analyses with numerical ecosystem modelling to incorporate molecular information into ecosystem-level dynamics.

## 2 Material and Methods

### 2.1 Bacteria-oriented ecosystem model

The bacteria-oriented ecosystem model was originally derived and modified from the 1-D variational data assimilation model for the coastal WAP region (WAP-1D-VAR v1.0; Kim et al. 2021) simulating 12 state variables, including diatoms, cryptophytes, bacteria, microzooplankton, krill, labile dissolved organic matter (LDOM), semi-labile DOM (SDOM), ammonium ($NH_4$), nitrate ($NO_3$), phosphate ($PO_4$), and particulate detritus (Figure 1; model equations in Appendix A; more details about model setup in Text S1-4). Refractory DOM (RDOM) and higher trophic levels were implicitly represented as

model closure terms. Distinct to this study's model compared to the WAP-1D-VAR model was the inclusion of HNA and LNA bacterial compartments (as biomass) and the partitioning of the bulk bacterial productivity by each bacterial compartment. The model was forced by mixed layer depth (MLD), photosynthetically active radiation (PAR) at the ocean surface, surface sea-ice concentration, water-column temperature, and eddy diffusivity (Figure S1) to simulate the stocks and flows of C, N, and P through the model state variables using a constant time step of 1 hour and a second-order Runge-Kutta

scheme. In essence, the time rate of change of the group-specific biomass (constrained by observations; section 2.4) was calculated as follows:

$$\frac{dC_{HNA}}{dt} = G^C_{HNA,LDOC} + G^C_{HNA,SDOC} - R^C_{HNA} - E^C_{HNA,RDOC} - E^C_{HNA,SDOC} - GZ^C_{HNA} - M^C_{HNA} \qquad (1)$$

where $C_{HNA}$ is biomass (mmol C m$^{-3}$), $G^C_{HNA,LDOC}$ is LDOC consumption (mmol C m$^{-3}$ d$^{-1}$; Eq. A.4.12), $G^C_{HNA,SDOC}$ is SDOC consumption (mmol C m$^{-3}$ d$^{-1}$; Eq. A.4.13), $R^C_{HNA}$ is respiration (mmol C m$^{-3}$ d$^{-1}$; Eq. A.4.25), $E^C_{HNA,RDOC}$ is RDOC excretion

(mmol C m$^{-3}$ d$^{-1}$; Eq. A.4.26), $E^C_{HNA,SDOC}$ is SDOC excretion (mmol C m$^{-3}$ d$^{-1}$; Eq. A.4.38, A.4.41), $GZ^C_{HNA}$ is C-specific grazed amount of cells by microzooplankton (mmol C m$^{-3}$ d$^{-1}$; Eq. A.4.44), and $M^C_{HNA}$ is viral mortality (mmol C m$^{-3}$ d$^{-1}$; Eq. A.4.47) of the HNA group (the same form applies to LNA group below).

$$\frac{dC_{LNA}}{dt} = G^C_{LNA,LDOC} + G^C_{LNA,SDOC} - R^C_{LNA} - E^C_{LNA,RDOC} - E^C_{LNA,SDOC} - GZ^C_{LNA} - M^C_{LNA} \qquad (2)$$

Total (bulk) bacterial production (BP; $BP = BP_{HNA} + BP_{LNA}$) was constrained by observations, and therefore, the group-

specific production ($BP_{HNA}$ and $BP_{LNA}$, mmol C m$^{-3}$ d$^{-1}$) was determined during optimization:

$$BP_{HNA} = G^C_{HNA,LDOC} + G^C_{HNA,SDOC} - R^C_{HNA} \qquad (3)$$
$$BP_{LNA} = G^C_{LNA,LDOC} + G^C_{LNA,SDOC} - R^C_{LNA} \qquad (4)$$

### 2.2 Modelling framework

The modelling framework consisted of a dynamic (mechanistic) part and a data-driven part (Figure 2): 1) the dynamic

part as the processes associated with the data-assimilative model (Figure 1) that made predictions of the microbial ecosystem processes based on prognostic, time-evolving coupled ordinary differential equations (Appendix A); and 2) the data-driven part representing how bacterial modes (Bowman et al. 2017) were compared to optimized model outputs. The analysis in the present study relied on the two types of modes, including taxonomic modes and functional modes, described in Bowman et al. (2017): taxonomic modes (modes hereafter) were determined from 16S rRNA gene sequence abundance, while functional

modes (*f*modes hereafter) were derived from predicted community metabolic structure. Briefly, sequence reads were categorized into closest estimated genomes and closest completed genomes via the paprica pipeline (Bowman and Ducklow, 2015) and the high dimensional community and metabolic structure data were reduced to 2-D space via a self-organizing map and K-means clustering of map units (Bowman et al. 2017). The final clustering of map units constitutes the modes, and each sample was assigned the mode of its closest map unit. In this approach the mode is a single categorical variable that succinctly

describes key structural attributes of the sample. It is important to recognize the categorical nature of these modes, and to

understand that – because of the 2-D nature of the map – there is no linear progression among modes. Thus mode 1, for example, is not necessarily more similar to mode 3 than it is to mode 7. Neither mode nor *f*modes is necessarily correlated to physiological traits of the bacteria (i.e., HNA- and LNA compartments). In other words, derived from separate observations of different parameters in the same bacterial samples, the relative abundance of HNA or LNA, mode, and *f*modes are independent with each other by design.

We selected a nearshore Palmer LTER water-column time-series station, Station B (64.77°S, 64.05°W), in the coastal WAP as the modelling site for this study (the mean depth of ~65 m). The Palmer LTER Station B datasets consisted of roughly bi-weekly physical, chemical, and biological profiles collected via a profiling Conductivity-Temperature-Density (CTD) and rosette. Additional observational data were utilized for bacterial flow cytometric (HNA and LNA) and 16S rRNA gene amplicon data collected from Arthur Harbour Station B at 10 m depth (situated 1 km from the Palmer Station B) or Palmer Station seawater intake at 6 m depth (Bowman et al. 2017). Three upper-ocean depth levels (0, 10, and 20 m) were modelled for 4 consecutive Palmer LTER growth seasons, including November 2010 - March 2011 (year 2010-11 hereafter), 2011-12, 2012-13, and 2013-14, but the only results from 10 m were analysed in detail due to the availability of bacterial traits data there. The 1-D modelling of the coastal WAP region could be justified given that the WAP is a region of relatively weak net advection compared with the Antarctic Circumpolar Current (ACC) or the subpolar gyres (Meredith et al., 2008, 2013), and the CTD observations at Palmer Station did not show the evidence of abrupt changes in physical and biogeochemical tracers due to lateral advection, but showed rather laterally homogeneous temperature and salinity distributions during the Antarctic growing season for the modelled depth and years in this study (Kim and Ducklow 2016).

Given the availability of the Palmer LTER observations over the Austral spring-summer season, we optimized the model each year separately over the timeframe of available observations. This way, each year possessed its own unique optimized model parameter set (model) equivalent to a model solution for the minimized model-observation misfit for that year. In addition to these four years (2010-11 to 2013-14), we optimized the model for the climatological year, referred as the climatological model. The climatological year was constructed by averaging observations in the four years (2010-11 to 2013-14), rather than the whole Palmer LTER multi-decadal period (since 1991), due to the limited availability of HNA and LNA biomass data only in those four years. Details on constructing the climatological year and model initialization, spin-up, and bottom boundary conditions are found in the Supplementary Material (Text S3).

**2.3 Data assimilation and parameter optimization**

The model utilized a variational adjoint data assimilation scheme (Lawson et al. 1995) to minimize the misfit between observations (i.e., assimilated data, section 2.4) and model output by optimizing a subset of model parameters (Friedrichs 2001; Spitz et al. 2001; Ward et al. 2010). The data-assimilation scheme (Figure 2) consisted of four main steps (Glover et al. 2011). First, the model was integrated forward in time from prescribed initial conditions and initial model parameter guess values (Table 1) to calculate the model-observation misfits referred as total cost function or total cost (section 2.5). Second, an adjoint model constructed using the Tangent linear and Adjoint Model Compiler (TAPENADE) was integrated backward in time and compute the gradients of the total cost with respect to the model parameters. Third, the computed gradients were passed to a limited-memory quasi-Newton optimization software M1QN3 3.1 (Gilbert and Lemaréchal 1989) to determine the direction and optimal step size by which the model parameters needed to be modified to reduce the total cost. Finally, a new forward mode simulation was performed using the new set of modified parameters from the third step. These four steps were conducted in an iterative manner until the pre-set convergence criteria were satisfied ensuring the convergence of the optimized parameters and a local minimum achieved by the total cost, via low gradients (sensitivity) of the total cost with respect to each optimized parameter and positive eigenvalues of the Hessian matrix (section 2.6).

The initial parameter subset submitted to optimization consisted of 10 different model parameters, with one parameter per each state variable, the change of which yielded the largest decrease in the total cost function during preliminary sensitivity

tests, including $\alpha_{DA}$ (initial slope of photosynthesis vs. irradiance curve of diatoms, mol C (g Chl $a$)$^{-1}$ d$^{-1}$ (W m$^{-2}$)$^{-1}$), $\alpha_{CR}$ (initial slope of photosynthesis vs. irradiance curve of cryptophytes, mol C (g Chl $a$)$^{-1}$ d$^{-1}$ (W m$^{-2}$)$^{-1}$), $\Theta$ (maximum Chl:N ratio, g Chl $a$ (mol N)$^{-1}$), $\mu_{HNA}$ (maximum HNA growth rate, d$^{-1}$), $r^A_{max,HNA}$ (maximum HNA active respiration rate, d$^{-1}$), $g_{HNA}$ (half-saturation density of HNA bacteria in microzooplankton grazing, mmol C m$^{-3}$), $\mu_{MZ}$ (maximum microzooplankton growth rate, d$^{-1}$), $\mu_{KR}$ (maximum krill growth rate, d$^{-1}$), and $rem_{V_{KR}}$ (krill removal rate by higher-trophic levels, (mmol C m$^{-3}$)$^{-1}$ d$^{-1}$; Tables S2-6). In most cases this initial subset led to positive eigenvalues of the inverse Hessian matrix, but in case of negative eigenvalues the corresponding parameters were eliminated from optimization (section 2.6) and a few other sensitive parameters were added and kept, one at each optimization cycle, if they further reduced the total cost function. Every assimilation cycle, we ensured that group-specific bacterial model parameters were optimized in the direction to properly represent the dynamics associated with each group (Table 1), in which we assigned different magnitudes of each parameter based on the best guesses and literatures (del Giorgio and Cole 1998; Jiao et al. 2010). For instance, maximum bacterial growth rate of the HNA group ($\mu_{HNA}$, d$^{-1}$) was ensured to be optimized to be higher than that of the LNA group ($\mu_{LNA}$, d$^{-1}$), so if $\mu_{HNA}$ was optimized smaller than $\mu_{LNA}$, $\mu_{HNA}$ was reset back to the original value instead of being updated.

## 2.4 Assimilated data

We assimilated Palmer LTER observational data from 0, 10, and 20 m corresponding to compartments and flows in the model, including NO$_3$, PO$_4$, phytoplankton taxonomic specific chlorophyll (Chl) for diatoms and cryptophytes (Schofield et al. 2017), microzooplankton biomass (Garzio et al. 2013), primary production (PP), bulk BP, HNA bacterial biomass, LNA bacterial biomass, semi-labile dissolved organic carbon (SDOC), particulate organic carbon (POC), and particulate organic nitrogen (PON). The group-specific Chl was not measured in 2011-12, but due to its importance in constraining the group-specific phytoplankton dynamics, the 4-year climatological value was assimilated for 2011-12. NO$_3$ was not assimilated in 2010-11, while POC, PON, and SDOC were not assimilated in 2012-13 and 2013-14 due to the lack of observations in those years. Krill biomass data were not assimilated due to the strong patchiness of the distribution (many zero values) that would hinder proper model optimization, while a single year measurement data of microzooplankton biomass (2010-11) was assimilated for all years to at least provide constraints on phytoplankton grazing parameters. Microzooplankton model-observation misfits were not examined due to the discrepancy in the timing and location of the data compared to this study.

SDOC was calculated by subtracting the background (RDOC) concentration (40.0 mmol m$^{-3}$) from climatological total DOC concentration. POC (PON) were assimilated to represent the model detrital pool, but its measurements contained living biomass from bottle filter experiments. Climatological observations showed that living phytoplankton and bacterial biomass accounted for 74% of total POC and 71% of total PON, so these fractions were used to exclude living biomass from the bulk particulate material pool. When converting Chl to phytoplankton C (N) biomass, the maximum Chl to N ratio was used along with other reference ratios (Tables S2-6). BP (mmol C m$^{-3}$ d$^{-1}$) was derived from $^3$H-leucine incorporation rate (pmol l$^{-1}$ h$^{-1}$) data using the conversion factor of 1.5 kgC mol$^{-1}$ leucine incorporated (Ducklow 2000). Group-specific bacterial biomass (mmol C m$^{-3}$) was estimated from bacterial abundance measured by flow cytometry (i.e., bulk bacterial biomass multiplied by the fraction of each group, $f_{HNA}$ or $f_{LNA}$, with the conversion factor of 10 fgC cell$^{-1}$) (Fukuda et al. 1998).

## 2.5 Cost function and portability index

The total cost function or cost ($J$) was defined as follows to represent the misfit between observations ($\hat{a}_{m,n}$) and model output ($a_{m,n}$) (Luo et al. 2010):

$$J = \sum_{m=1}^{M} \frac{1}{N_m} \sum_{n=1}^{N_m} \left( \frac{a_{m,n} - \hat{a}_{m,n}}{\sigma_m} \right)^2 \tag{5}$$

where $m$ and $n$ represent assimilated data types and data points, respectively, $M$ and $N_m$ are the total number of assimilated data types and data points for data type $m$, respectively, and $\sigma_m$ is the target error for data type $m$. Hereafter, we referred the

total cost as the total cost normalized by $M$ ($J' = J/M$) and normalized costs of individual data types ($J'_m$) as the model-observation misfit equivalent to a reduced Chi-square estimate of model goodness of fit (i.e., $J' = 1$ as a good fit from optimization, $J' \gg 1$ as a poor fit due to underestimation of the error variance or the fit not fully capturing the data, and $J' \ll 1$ as an overfitting of the data, fitting the noise, or overestimation of the error variance). The base-10 logarithm of Chl and PP was used in Eq. 5 to account for high productivity of the WAP waters and the approximate log-normal distribution of those data types (Campbell 1995; Glover et al. 2018). The target error $\sigma_m$ was calculated for each data type $m$ as:

$$\sigma_m = \overline{\hat{a}_{m,n}} \cdot CV_m \tag{6}$$

where $\overline{\hat{a}_{m,n}}$ is the climatological mean of the observations and $CV_m$ is the adjusted coefficient of variation (CV) of the observations of each data type over 0, 10, and 20 m (due to observational error and seasonal and interannual variations). The average CV of each data type at a single depth across the modelled years was higher compared to those across every measured depth within the mixed layer over an extended year period modelled in the WAP-1D-VAR model (2002-03 to 2011-12; Kim et al. 2021) and was therefore reduced to the level in the mixed layer to avoid an overestimated target error of each data type. The rationale behind using the adjusted CV in the target error calculation was based on Luo et al (2010), in which all properties in the mixed layer should be completely mixed, a perfect measurement without significant errors should generate similar data values at every measured depth within the mixed layer, and the average CV of all the depth profiles can be used as CV in the target error calculation. The standard deviation was used as target errors of the log-converted data types. The CV of the log-converted data type was estimated as the average of ± 1 standard deviation in log space converted back into normal space (Doney et al. 2003; Glover et al. 2018).

We computed the portability index to evaluate the broader applicability of the optimized model parameter set of each year in predicting dynamics of the other year (Friedrichs et al. 2007):

$$\text{Portability index} = J'_c/J'_x \tag{7}$$

where $J'_x$ is the normalized cross-validation cost when a model parameter set optimized for a given year is used to simulate another year, and $J'_c$ is the normalized total cost of the climatological model. A portability index close to 1 indicates a more portable model, or a system that is not particularly sensitive to year-to-year variations in optimized model parameters, while an index $\ll 1$ indicates a less portable model, or a system sensitive to year-to-year variations in optimized model parameters.

**2.6 Uncertainty analysis**

The uncertainties of the optimized parameters were estimated from a finite difference approximation of the complete Hessian matrix during iterative data assimilation processes (i.e., second derivatives of the cost function with respect to the model parameters; Text S4). When computed at the minimum of the cost function value, the square root of a diagonal element in the inversed Hessian matrix is the logarithm of the relative uncertainty of the corresponding optimized parameter. The absolute uncertainty of the constrained parameter was calculated as $p_f \times e^{\pm \sigma_i}$ where $p_f$ is the value of optimized parameter and $\sigma_f$ is the relative uncertainty of the corresponding optimized parameter. We then conducted Monte Carlo experiments to calculate the impact of the optimized parameter uncertainties on the model results. The Monte Carlo experiments consisted of 1) creating an ensemble of parameter sets ($N = 1,000$) by randomly sampling values within the uncertainty ranges of the constrained parameters and 2) then performing a model simulation using each parameter set. All uncertainty estimates were calculated following standard error propagation rules and presented herein as ± 1 standard deviation.

## 3 Results

### 3.1 Model skill assessment

The iterative optimization procedure (Figure 2) reduced by 24-93% the misfits between observations and model output for each year and the climatological year, compared to the misfits obtained using the initial guess parameters (Table 2). The optimized parameter sets satisfied the pre-set convergence criteria, including local minima achieved by the total costs, low gradients of the total costs with respect to each optimized parameter, and positive eigenvalues of the Hessian matrix. The total costs were reduced by optimizing only a subset of the model parameters (5-7 constrained parameters and 3-6 optimized parameters; Tables S2-6). The optimized parameters in common across all years were $\alpha_{DA}$ (initial slope of photosynthesis vs. irradiance curve of diatoms, mol C (g Chl $a$)$^{-1}$ d$^{-1}$ (W m$^{-2}$)$^{-1}$), $\mu_{HNA}$ (maximum HNA bacterial growth rate, d$^{-1}$), $\mu_{LNA}$ (maximum LNA bacterial growth rate, d$^{-1}$), and $g_{CR}$ (half-saturation density of cryptophytes in microzooplankton grazing, mmol C m$^{-3}$). $g_{HNA}$ (half-saturation density of HNA bacteria in microzooplankton grazing, mmol C m$^{-3}$), $g_{MZ}$ (half-saturation density of microzooplankton in krill grazing, mmol C m$^{-3}$), and $\mu_{KR}$ (maximum krill growth rate, d$^{-1}$) were next frequently optimized, at least for 4 years out of a total of 5 modelled years.

Because this study focused on composites of the modelled ecosystem functions as a function of bacterial mode and of the fraction of different physiological groups (section 3.2), rather than of year (Figures S2-5), we combined observations and model results from all four years together for further model skill assessment. The Taylor diagrams indicated similar model skills between the four years (Figure 3a) and the climatological year (Figure 3b). Three core bacterial variables in this study, including HNA biomass, LNA biomass, and BP, showed overall better model-observation agreements than other data types, with relatively high correlations, low centred (bias removed) root-mean-square difference (RMSD), and normalized standard deviation closer to 1. These three variables also had better fits to the four-year seasonal cycles of the observations than other data types (Figure S7). However, the model skill for HNA biomass slightly degraded in the climatological model (Figure 3b), with lower correlations and normalized standard deviation and higher RMSD than the four years together (Figure 3a). The optimized models captured best the temporal and spatial (depth) variability of PP, as shown by its high correlations (Figure 3), but the models tended to underestimate PP with relatively larger errors than for other data types (Figure S7). By contrast, there were slight positive model biases for POC and PON (Figure S7) and their variability was not well captured as shown by the negative correlations (Figure 3).

Cross-validation cost analyses showed increased model-observation misfits when a set of parameters optimized for one year was used to simulate another year's dynamics (Tables 1-2), suggesting that each year was best modelled using its own unique set of optimized parameters. The magnitude of the cost function increase varied by year pair, with the portability index values indicating that the optimized model parameters for 2012-13 was most portable (0.76 ± 0.11), followed by those for 2013-14 (0.73 ± 0.17), 2010-11 (0.67 ± 0.08), and 2011-12 (0.61 ± 0.12; Table 2).

### 3.2 Bacterial carbon stocks and flows

C stocks and flows for each bacterial group showed significant seasonal and interannual variability (Figure 4A, Figure S6). Across years HNA cells had significantly higher seasonal maximum values than their LNA counterparts, when normalized by the group-specific biomass. These so-called cell-specific, seasonal maximum rates of the HNA group ranged from 0.10 ± 0.00 to 0.59 ± 0.24 d$^{-1}$, 0.03 ± 0.00 to 0.18 ± 0.12 d$^{-1}$, 0.02 ± 0.01 to 0.18 ± 0.08 d$^{-1}$, 0.05 ± 0.00 to 0.57 ± 0.26 d$^{-1}$, and 0.07 ± 0.03 to 0.36 ± 0.17 d$^{-1}$ for LDOC uptake, SDOC uptake, respiration, BP, and grazing rates, respectively (Figure 4). For the LNA group, maximum cell-specific rates ranged from 0.01 ± 0.00 to 0.12 ± 0.02 d$^{-1}$, 0.00 ± 0.00 to 0.03 ± 0.01 d$^{-1}$, 0.01 ± 0.00 to 0.02 ± 0.01 d$^{-1}$, 0.00 to 0.13 ± 0.02 d$^{-1}$, and 0.02 ± 0.00 to 0.17 ± 0.03 d$^{-1}$ for LDOC uptake, SDOC uptake, respiration, BP, and grazing rates, respectively (Figure 4). C stocks and flows averaged over the growth (Figure 5) and normalized by NPP

(normalized by NPP in 1-day for C stocks; Figure S9) season for each year summarized an annual snapshot of the group-specific bacterial dynamics. The annual mean LNA biomass was ~17 times larger than that of HNA biomass in 2011-12 (Figure 5b), in contrast to relatively similar mean biomass of both groups in other years (Figures 5a, c, d). Bacterial carbon demand (BCD; i.e., BCD = BP + bacterial respiration; blue arrows for respiration in Figure 5) was mostly supported by the LDOC pool (67-81%) for both bacterial groups.

The rest of the model stocks and flows fell into one of three categories: 1) the variable for a single year values were assimilated (i.e., microzooplankton biomass); 2) the variables for which observational values for the given year were assimilated (i.e., nutrients, POC or detritus, and SDOC), and 3) the variables that were not assimilated (i.e., krill biomass, LDOC, $NH_4$, and particle sinking flux). There was a little interannual variability in the average microzooplankton carbon biomass (Figure 5). $NO_3$, POC, and SDOC in unassimilated years were modelled to values comparable to those in other assimilated years (Figure 5). Modelled LDOC and $NH_4$ were also within the reasonable range of their typically small value (< 1 μM).

## 3.3 Bacterial physiological and taxonomic association with ecosystem functions

A property map of the emergent self-organizing map nodes (as generated by Bowman et al. 2017) showed the mode association with community structure (Figure 6). The coloured map units (the circles in the background) were clustered into taxonomic mode membership or modes (Figure 6A), showing a different frequency of appearance year-to-year (Figure 6B). Each mode was dominated by unique bacterial taxa. For example, *Candidatus* Pelagibacter was most abundant in mode 6 (Figure 6C), *Dokdonia* sp. MED134 in mode 7 (Figure 6D), *Candidatus* Thioglobus singularis PS1 in mode 1 (Figure 6E), *Owenweeksia hongkongensis* DSM 17368 in mode 2 (Figure 6F), Rhodobacteraceae in mode 5 (Figure 6G), and *Planktomarina temperata* RCA23 in mode 4 (Figure 6H).

To explore a potential link between mode and the key ecosystem functions, we first extracted the modelled net primary production (NPP), POC sinking flux, and BCD from the ecosystem model at the time of bacterial samples and depth (10 m) that were placed into a single, observed mode. We then performed a linear regression with mode as a factor (i.e., mode as a categorical predictor with 8 modes rather than an ordinal or continuous variable; equivalent to a one-way ANOVA with 8 different categories). *f*mode did not have a significant relationship with any of the modelled ecosystem functions examined (all $p > 0.05$; not shown). By contrast, 27%, 36%, and 77% of the total variance in the modelled NPP, POC sinking flux, and BCD was explained by mode (Figures 7A-C). In particular, modes 3, 5, and 7 were associated with 2-3 times higher NPP, POC sinking flux, and BCD, compared to when mode 4 dominated (two-sample $t$-test with unequal sample size, $p = 0.02$ for NPP and $p < 0.001$ for POC sinking flux and BCD), or to when mode 6 dominated ($p = 0.03$ for NPP, $p = 0.003$ for POC sinking flux, and $p < 0.001$ for BCD).

The observed mode was positively correlated to the observed *f*HNA ($r^2 = 0.52$, $p < 0.001$; not shown). Thus, we also examined a potential link between the observed *f*HNA and the key model ecosystem functions as described above (i.e., linear regression with an observed *f*HNA as a predictor and the modelled ecosystem functions as dependent variables). The observed *f*HNA was positively correlated to the modelled NPP ($r^2 = 0.33$, $p < 0.001$; Figure 7d), and to a stronger extent, to the modelled POC sinking flux ($r^2 = 0.51$, $p < 0.001$; Figure 7E) and to the modelled BCD ($r^2 = 0.54$, $p < 0.001$; Figure 7F). The stepwise addition of one predictor variable to the other predictor variable (i.e., *f*HNA adding to mode or vice versa) did not improve the model performance (not shown). These results suggest a clear link between the modelled ecosystem functions and observed bacterial taxonomic (modes) and physiological (*f*HNA) traits observations.

**3.4 Climate change experiments**

We explored the response of the modelled bacterial dynamics and ecosystem functions (sections 3.2-3.3) to changing climate along the WAP (Figure 8). Due to a varying range of portability of the optimized model solution for each year, we used the climatological model parameter set (Table S6) to simulate an overall system response under perturbed ocean temperature (+0.5°C and +1.0°C relative to observed temperatures) and sea-ice forcing fields (5% and 10% loss of sea-ice relative to observed sea-ice concentrations). These experiments were conducted under each perturbed environmental condition separately (i.e., warming alone in Figure S10 versus melting alone in Figure S11) and simultaneously (i.e., climate change experiments; Figure 8). We only focused on the results from the climate change experiments in this section, given that despite different impacts of each physical forcing changes (i.e., temperature impacts on rate processes versus sea ice impacts on light and photosynthesis but not MLD) climate change would cause simultaneous changes in sea ice and water temperature along the WAP.

The climate change experiments resulted in a combination of changes in overall bacterial stocks and rates, as well as the key ecosystem functions, and shifts in the seasonal timing or phenology (Figure 8A), compared to the base state (first row as the base state and second and third rows as anomalies under perturbed conditions in Figure 8B). HNA bacterial stock and rates responded more strongly to the perturbed climate conditions compared to LNA bacterial stock and rates. Under combined warming/melting (+1.0°C/-10%) conditions, there were maximum increases of the HNA variables by 19-35% ($29 \pm 89\%$ for biomass, $22 \pm 67\%$ for LDOC uptake, $35 \pm 111\%$ for SDOC uptake, $26 \pm 79\%$ for respiration, $25 \pm 78\%$ for BP, $29 \pm 89\%$ for viral mortality, $19 \pm 26\%$ for grazing, and $29 \pm 89\%$ for RDOC excretion), compared to the maximum increases of the LNA variables by 3-15% ($3 \pm 2\%$ for biomass, $6 \pm 11\%$ for LDOC uptake, $15 \pm 27\%$ for SDOC uptake, $8 \pm 3\%$ for respiration, $7 \pm 6\%$ for BP, $3 \pm 2\%$ for viral mortality, $7 \pm 18\%$ for grazing, and $3 \pm 2\%$ for RDOC excretion). In contrast to most bacterial variables that increased consistently throughout the growth season, microzooplankton grazing rates showed seasonally mixed responses for both HHA and LNA groups (i.e., the maximum decreases of $8 \pm 32\%$ for HNA and of $4 \pm 32\%$ for LNA). Similarly, there were maximum increases of NPP and POC sinking flux by $14 \pm 15\%$ and $3 \pm 22\%$, and maximum decreases by $4 \pm 11\%$ and $3 \pm 13\%$, respectively. SDOC exhibited the maximum increase by $2 \pm 1\%$ early in the season but shortly became depleted as the season progressed. LDOC decreased always in response to the perturbed conditions, with the maximum decrease by $10 \pm 43\%$.

**4 Discussion**

**4.1 Model skill assessment**

Despite the important biogeochemical role that bacteria play in the ocean, the vast majority of marine ecosystem models neither include bacteria as a model compartment nor explicitly simulate bacterial processes. Most existing models parameterize the complex bacterial remineralization processes of the sinking organic matter with depth by using empirical relationships, such as, by fitting the power law functions, or other similarly-derived approaches and parameterizations (Buesseler et al. 2020; Cael and Bisson, 2018). Cellular functions, taxa, and functional gene expression of other prokaryotes, such as cyanobacteria (Hellweger 2010; Martín-Figueroa et al. 2000; Miller et al. 2013), or a diverse suite of microbial functional groups (Coles et al. 2017; Dutkiewicz et al. 2020) have been modelled so far; however, our study serve as the first to explicitly model bacterial groups of different physiological traits.

In this study, only a subset of the model parameters was optimized to best simulate bacterial and other ecological patterns for each year, consistent with other data assimilation modelling studies (Friedrichs 2001; Friedrichs et al. 2006, 2007; Luo et al. 2010, 2012). In general, optimization of this class of marine ecosystem models requires adjustment of a small number

of independent model parameters to achieve well-posed model solutions, due to the highly cross-correlated nature of parameters in the inherently nonlinear model equations (Fennel et al. 2001; Harmon and Challenor 1997; Matear 1996; Prunet et al. 1996). Most of the constrained parameters in this study were directly associated with bacterial processes, with overall better model-observation fits for bacterial data types, giving confidence in the simulated bacterial C stocks and flows.

Optimization also sheds light on major unknown parameters in the bacterial grazing process, including $g_{HNA}$ and $g_{LNA}$ (half-saturation density of HNA and LNA bacteria in microzooplankton grazing, respectively). Microzooplankton grazing of the given bacterial group was simulated using Holling Type 2 density-dependent grazing with a preferential prey selection on diatoms, cryptophytes, and the other bacterial group, in which the single microzooplankton maximum grazing rate is implemented for both bacterial groups for model simplicity purposes (Text S1, Tables S2-6). Thus, it was the half-saturation

density that determined the degree of preferential grazing by microzooplankton on a certain bacterial group, the change of which would ultimately depend on biomass of each bacterial group. Due to the lack of a priori knowledge on the relative magnitude of $g_{HNA}$ and $g_{LNA}$, we assigned the identical initial guess value (Table 1) to let the data assimilation scheme find the values that best fit overall observations of the bacterial group-specific dynamics. Compared to $g_{LNA}$, smaller optimized $g_{HNA}$ (Tables S2-6) reflected preferential grazing of HNA cells by microzooplankton, consistent with previous speculations that

grazers selectively remove larger and more active cells (Giorgio et al. 1996; Gonzalez et al. 1990; Sherr et al. 1992), so HNA cells (Garzio et al. 2013). Together with the higher mean cell-specific grazing rates for the HNA group (section 3.2), our results suggest preferential grazing of HNA cells by microzooplankton.

         In this study, the model portability index reflects the extent to which a single model framework represented by model parameters and equations captures the observed variability in different years, given variable environmental forcing and the

370 accompanying shift in plankton ecosystem structure. The optimized model parameter set for 2012-13 was most portable and the optimized parameter set for 2011-12 was the least portable (Table 2), in which the most ($n$ = 7 out of total 11) and the least numbers ($n$ = 5 out of total 11) of parameters were constrained (i.e., optimized with low uncertainties), respectively (Tables S3-4). The other two years exhibited intermediate levels of portability, wth similar portability index values characterized by the same number of constrained parameters ($n$ = 6 out of total 10 for 2010-11 and $n$ = 6 out of total 12 for 2013-14; Tables S2,

S5). In other words, it was the number of well-constrained parameters that mattered most in driving high model portability, suggesting the connection between overfitting and portability of optimized models in this study. Also, varying degrees of model portability across the years rendered it difficult to select one particular year's model solution representing the climatological dynamics, consistent with the characteristics of the original WAP-1D-VAR model. Instead, better model skill was found by utilizing parameters from assimilating climatological observations into a more general version of the model

(section 4.4).

## 4.2 Bacterial carbon stocks and flows

         Assimilating each bacterial group's biomass allows for the partitioning of the BP for each group as well as other physiological processes (e.g., SDOC uptake rates) that were never measured in this study. First, optimized models yielded in

significantly higher cell-specific BP of the HNA group across all years, which could be attributed to the way the parameter optimization was performed to keep higher maximum cell-specific growth rates of HNA cells. However, it should also be noted that the cell-specific BP rates were driven by biomass stocks that were determined from the modelled trophic interactions. As with phylogenetic groups (Fuchs et al. 2000; Teira et al. 2009; Yokokawa et al. 2004), cell-specific bacterial growth rates are expected to differ among distinct bacterial physiological groups, but there are limited studies focusing on group-specific

cell activities (Gasol et al. 1999; Giorgio et al. 1996; Günter et al. 2008; Longnecker et al. 2005; Moràn et al. 2011). Moràn et al (2011) showed that HNA cells greatly outgrew LNA cells in Waquoit Bay Estuary, with a cell-specific growth rate of up to 2.26 d$^{-1}$ for HNA cells versus < 0.5 d$^{-1}$ for LNA cells. Second, our model results revealed HNA group's significantly higher

uptake rates of both LDOC and SDOC pools than their LNA counterparts. Several studies have demonstrated that HNA cells depend more than LNA cells on phytoplankton substrates for growth and metabolism (Li et al. 1995; Morán et al. 2007; Scharek and Latasa 2007). The hypothesis that WAP bacteria might rely on SDOC has received indirect support previously, presumably due to LDOC limitation (Ducklow et al. 2011; Kim and Ducklow 2016; Luria et al. 2017), but our study is the first to show the importance of the SDOC pool for HNA cells' C demand.

Although much of the discussion focuses on bacteria, the model also captured well the rest of the ecosystem variables. Modelled nutrient stocks were above detect limits and indicated the lack of macronutrient limitations. The WAP typically exhibits strong interannual variability (Ducklow et al. 2007), but the lack of the strong interannual variability in model microzooplankton biomass is due to assimilating climatological observations. One exception is krill biomass that was modelled 3-8 times larger than the maximum value from the available field data in 2017-18 (0.57 mmol C m$^{-3}$; not shown). It should be noted that there were inconsistences in the nature of the assimilated data types, such as a single-year observation of microzooplankton (versus each year-specific observations of others) and two unassimilated data types including krill biomass. Also, there can be compensating errors in krill grazing rate and metabolism values given that krill are mobile laterally. These observational limitations made it challenging to construct a complete microbial carbon budget without significant uncertainties. A more complete assimilation of zooplankton data should be the next effort to improve the model fits and minimize uncertainties in the bacterial variables and, therefore, to expand the current study.

## 4.3 Bacterial physiological and taxonomic association with ecosystem functions

The positive associations of the observed *f*HNA with the modelled NPP and POC sinking flux suggest a relatively strong resource control on these actively-growing cells, compared to slowly-growing LNA cells. This is consistent with previous studies that showed increased HNA growth rates in response to enhanced phytoplankton-derived organic substrate (Morán et al. 2010) and more abundant HNA cells in areas or periods where bacterial assemblages were predominantly controlled by resources, rather than grazing (Morán et al. 2007). It has been hypothesized that due to minimal inputs of terrestrial organic matter, bacteria must ultimately rely on *in situ* NPP for organic matter source in the WAP (Ducklow et al. 2012b), supporting the importance of resource control on actively-growing bacteral populations.

In this study, modes 3, 5, and 7, characterized by copiotrophic taxa with large genomes and more 16S rRNA gene copies (Bowman et al. 2017), were associated with high modelled NPP, POC sinking flux, and BCD, while modes 4 and 6, characterized by taxa associated with more oligotrophic conditions, were associated with low modelled NPP, POC sinking flux, and BCD. *Dokdonia* sp. MED134, a common bacterial species of the modes associated with high NPP, POC sinking flux, and BCD, is a proteorhodopsin-containing marine flavobacterium that grows faster with light (Gómez-Consarnau et al. 2007; Kimura et al. 2011) and in conditions under which resources are abundant (Gómez-Consarnau et al. 2007). Given the coastal WAP being primarily light-limited (Ducklow et al. 2012), the correspondence of *D. Dokdonia* MED134 to high modelled NPP suggests light-enhanced growth rates and cell yields from sufficient irradiance. By contrast, mode 4, dominated by *Planktomarina temperata* RCA23, is a slowly growing bacterium that specializes in using complex organic substrates (Giebel et al. 2013). These attributes are consistent with high occurrence of mode 4 during the periods of low modelled NPP and POC sinking flux. *Candidatus* Pelagibacter, abundant in mode 6, is generally known as an oligotrophic specialist with a low DOC requirement, but often observed during the Antarctic phytoplankton blooms (Delmont et al. 2014; Luria et al. 2014), the characteristics of which support its occurrence during the periods of high modelled NPP.

In summary, our study provides a novel framework connecting the dynamics of different ecosystem functions with microbial physiology and taxonomy. Certain modes represent distinct WAP ecosystem states and the mode-state associations are reasonably explained from microbial perspectives. However, we did not investigate a seasonal succession and development in mode itself or the mode association of the key WAP ecosystem states. Future investigations should focus on including a

few dominant or seasonally distinct modes in data assimilation, in order to fully resolve the seasonality of the mode-ecosystem state associations along the WAP.

## 4.4 Climate change experiments

The WAP has experienced significant atmospheric and ocean warming and resulting changes in marine ecosystem processes and further climate change is projected for the next several decades. The magnitudes of the perturbations used in the climate change experiments (+0.5º/+1.0ºC compared to observed temperature fields and -5%/-10% compared to observed sea-ice fields) are within the range of the long-term changes in temperature and sea-ice duration along the WAP continental shelf. The temperature of the ACC water that has direct access to the WAP shelf has shown a large increase after the 1980s, equivalent to a uniform warming of the upper 300 m layer by 0.7ºC (Ducklow et al., 2012). The trend in the annual ice season duration is -1.5 days per year over 1979-80 to 2017-18 field season (Henley et al. 2019). The degree of melting (5-10%) chosen for the climate change experiments is translated into the shortening of the ice season duration by 1-3 days (not shown), falling within the range of the trend in Henley et al. (2019).

Under combined warming/melting conditions, we expected that increased NPP and phytoplankton accumulations early in the season would result in a significant build-up of DOC pools. This was the case only for SDOC, and bacteria were soon LDOC-limited due to their preferential LDOC uptake for their primary carbon source. The growth of bacteria and increased bacterial rates during LDOC limitation was because bacteria depended on SDOC to meet the rest of their carbon demand, resulting in the depletion of SDOC pool later in the season. In other words, bacteria were more likely resource-limited, in particular by the labile DOC pool, and SDOC subsequently played an increasingly important role. This change was particularly important in HNA cells, as shown by a relatively large increase of their BCD via SDOC, compared to LNA cells. Temperature is often regarded as a major factor regulating bacterial physiological rates by changing the rate of enzymatic reactions (Kirchman et al. 2009; White et al. 1991). In this study the modelled stocks and rates of HNA cells increased under the warming alone experiment (Figure S10) but equally or more than under the melting alone experiment (i.e., increased photosynthesis and resource availability; Figure S11). This suggests that temperature *per se* is not necessarily a more important limiting factor for bacterial, at least HNA, growth than resource availability (Ducklow et al. 2012a), and warming may rather enhance HNA utilization of the already increased organic matter from the increased productivity. Also, future climate may impact the (re)distribution of bacterial taxonomic groups, with a potential shift to more abundant HNA cells in the WAP bacterial communities, due to their preferential SDOC utilization.

The major limitation of our climate change experiments is the duration of the simulations. An ideal set of climate change simulations should be performed for longer-term periods as well as continuous across many years, not simply limited to growth seasons. This study could not accommodate these requirements due to the limited observations and existing data gaps in each year. Despite this limitation, we were able to validate the climatological model's capacity to partly reproduce the already observed, climate-driven trends of some variables along the WAP. Under each year's forcing fields, the climatological model parameter set reproduced the interannual variability fairly well compared to the observed interannual variability, except for a few cases (e.g., overestimated BP and HNA biomass in 2011-12, underestimated PP in 2012-13 and 2013-14; Table S7), providing confidence in its usage for climate change impacts. 2011-12 was characterized by the negative temperature anomaly (-0.13 ± 0.83ºC versus 0.03 ± 0.84ºC for the 4-year climatology) and the positive sea-ice anomaly (24 ± 38% versus 21 ± 29% for the 4-year climatology), with lower temperature and higher sea-ice cover than the other three years (all $p < 0.05$, two-sample $t$-test). This coldest year had the lowest values of BP, HNA biomass, and PP observations (Table S7), consistent with increases in the modelled BP, HNA biomass, and PP under the combined warming/melting conditions. A combination of low HNA biomass, low PP, and low POC flux was also modelled in 2011-12, largely responsible for driving the positive association of the observed $f$HNA with the modelled NPP and POC sinking across years (section 4.3). Sea ice did not retreat until mid-December in 2011-12 (Figure S1), and due to subsequently low light levels PP was modelled to be low. The low modelled PP

drove both low HNA biomass and low particle export, reinforcing the strong resource control on fast-growing bacterial populations and the conventional "high PP-high export" paradigm along the WAP.

Finally, our climate change simulations share similar results with those performed with the WAP-1D-VAR model with one bacterial compartment (Kim et al. 2021). In the original model, simultaneous combined warming and reduced sea-ice conditions resulted in increased NPP, net community production, POC sinking flux, bulk bacterial productivity and biomass, and SDOC, in contrast to LDOC that was strongly limited early in the season. This potential shift to a more productive and efficient export system state is partially in agreement with suggestions made by previous studies that warming may induce more recycling favourable and microbial-dominated food-webs (Moline et al. 2004; Sailley et al. 2013). Despite the increased productivity and plankton accumulations, LDOC may become strongly depleted and, therefore, bacteria may need to depend more on SDOC to meet a significant part of their carbon demand (i.e., an increasing important role of SDOC for bulk bacterial communities). Most of these results convey the same story as this study's climate change experiments, thereby adding confidence in the climate change simulations. However, it should be noted that the increased complexity of bacterial dynamics in the modified model adds two important contributions to the original model including: 1) the dominance of the HNA group over the LNA group in the warming WAP waters and 2) bacterial taxonomic (i.e., mode) and physiological (i.e., $f$HNA) traits being a significant indicator of key WAP ecosystem functions.

## 5 Conclusions

Heterotrophic microbial diversity has seldom been considered in detail in the formulation and analysis of marine pelagic ecosystem models reflecting in part the lack of suitable field data for model evaluation. Utilizing genomic products to prescribe taxonomic aspects of the bacterial model dynamics, this study investigated the association of bacterial abundance with different physiological states, bacterial community structure and key ecosystem functions. The modelling approach used in the present study enabled the observations in different bacterial populations to constrain the group-specific processes and model parameters that have been poorly understood. These included the partitioning of BP specific to HNA and LNA groups, the partitioning of the bacterial uptake of DOC pools with different lability, and the half-saturation density of each bacterial group in microzooplankton grazing. The model also served as an effective platform to explore the WAP microbial response to changing climate conditions, in which warming and decreasing sea ice would induce a potential shift to the dominance of HNA bacteria in more productive waters due to their increasing dependence on SDOC.

## Code availability

The model simulation results and codes are available in a NetCDF data structure and Fortran 90 at HHK's GitHub data repository (https://github.com/hyewon-kim-whoi).

## Data availability

Complete Palmer LTER time-series data used for data assimilation are available online (http://pal.lternet.edu/data). Surface downward solar radiation flux data used for physical forcing of the model simulations can be found in the National Centers for Environmental Prediction website (https://www.esrl.noaa.gov/psd/data/gridded/data.ncep.reanalysis.surface.html). The Tangent linear and Adjoint Model Compiler (TAPENADE) used to construct an adjoint model is available online (http://www-sop.inria.fr/tropics/).

## Author contribution

HHK designed the research, performed model simulations, and wrote the manuscript. JSB provided observational data and helped data analyses and interpretation. HWD, OMS, and DKS provided observational data. YWL contributed model simulations. SCD supervised the research and significantly revised the manuscript.

**Competing interest**

The authors declare that they have no conflict of interest.

**Acknowledgements**

This study leverages the wealth of marine data collected by the Palmer Long-Term Ecological Research (Palmer LTER) program along the west Antarctic Peninsula, and the authors thank the scientists, students, technicians, station support and
525 logistical staff, and ship captains, officers and crew involved. This research was supported, in part, by the U.S. National Science Foundation Office of Polar Programs through award NSF PLR-1440435 and the U.S. National Aeronautics and Space Administration Ocean Biology and Biogeochemistry Program through award NASA NNX14AL86G. HK was also supported by the Investment in Science Fund and the Reuben F. and Elizabeth B. Richards Endowed Fund from Woods Hole Oceanographic Institution.

**Appendix A**

**1. Temperature effect**

$$T_f = \exp\{ -A_E \times (1/T - 1/T_{ref}) \} \tag{A.1.1}$$

**2. Diatom processes**

- Cellular quota (ratio):

$$Q^C_{N,DA} = N_{DA}/C_{DA} \tag{A.2.1}$$
$$Q^C_{P,DA} = P_{DA}/C_{DA} \tag{A.2.2}$$
$$Q^C_{CHL,DA} = CHL_{DA}/C_{DA} \tag{A.2.3}$$

- N and P limitation function:

$$N_{f,DA} = (Q^C_{N,DA} - q^C_{N,MIN,DA})/(q^C_{N,RDF,DA} - q^C_{N,MIN,DA}) \qquad 0 \le N_{f,DA} \le 1 \tag{A.2.4}$$
$$P_{f,DA} = (Q^C_{P,DA} - q^C_{P,MIN,DA})/(q^C_{P,RDF,DA} - q^C_{P,MIN,DA}) \qquad 0 \le P_{f,DA} \le 1 \tag{A.2.5}$$

- Maximum photosynthesis rate:

$$P^C_{MAX} = \mu_{DA} \times T_f \times \min(N_{f,DA},\ P_{f,DA}) \tag{A.2.6}$$

- C-specific gross primary production:

$$G^C_{DA} = C_{DA} \times P^C_{MAX} \times\{ 1- \exp(-\alpha_{DA} \times Q^C_{CHL} \times PAR)/P^C_{MAX}) \} \times \exp(-\beta_{DA} \times PAR) \tag{A.2.7}$$

- Limitation on N and P uptake:

$$V^N_{MAX} = (q^C_{N,MAX} - Q^C_{N,DA})/(q^C_{N,MAX} - q^C_{N,RDF}) \qquad 0 \le V^N_{MAX} \le 1 \tag{A.2.8}$$
$$V^P_{MAX} = (q^C_{P,MAX} - Q^C_{P,DA})/(q^C_{P,MAX} - q^C_{P,RDF}) \qquad 0 \le V^P_{MAX} \le 1 \tag{A.2.9}$$

- N assimilation:

$$G^{NH4}_{DA} = C_{DA} \times V^N_{REF} \times T_f \times V^N_{MAX} \times\{ NH4/(NH4 + k^{NH4} + NO3 \times k^{NH4}/k^{NO3}) \} \tag{A.2.10}$$
$$G^{NO3}_{DA} = C_{DA} \times V^N_{REF} \times T_f \times V^N_{MAX} \times\{ NO3/(NO3 + k^{NO3} + NH4 \times k^{NO3}/k^{NH4}) \} \tag{A.2.11}$$
$$G^N_{DA} = G^{NH4}_{DA} + G^{NO3}_{DA} \tag{A.2.12}$$

- P assimilation:

$$G^{PO4}_{DA} = C_{DA} \times V^P_{REF} \times T_f \times V^P_{MAX} \times\{ PO4/(PO4 + k^{PO4}) \} \tag{A.2.13}$$

- Chlorophyll production:

$$G^{CHL}_{DA} = \theta \times (G^{NH4}_{DA} + G^{NO3}_{DA}) \times \{ G^C_{DA}/\alpha \times CHL_{DA} \times PAR \times \exp(-\beta \times PAR) \} \tag{A.2.14}$$

- Respiration:

$$R^C_{DA} = G^{NO3}_{DA} \times \zeta^{NO3} \tag{A.2.15}$$

- Passive excretion of LDOM:

$$E^C_{DA,LDOC,PSV} = ex_{DA,PSV} \times C_{DA} \tag{A.2.16}$$

$$E^N_{DA,LDON,PSV} = ex_{DA,PSV} \times N_{DA} \tag{A.2.17}$$

$$E^P_{DA,LDOP,PSV} = ex_{DA,PSV} \times P_{DA} \tag{A.2.18}$$

- Active excretion of LDOC:

$$E^C_{DA,LDOC,ACT} = ex_{DA,ACT} \times G^C_{DA} \tag{A.2.19}$$

- Active excretion of SDOC:

$$E^C_{DA,SDOC,ACT} = 0.5 \times C_{DA} \times \max( 1 - Q^C_{N,DA}/q^C_{N,RDF,DA}, 1 - Q^C_{P,DA}/q^C_{P,RDF,DA}, 0 ) \tag{A.2.20}$$

- Active excretion of SDON and SDOP (if $EX^C_{DA,SDOC,ACT} > 0$, otherwise 0):

$$E^N_{DA,SDON,ACT} = 0.5 \times 0.25 \times N_{DA} \times \max( 1 - Q^N_{P,DA}/q^C_{P,RDF,DA}/q^C_{N,RDF,DA}, 0 ) \tag{A.2.21}$$

$$E^P_{DA,SDOP,ACT} = 0.5 \times 0.25 \times P_{DA} \times \max( 1 - Q^P_{N,DA}/q^C_{N,RDF,DA}/q^C_{P,RDF,DA}, 0 ) \tag{A.2.22}$$

- Partitioning between LDOM and SDOM:

$$E^C_{DA,LDOC} = E^C_{DA,LDOC,PSV} + 0.75 \times E^C_{DA,LDOC,ACT} \tag{A.2.23}$$

$$E^N_{DA,LDON} = E^N_{DA,LDON,PSV} \tag{A.2.24}$$

$$E^P_{DA,LDOP} = E^P_{DA,LDOP,PSV} \tag{A.2.25}$$

$$E^C_{DA,SDOC} = E^C_{DA,SDOC,ACT} + 0.25 \times E^C_{DA,LDOC,ACT} \tag{A.2.26}$$

$$E^N_{DA,SDON} = E^N_{DA,SDON,ACT} \tag{A.2.27}$$

$$E^P_{DA,SDOP} = E^P_{DA,SDOP,ACT} \tag{A.2.28}$$

- POM production by aggregation:

$$D^C_{DA} = pom_{DA} \times C_{DA} \times C_{DA} \tag{A.2.29}$$

$$D^N_{DA} = Q^C_{N,DA} \times D^C_{DA} \tag{A.2.30}$$

$$D^P_{DA} = Q^C_{P,DA} \times D^C_{DA} \tag{A.2.31}$$

$$D^{CHL}_{DA} = Q^C_{CHL,DA} \times D^C_{DA} \tag{A.2.32}$$

- Grazing by microzooplankton:

$$GZ^C_{DA,MZ} = T_f \times \mu_{MZ} \times C_{MZ}$$
$$\times [(C_{DA} - \epsilon_{DA})^2/\{(C_{DA} - \epsilon_{DA})^2 + g_{DA}^2 + (C_{CR} \times g_{DA}/g_{CR})^2 + (C_{BAC} \times g_{DA}/g_{BAC})^2\}] \tag{A.2.33}$$

$$GZ^N_{DA,MZ} = Q^C_{N,DA} \times GZ^C_{DA,MZ} \tag{A.2.34}$$

$$GZ^P_{DA,MZ} = Q^C_{P,DA} \times GZ^C_{DA,MZ} \tag{A.2.35}$$

$$GZ^{CHL}_{DA,MZ} = Q^C_{CHL,DA} \times GZ^C_{DA,MZ} \tag{A.2.36}$$

- Grazing by krill:

$$GZ^C_{DA,KR} = T_f \times \mu_{KR} \times C_{KR}$$
$$\times [ C_{DA}^2/\{C_{DA}^2 + g'_{DA}^2 + (C_{MZ} \times g'_{DA}/g_{MZ})^2\} ] \tag{A.2.37}$$

$$GZ^N_{DA,KR} = Q^C_{N,DA} \times GZ^C_{DA,KR} \tag{A.2.38}$$

$$GZ^P_{DA,KR} = Q^C_{P,DA} \times GZ^C_{DA,KR} \tag{A.2.39}$$

$$GZ^{CHL}_{DA,KR} = Q^C_{CHL,DA} \times GZ^C_{DA,KR} \tag{A.2.40}$$

- The net growth rate equations:

$$\frac{dC_{DA}}{dt} = G^C_{DA} - E^C_{DA,LDOC} - E^C_{DA,SDOC} - D^C_{DA} - R^C_{DA} - GZ^C_{DA,MZ} - GZ^C_{DA,KR} \tag{A.2.41}$$

$$\frac{dN_{DA}}{dt} = G^N{}_{DA} - E^N{}_{DA,LDON} - E^N{}_{DA,SDON} - D^N{}_{DA} \qquad - GZ^N{}_{DA,MZ} - GZ^N{}_{DA,KR} \qquad (A.2.42)$$

$$\frac{dP_{DA}}{dt} = G^P{}_{DA} - E^P{}_{DA,LDOP} - E^P{}_{DA,SDOP} - D^N{}_{DA} \qquad - GZ^P{}_{DA,MZ} - GZ^P{}_{DA,KR} \qquad (A.2.43)$$

$$\frac{dCHL_{DA}}{dt} = G^{CHL}{}_{DA} \qquad\qquad - D^{CHL}{}_{DA} \qquad - GZ^{CHL}{}_{DA,MZ} - GZ^{CHL}{}_{DA,KR} \qquad (A.2.44)$$

## 3. Cryptophyte processes

- Cellular quota (ratio):

$$Q^C{}_{N,CR} = N_{CR}/C_{CR} \qquad\qquad (A.3.1)$$

$$Q^C{}_{P,CR} = P_{CR}/C_{CR} \qquad\qquad (A.3.2)$$

$$Q^C{}_{CHL,CR} = CHL_{CR}/C_{CR} \qquad\qquad (A.3.3)$$

- N and P limitation function:

$$N_{f,CR} = (Q^C{}_{N,CR} - q^C{}_{N,MIN,CR})/(q^C{}_{N,RDF,CR} - q^C{}_{N,MIN,CR}) \qquad 0 \le N_{f,CR} \le 1 \qquad (A.3.4)$$

$$P_{f,CR} = (Q^C{}_{P,CR} - q^C{}_{P,MIN,CR})/(q^C{}_{P,RDF,CR} - q^C{}_{P,MIN,CR}) \qquad 0 \le P_{f,CR} \le 1 \qquad (A.3.5)$$

- Maximum primary production rate:

$$P^C{}_{MAX} = \mu_{CR} \times T_f \times \min(N_{f,CR}, P_{f,CR}) \qquad\qquad (A.3.6)$$

- C-specific gross primary production:

$$G^C{}_{CR} = C_{CR} \times P^C{}_{MAX} \times \{ 1 - \exp(-\alpha_{CR} \times Q^C{}_{CHL} \times PAR)/P^C{}_{MAX}) \} \times \exp(-\beta_{CR} \times PAR) \qquad (A.3.7)$$

- Limitation on N and P uptake:

$$V^N{}_{MAX} = (q^C{}_{N,MAX} - Q^C{}_{N,CR})/(q^C{}_{N,MAX} - q^C{}_{N,RDF}) \quad 0 \le V^N{}_{MAX} \le 1 \qquad (A.3.8)$$

$$V^P{}_{MAX} = (q^C{}_{P,MAX} - Q^C{}_{P,CR})/(q^C{}_{P,MAX} - q^C{}_{P,RDF}) \qquad 0 \le V^P{}_{MAX} \le 1 \qquad (A.3.9)$$

- Nitrogen assimilation:

$$G^{NH4}{}_{CR} = C_{CR} \times V^N{}_{REF} \times T_f \times V^N{}_{MAX} \times \{ NH4/(NH4 + k^{NH4} + NO3 \times k^{NH4}/k^{NO3}) \} \qquad (A.3.10)$$

$$G^{NO3}{}_{CR} = C_{CR} \times V^N{}_{REF} \times T_f \times V^N{}_{MAX} \times \{ NO3/(NO3 + k^{NO3} + NH4 \times k^{NO3}/k^{NH4}) \} \qquad (A.3.11)$$

$$G^N{}_{CR} = G^{NH4}{}_{CR} + G^{NO3}{}_{CR} \qquad\qquad (A.3.12)$$

- Phosphorus assimilation:

$$G^{PO4}{}_{CR} = C_{CR} \times V^P{}_{REF} \times T_f \times V^P{}_{MAX} \times \{ PO4/(PO4 + k^{PO4}) \} \qquad (A.3.13)$$

- Chlorophyll production:

$$G^{CHL}{}_{CR} = \theta \times (G^{NH4}{}_{CR} + G^{NO3}{}_{CR}) \times \{ G^C{}_{CR}/\alpha \times CHL_{CR} \times PAR \times \exp(-\beta \times PAR) \} \qquad (A.3.14)$$

- Respiration:

$$R^C{}_{CR} = G^{NO3}{}_{CR} \times \zeta^{NO3} \qquad\qquad (A.3.15)$$

- Passive excretion of LDOM:

$$E^C{}_{CR,LDOC,PSV} = ex_{CR,PSV} \times C_{CR} \qquad\qquad (A.3.16)$$

$$E^N{}_{CR,LDON,PSV} = ex_{CR,PSV} \times N_{CR} \qquad\qquad (A.3.17)$$

$$E^P{}_{CR,LDOP,PSV} = ex_{CR,PSV} \times P_{CR} \qquad\qquad (A.3.18)$$

- Active excretion of LDOC:

$$E^C{}_{CR,LDOC,ACT} = ex_{CR,ACT} \times G^C{}_{CR} \qquad\qquad (A.3.19)$$

- Active excretion of SDOC:

$$E^C{}_{CR,SDOC,ACT} = 0.5 \times C_{CR} \times \max( 1 - Q^C{}_{N,CR}/q^C{}_{N,RDF,CR}, 1 - Q^C{}_{P,CR}/q^C{}_{P,RDF,CR}, 0 ) \qquad (A.3.20)$$

- Active excretion of SDON and SDOP (if $EX^C{}_{CR,SDOC,ACT} > 0$, otherwise 0):

$$E^N_{CR,SDON,ACT} = 0.5 \times 0.25 \times N_{CR} \times \max(1 - Q^N_{P,CR}/q^C_{P,RDF,CR}/q^C_{N,RDF,CR}, 0) \qquad (A.3.21)$$

$$E^P_{CR,SDOP,ACT} = 0.5 \times 0.25 \times P_{CR} \times \max(1 - Q^P_{N,CR}/q^C_{N,RDF,CR}/q^C_{P,RDF,CR}, 0) \qquad (A.3.22)$$

- Partitioning between LDOM and SDOM:

$$E^C_{CR,LDOC} = E^C_{CR,LDOC,PSV} + 0.75 \times E^C_{CR,LDOC,ACT} \qquad (A.3.23)$$

$$E^N_{CR,LDON} = E^N_{CR,LDON,PSV} \qquad (A.3.24)$$

$$E^P_{CR,LDOP} = E^P_{CR,LDOP,PSV} \qquad (A.3.25)$$

$$E^C_{CR,SDOC} = E^C_{CR,SDOC,ACT} + 0.25 \times E^C_{CR,LDOC,ACT} \qquad (A.3.26)$$

$$E^N_{CR,SDON} = E^N_{CR,SDON,ACT} \qquad (A.3.27)$$

$$E^P_{CR,SDOP} = E^P_{CR,SDOP,ACT} \qquad (A.3.28)$$

- POM production by aggregation:

$$D^C_{CR} = pom_{CR} \times C_{CR} \times C_{CR} \qquad (A.3.29)$$

$$D^N_{CR} = Q^C_{N,CR} \times A^C_{CR} \qquad (A.3.30)$$

$$D^P_{CR} = Q^C_{P,CR} \times A^C_{CR} \qquad (A.3.31)$$

$$D^{CHL}_{CR} = Q^C_{CHL,CR} \times A^C_{CR} \qquad (A.3.32)$$

- Grazing by microzooplankton:

$$GZ^C_{CR} = T_f \times \mu_{MZ} \times C_{MZ}$$
$$\times [(C_{CR} - \epsilon_{CR})^2/\{(C_{CR} - \epsilon_{CR})^2 + g_{CR}{}^2 + (C_{DA} \times g_{CR}/g_{DA})^2 + (C_{BAC} \times g_{CR}/g_{BAC})^2\}] \qquad (A.3.33)$$

$$GZ^N_{CR} = Q^C_{N,CR} \times GZ^C_{CR,MZ} \qquad (A.3.34)$$

$$GZ^P_{CR} = Q^C_{P,CR} \times GZ^C_{CR,MZ} \qquad (A.3.35)$$

$$GZ^{CHL}_{CR} = Q^C_{CHL,CR} \times GZ^C_{CR,MZ} \qquad (A.3.36)$$

- The net growth rate equations:

$$\frac{dC_{CR}}{dt} = G^C_{CR} - E^C_{CR,LDOC} - E^C_{CR,SDOC} - D^C_{CR} - R^C_{CR} - GZ^C_{CR} \qquad (A.3.37)$$

$$\frac{dN_{CR}}{dt} = G^N_{CR} - E^N_{CR,LDON} - E^N_{CR,SDON} - D^N_{CR} \qquad - GZ^N_{CR} \qquad (A.3.38)$$

$$\frac{dP_{CR}}{dt} = G^P_{CR} - E^P_{CR,LDOP} - E^P_{CR,SDOP} - D^N_{CR} \qquad - GZ^P_{CR} \qquad (A.3.39)$$

$$\frac{dCHL_{CR}}{dt} = G^{CHL}_{CR} \qquad - D^{CHL}_{CR} \qquad - GZ^{CHL}_{CR} \qquad (A.3.40)$$

## 4. Bacterial processes (for both HNA and LNA groups)

- Cellular quota (ratio):

$$Q^C_{N,BAC} = N_{BAC}/C_{BAC} \qquad (A.4.1)$$

$$Q^C_{P,BAC} = P_{BAC}/C_{BAC} \qquad (A.4.2)$$

$$Q^P_{N,BAC} = N_{BAC}/P_{BAC} \qquad (A.4.3)$$

$$Q^C_{N,LDOM} = N_{LDOM}/C_{LDOM} \qquad (A.4.4)$$

$$Q^C_{P,LDOM} = P_{LDOM}/C_{LDOM} \qquad (A.4.5)$$

$$Q^C_{N,SDOM} = N_{SDOM}/C_{SDOM} \qquad (A.4.6)$$

$$Q^C_{P,SDOM} = P_{SDOM}/C_{SDOM} \qquad (A.4.7)$$

- N and P limitation function:

$$N_{f,BAC} = Q^C_{N,BAC}/q^C_{N,BAC} \qquad\qquad 0 \le N_{f,BAC} \le 1 \qquad (A.4.8)$$

$$P_{f,BAC} = Q^C_{P,BAC}/q^C_{P,BAC} \qquad\qquad 0 \le P_{f,BAC} \le 1 \qquad (A.4.9)$$

- Maximum available LDOC and SDOC:

$$ALC = C_{LDOC} \tag{A.4.10}$$

$$ASC = r_{SDOC} \times C_{SDOC} \tag{A.4.11}$$

- Bacterial uptake of LDOC and SDOC (i.e., bacterial gross C growth):

$$G^C_{BAC,LDOC} = \mu_{BAC} \times T_f \times C_{BAC} \times \min(N_{f,BAC}, P_{f,BAC}) \times \{\, ALC/(ALC + k^{DOC} + ASC)\, \} \tag{A.4.12}$$

$$G^C_{BAC,SDOC} = \mu_{BAC} \times T_f \times C_{BAC} \times \min(N_{f,BAC}, P_{f,BAC}) \times \{\, ASC/(ASC + k^{DOC} + ALC)\, \} \tag{A.4.13}$$

$$G^C_{BAC,DOC} = G^C_{BAC,LDOC} + G^C_{BAC,SDOC} \tag{A.4.14}$$

- Bacterial N uptake:

$$G^C_{BAC,LDON} = G^C_{BAC,LDOC} \times Q^C_{N,LDOM} \tag{A.4.15}$$

$$G^C_{BAC,SDON} = G^C_{BAC,SDOC} \times \min\{\, q^C_{N,BAC}, Q^C_{N,SDOM} + f_S/N_{f,BAC} \times (q^C_{N,BAC} - Q^C_{N,SDOM})\, \} \tag{A.4.16}$$

$$G^C_{BAC,NH4} = G^C_{BAC,LDON} \times NH4/N_{LDOM}/\min(1, N_{f,BAC}) \tag{A.4.17}$$

if $N_{f,BAC} < 1$,

$$G^C_{BAC,NO3} = \min\{\, 0.1 \times NO3 \times 1/\min(1, N_{f,BAC}) \times (G^C_{BAC,LDON} + G^C_{BAC,SDON})/(N_{LDOM} + N_{SDOM}),$$
$$(NO3 + NH4) \times (G^N_{BAC,LDON} + G^N_{BAC,SDON})/(N_{LDOM} + N_{SDOM}) - G^{NH4}_{BAC}\, \} \tag{A.4.18}$$

else, $G^C_{BAC,NO3} = 0$ \hfill (A.4.19)

$$G^C_{BAC,N} = G^C_{BAC,LDON} + G^C_{BAC,SDON} + G^C_{BAC,NH4} + G^C_{BAC,NO3} \tag{A.4.20}$$

- Bacterial P uptake:

$$G^C_{BAC,LDOP} = G^C_{BAC,LDOC} \times Q^C_{P,LDOM} \tag{A.4.21}$$

$$G^C_{BAC,SDOP} = G^C_{BAC,SDOC} \times \min\{\, q^C_{P,BAC}, Q^C_{P,SDOM} + f_S/P_{f,BAC} \times (q^C_{P,BAC} - Q^C_{P,SDOM})\, \} \tag{A.4.22}$$

$$G^C_{BAC,PO4} = G^C_{BAC,LDON} \times PO4/P_{LDOM}/\min(1, P_{f,BAC}) \tag{A.4.23}$$

$$G^C_{BAC,P} = G^C_{BAC,LDOP} + G^C_{BAC,SDOP} + G^C_{BAC,PO4} \tag{A.4.24}$$

- Respiration:

$$R^C_{BAC} = \zeta^{NO3} \times G^C_{BAC,NO3} + r^B_{BAC} \times T_f \times C_{BAC}$$
$$+ \{\, r^A_{min,BAC} + (r^A_{max,BAC} - r^A_{min,BAC}) \times \exp(-b_{R,BAC} \times G^C_{BAC,DOC})\, \} \times G^C_{BAC,DOC} \tag{A.4.25}$$

- RDOC release:

$$E^C_{BAC,RDOC} = refr_{BAC} \times C_{BAC} \tag{A.4.26}$$

$$E^N_{BAC,RDON} = E^C_{BAC,RDOC} \times q^C_{N,RDOM} \tag{A.4.27}$$

$$E^P_{BAC,RDOP} = E^C_{BAC,RDOC} \times q^C_{P,RDOM} \tag{A.4.28}$$

- Remineralization of inorganic nutrients:

if $Q^C_{N,BAC} > q^C_{N,BAC}$ and $Q^C_{P,BAC} > q^C_{P,BAC}$ (i.e., C in short)

$$REMI^N_{BAC} = remi_{BAC} \times (N_{BAC} - C_{BAC} \times q^C_{N,BAC}) \tag{A.4.29}$$

$$REMI^P_{BAC} = remi_{BAC} \times (P_{BAC} - C_{BAC} \times q^C_{P,BAC}) \tag{A.4.30}$$

$$remi_{BAC}$$

elseif $Q^C_{N,BAC} < q^C_{N,BAC}$ and $Q^P_{N,BAC} < q^C_{N,BAC}/q^C_{P,BAC}$ (i.e., N in short)

$$REMI^N_{BAC} = 0 \tag{A.4.31}$$

$$REMI^P_{BAC} = 0 \tag{A.4.32}$$

else (i.e., P in short)

$$REMI^N_{BAC} = 0 \tag{A.4.33}$$

$$REMI^P_{BAC} = 0 \tag{A.4.34}$$

- SDOM excretion to adjust stoichiometry:

if $Q^C_{N,BAC} > q^C_{N,BAC}$ and $Q^C_{P,BAC} > q^C_{P,BAC}$ (i.e., C in short)

$$E^C_{BAC,SDOC} = 0 \tag{A.4.35}$$

$$E^N_{BAC,SDON} = 0 \tag{A.4.36}$$

$$E^P_{BAC,SDOP} = 0 \tag{A.4.37}$$

elseif $Q^C_{N,BAC} < q^C_{N,BAC}$ and $Q^P_{N,BAC} < q^C_{N,BAC}/q^C_{P,BAC}$ (i.e., N in short)

$$E^C_{BAC,SDOC} = ex_{ADJ,BAC} \times (C_{BAC} - N_{BAC}/q^C_{N,BAC}) \tag{A.4.38}$$

$$E^N_{BAC,SDOC} = 0 \tag{A.4.39}$$

$$E^P_{BAC,SDOP} = ex_{ADJ,BAC} \times (P_{BAC} - N_{BAC}/q^C_{N,BAC} \times q^C_{P,BAC}) \tag{A.4.40}$$

else (i.e., P in short)

$$E^C_{BAC,SDOC} = ex_{ADJ,BAC} \times (C_{BAC} - P_{BAC}/q^C_{P,BAC}) \tag{A.4.41}$$

$$E^N_{BAC,SDON} = ex_{ADJ,BAC} \times (N_{BAC} - P_{BAC}/q^C_{P,BAC} \times q^C_{N,BAC}) \tag{A.4.42}$$

$$E^N_{BAC,SDOP} = 0 \tag{A.4.43}$$

- Grazing by microzooplankton:

$$GZ^C_{BAC} = T_f \times \mu_{MZ} \times C_{MZ}$$
$$\times [\ C_{BAC}^2/\{C_{CR}^2 + g_{BAC}^2 + (C_{DA} \times g_{BAC}/g_{DA})^2 + (C_{CR} \times g_{BAC}/g_{CR})^2\}\ ] \tag{A.4.44}$$

$$GZ^N_{BAC} = GZ^C_{BAC} \times Q^C_{N,BAC} \tag{A.4.45}$$

$$GZ^P_{BAC} = GZ^C_{BAC} \times Q^C_{P,BAC} \tag{A.4.46}$$

- Viral mortality:

$$M^C_{BAC} = m_{BAC} \times C_{BAC} \tag{A.4.47}$$

$$M^N_{BAC} = m_{BAC} \times N_{BAC} \tag{A.4.48}$$

$$M^P_{BAC} = m_{BAC} \times P_{BAC} \tag{A.4.49}$$

- Net flux of inorganic nutrients through bacteria:

$$FLUX^{NH4}_{BAC} = REMI^N_{BAC} - G^C_{BAC,NH4} \tag{A.4.50}$$

$$FLUX^{NO3}_{BAC} = -G^C_{BAC,NO3} \tag{A.4.51}$$

$$FLUX^{PO4}_{BAC} = REMI^P_{BAC} - G^C_{BAC,PO4} \tag{A.4.52}$$

- The net growth rate equations:

$$\frac{dC_{BAC}}{dt} = G^C_{BAC,DOC} - E^C_{BAC,SDOC} - E^C_{BAC,RDOC} - R^C_{BAC} - GZ^C_{BAC} - M^C_{BAC} \tag{A.4.53}$$

$$\frac{dN_{BAC}}{dt} = G^N_{BAC,DON} - E^N_{BAC,SDON} - E^N_{BAC,RDON} - R^N_{BAC} - GZ^N_{BAC} - M^N_{BAC} \tag{A.4.54}$$

$$\frac{dP_{BAC}}{dt} = G^P_{BAC,DOP} - E^P_{BAC,SDOP} - E^P_{BAC,RDOP} - R^P_{BAC} - GZ^P_{BAC} - M^P_{BAC} \tag{A.4.55}$$

## 5. Microzooplankton processes

- Cellular quota (ratio):

$$Q^C_{N,MZ} = N_{MZ}/C_{MZ} \tag{A.5.1}$$

$$Q^C_{P,MZ} = C_{MZ}/P_{MZ} \tag{A.5.2}$$

- Gross growth:

$$G^C_{MZ} = GZ^C_{DA,MZ} + GZ^C_{CR} + GZ^C_{BAC} \tag{A.5.3}$$

$$G^N_{MZ} = GZ^N_{DA,MZ} + GZ^N_{CR} + GZ^N_{BAC} \tag{A.5.4}$$

$$G^P_{MZ} = GZ^P_{DA,MZ} + GZ^P_{CR} + GZ^P_{BAC} \tag{A.5.5}$$

- LDOM excretion:

$$E^C_{MZ,LDOC} = f_{ex,MZ} \times ex_{MZ} \times G^C_{MZ} \tag{A.5.6}$$

$$E^N_{MZ,LDON} = f_{ex,MZ} \times ex_{MZ} \times G^N_{MZ} \tag{A.5.7}$$

$$E^P_{MZ,LDOP} = f_{ex,MZ} \times ex_{MZ} \times G^P_{MZ} \tag{A.5.8}$$

- SDOM excretion:

$$E^C_{MZ,SDOC,1} = (1 - f_{ex,MZ}) \times ex_{MZ} \times G^C_{MZ} \tag{A.5.9}$$

$$E^N_{MZ,SDON,1} = (1 - f_{ex,MZ}) \times ex_{MZ} \times G^N_{MZ} \times Q^C_{N,MZ}/q^C_{N,MZ} \tag{A.5.10}$$

$$E^P_{MZ,SDOP,1} = (1 - f_{ex,MZ}) \times ex_{MZ} \times G^P_{MZ} \times Q^C_{P,MZ}/q^C_{P,MZ} \tag{A.5.11}$$

- SDOM excretion to adjust stoichiometry:

$$E^C_{MZ,SDOC,2} = ex_{ADJ,MZ} \times C_{MZ}$$
$$\times \max(0,\, 1 - Q^C_{N,MZ}/q^C_{N,MZ},\, 1 - Q^C_{P,MZ}/q^C_{P,MZ}) \tag{A.5.12}$$

$$E^N_{MZ,SDON,2} = 0.5 \times E^C_{MZ,SDOC,2} \times Q^C_{N,MZ} \tag{A.5.13}$$

$$E^P_{MZ,SDOP,2} = 0.5 \times E^C_{MZ,SDOC,2} \times Q^C_{P,MZ} \tag{A.5.14}$$

- Remineralization of inorganic nutrients:

$$REMI^N_{MZ} = remi_{MZ} \times \max(0,\, N_{MZ} - C_{MZ} \times q^C_{N,MZ},$$
$$N_{MZ} - q^C_{N,MZ}/P_{MZ} \times q^C_{P,MZ}) \tag{A.5.15}$$

$$REMI^P_{MZ} = remi_{MZ} \times \max(0,\, P_{MZ} - C_{MZ} \times q^C_{P,MZ},$$
$$P_{MZ} - q^C_{P,MZ}/N_{MZ} \times q^C_{N,MZ}) \tag{A.5.16}$$

- Respiration:

$$R^C_{MZ} = r^B_{MZ} \times T_f \times C_{MZ} + r^A_{MZ} \times G^C_{MZ} \tag{A.5.17}$$

- POM production:

$$P^C_{MZ} = pom_{MZ} \times G^C_{MZ} \tag{A.5.18}$$

$$P^N_{MZ} = q^C_{N,POM} \times G^C_{MZ} \tag{A.5.19}$$

$$P^P_{MZ} = q^C_{P,POM} \times G^C_{MZ} \tag{A.5.20}$$

- Grazing by krill:

$$GZ^C_{MZ} = T_f \times \mu_{MZ} \times C_{KR}$$
$$\times [\, C_{MZ}^2/\{C_{MZ}^2 + g_{MZ} + (C_{DA} \times g_{MZ}/g_{DA})^2\}\, ] \tag{A.5.21}$$

$$GZ^N_{MZ} = Q^C_{N,MZ} \times GZ^C_{MZ} \tag{A.5.22}$$

$$GZ^P_{MZ} = Q^C_{P,MZ} \times GZ^C_{MZ} \tag{A.5.23}$$

- The net growth rate equations:

$$\frac{dC_{MZ}}{dt} = G^C_{MZ} - E^C_{MZ,LDOC} - E^C_{MZ,SDOC,1} - E^C_{MZ,SDOC,2}$$
$$- P^C_{MZ} - R^C_{MZ} - GZ^C_{MZ} \tag{A.5.24}$$

$$\frac{dN_{MZ}}{dt} = G^N_{MZ} - E^N_{MZ,LDON} - E^N_{MZ,SDON,1} - E^N_{MZ,SDON,2}$$
$$- P^N_{MZ} - R^N_{MZ} - GZ^N_{MZ} \tag{A.5.25}$$

$$\frac{dP_{MZ}}{dt} = G^P_{MZ} - E^P_{MZ,LDOP} - E^P_{MZ,SDOP,1} - E^P_{MZ,SDOP,2}$$
$$- P^P_{MZ} - R^P_{MZ} - GZ^P_{MZ} \tag{A.5.26}$$

## 6. Krill processes

- Cellular quota (ratio):

$$Q^C_{N,KR} = N_{KR}/C_{KR} \tag{A.6.1}$$

$$Q^C_{P,KR} = C_{KR}/P_{KR} \tag{A.6.2}$$

- Gross growth:

$$G^C_{KR} = GZ^C_{DA,KR} + GZ^C_{MZ} \tag{A.6.3}$$

$$G^N_{KR} = GZ^N_{DA,KR} + GZ^N_{MZ} \tag{A.6.4}$$
$$G^P_{KR} = GZ^P_{DA,KR} + GZ^P_{MZ} \tag{A.6.5}$$

- LDOM excretion:
$$E^C_{KR,LDOC} = f_{ex,KR} \times ex_{KR} \times G^C_{KR} \tag{A.6.6}$$
$$E^N_{KR,LDON} = f_{ex,KR} \times ex_{KR} \times G^N_{KR} \tag{A.6.7}$$
$$E^P_{KR,LDOP} = f_{ex,KR} \times ex_{KR} \times G^P_{KR} \tag{A.6.8}$$

- SDOM excretion:
$$E^C_{KR,SDOC,1} = (1 - f_{ex,KR}) \times ex_{KR} \times G^C_{KR} \tag{A.6.9}$$
$$E^N_{KR,SDON,1} = (1 - f_{ex,KR}) \times ex_{KR} \times G^N_{KR} \times Q^C_{N,KR}/q^C_{N,KR} \tag{A.6.10}$$
$$E^P_{KR,SDOP,1} = (1 - f_{ex,KR}) \times ex_{KR} \times G^P_{KR} \times Q^C_{P,KR}/q^C_{P,KR} \tag{A.6.11}$$

- SDOM excretion to adjust stoichiometry:
$$E^C_{KR,SDOC,2} = ex_{ADJ,KR} \times C_{KR}$$
$$\times \max(0, 1 - Q^C_{N,KR}/q^C_{N,KR}, 1 - Q^C_{P,KR}/q^C_{P,KR}) \tag{A.6.12}$$
$$E^N_{KR,SDON,2} = 0.5 \times E^C_{KR,SDOC,2} \times Q^C_{N,KR} \tag{A.6.13}$$
$$E^P_{KR,SDOP,2} = 0.5 \times E^C_{KR,SDOC,2} \times Q^C_{P,KR} \tag{A.6.14}$$

- Remineralization of inorganic nutrients:
$$REMI^N_{KR} = remi_{KR} \times \max(0, N_{KR} - C_{KR} \times q^C_{N,KR},$$
$$N_{KR} - q^C_{N,KR}/P_{KR} \times q^C_{P,KR}) \tag{A.6.15}$$
$$REMI^P_{KR} = remi_{KR} \times \max(0, P_{KR} - C_{KR} \times q^C_{P,KR},$$
$$P_{KR} - q^C_{P,KR}/N_{KR} \times q^C_{N,KR}) \tag{A.6.16}$$

- Respiration:
$$R^C_{KR} = r^B_{KR} \times T_f \times C_{KR} + r^A_{KR} \times G^C_{KR} \tag{A.6.17}$$

- POM production:
$$P^C_{KR} \quad = pom_{KR} \times G^C_{KR} \tag{A.6.18}$$
$$P^N_{KR} \quad = q^C_{N,POM} \times G^N_{KR} \tag{A.6.19}$$
$$P^P_{KR} \quad = q^C_{P,POM} \times G^P_{KR} \tag{A.6.20}$$

- RDOC release:
$$E^C_{KR,RDOC} = refr_{KR} \times C_{KR} \tag{A.6.21}$$
$$E^N_{KR,RDON} = E^C_{KR,RDOC} \times q^C_{N,RDOM} \tag{A.6.22}$$
$$E^P_{KR,RDOP} = E^C_{KR,RDOC} \times q^C_{P,RDOM} \tag{A.6.23}$$

- Removal by higher trophic levels
$$M^C_{KR} = mort_{KR} \times C_{KR} \times C_{KR} \tag{A.6.24}$$
$$M^N_{KR} = M^C_{KR,RDOC} \times Q^C_{N,KR} \tag{A.6.25}$$
$$M^P_{KR} = M^C_{KR,RDOC} \times Q^C_{P,KR} \tag{A.6.26}$$

- The net growth rate equations:
$$\frac{dC_{KR}}{dt} = G^C_{KR} - E^C_{KR,LDOC} - E^C_{KR,SDOC,1} - E^C_{KR,SDOC,2} - E^C_{KR,RDOC}$$
$$- P^C_{KR} - R^C_{KR} - M^C_{KR} \tag{A.6.27}$$
$$\frac{dN_{KR}}{dt} = G^N_{KR} - E^N_{KR,LDON} - E^N_{KR,SDON,1} - E^N_{KR,SDON,2} - E^N_{KR,RDON}$$
$$- P^N_{KR} - R^N_{KR} - M^N_{KR} \tag{A.6.28}$$

$$\frac{dP_{KR}}{dt} = G^P_{KR} - E^P_{KR,LDOC} - E^P_{KR,SDOC,1} - E^P_{KR,SDOC,2} - E^P_{KR,RDOC}$$
$$- P^P_{KR} - R^P_{KR} - M^P_{KR} \tag{A.6.29}$$

### 7. Detrital processes

- Dissolution:

$$DISS^C_{DET} = diss \times C_{DET} \tag{A.7.1}$$
$$DISS^N_{DET} = diss \times prf_N \times N_{DET} \tag{A.7.2}$$
$$DISS^P_{DET} = diss \times prf_P \times P_{DET} \tag{A.7.3}$$

- The net change equations:

$$\frac{dC_{DET}}{dt} = D^C_{DA} + D^C_{CR} + D^C_{MZ} + D^C_{KR} + DISS^C_{HZ} - DISS^C_{DET} \tag{A.7.4}$$

$$\frac{dN_{DET}}{dt} = D^N_{DA} + D^N_{CR} + D^N_{MZ} + D^N_{KR} + DISS^N_{HZ} - DISS^N_{DET} \tag{A.7.5}$$

$$\frac{dP_{DET}}{dt} = D^P_{DA} + D^P_{CR} + D^P_{MZ} + D^P_{KR} + DISS^P_{HZ} - DISS^P_{DET} \tag{A.7.6}$$

where $DISS^C_{HZ} = f_{POM,HZ} \times M^C_{KR}$
$$DISS^N_{HZ} = f_{POM,HZ} \times M^N_{KR}$$
$$DISS^P_{HZ} = f_{POM,HZ} \times M^P_{KR}$$

### 8. DOM processes

- Conversion of SDOM to RDOM:

$$REFR^C_{SDOM} = ex_{REFR,SDOM} \times C_{SDOM} \times \exp\{ 1 - \min(Q^C_{N,SDOM}/q^C_{N,RDOM}, Q^C_{P,SDOM}/q^C_{P,RDOM}) \} \tag{A.8.1}$$
$$REFR^N_{SDOM} = REFR^C_{SDOM} \times q^C_{N,RDOM} \tag{A.8.2}$$

$$REFR^P_{SDOM} = REFR^C_{SDOM} \times q^C_{P,RDOM} \tag{A.8.3}$$

- The net change equations:

$$\frac{dC_{LDOM}}{dt} = E^C_{DA,LDOC} + E^C_{CR,LDOC} + E^C_{MZ,LDOC} + E^C_{KR,LDOC} + M^C_{BAC} - G^C_{BAC,LDOC} \tag{A.8.4}$$

$$\frac{dN_{LDOM}}{dt} = E^N_{DA,LDON} + E^N_{CR,LDON} + E^N_{MZ,LDON} + E^N_{KR,LDON} + M^N_{BAC} - G^N_{BAC,LDON} \tag{A.8.5}$$

$$\frac{dP_{LDOM}}{dt} = E^P_{DA,LDOP} + E^P_{CR,LDOP} + E^P_{MZ,LDOP} + E^P_{KR,LDOP} + M^P_{BAC} - G^P_{BAC,LDOP} \tag{A.8.6}$$

$$\frac{dC_{SDOM}}{dt} = E^C_{DA,SDOC} + E^C_{CR,SDOC} + E^C_{BAC,SDOC} + E^C_{MZ,SDOC,1} + E^C_{MZ,SDOC,2}$$
$$+ E^C_{KR,SDOC,1} + E^C_{KR,SDOC,2} + E^C_{HZ,SDOC} + DISS^C_{DET} - REFR^C_{SDOM} - G^C_{BAC,SDOC} \tag{A.8.7}$$

$$\frac{dN_{SDOM}}{dt} = E^N_{DA,SDON} + E^N_{CR,SDON} + E^N_{BAC,SDON} + E^N_{MZ,SDON,1} + E^N_{MZ,SDON,2}$$
$$+ E^N_{KR,SDON,1} + E^N_{KR,SDON,2} + E^N_{HZ,SDON} + DISS^N_{DET} - REFR^N_{SDOM} - G^N_{BAC,SDON} \tag{A.8.8}$$

$$\frac{dP_{SDOM}}{dt} = E^P_{DA,SDOP} + E^P_{CR,SDOP} + E^P_{BAC,SDOP} + E^P_{MZ,SDOP,1} + E^P_{MZ,SDOP,2}$$

$$+ E^P_{KR,SDOP,1} + E^P_{KR,SDOP,2} + E^P_{HZ,SDOP} + DISS^P_{DET} - REFR^P_{SDOM} - G^P_{BAC,SDOP} \tag{A.8.9}$$

### 9. Dissolved inorganic nutrient processes

- Nitrification:

$$NTRF = r_{ntrf} \times NH4 \tag{A.9.1}$$

- The net change equations:

$$\frac{dNH4}{dt} = FLUX^{NH4}{}_{BAC} + REMI^{N}{}_{MZ} + REMI^{N}{}_{KR} + REMI^{N}{}_{HZ} - G^{NH4}{}_{DA} - G^{NH4}{}_{CR} - NTRF \tag{A.9.2}$$

$$\frac{dNO3}{dt} = FLUX^{NO3}{}_{BAC} - G^{NO3}{}_{DA} - G^{NO3}{}_{CR} + NTRF \tag{A.9.3}$$

$$\frac{dPO4}{dt} = FLUX^{PO4}{}_{BAC} + REMI^{P}{}_{MZ} + REMI^{P}{}_{KR} + REMI^{P}{}_{HZ} - G^{PO4}{}_{DA} - G^{PO4}{}_{CR} \tag{A.9.4}$$

where $REMI^{N}{}_{HZ} = M^{N}{}_{KR} - D^{N}{}_{HZ} - E^{SDON}{}_{HZ}$
$REMI^{P}{}_{HZ} = M^{N}{}_{KR} - D^{P}{}_{HZ} - E^{SDOP}{}_{HZ}$

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

**Figures**

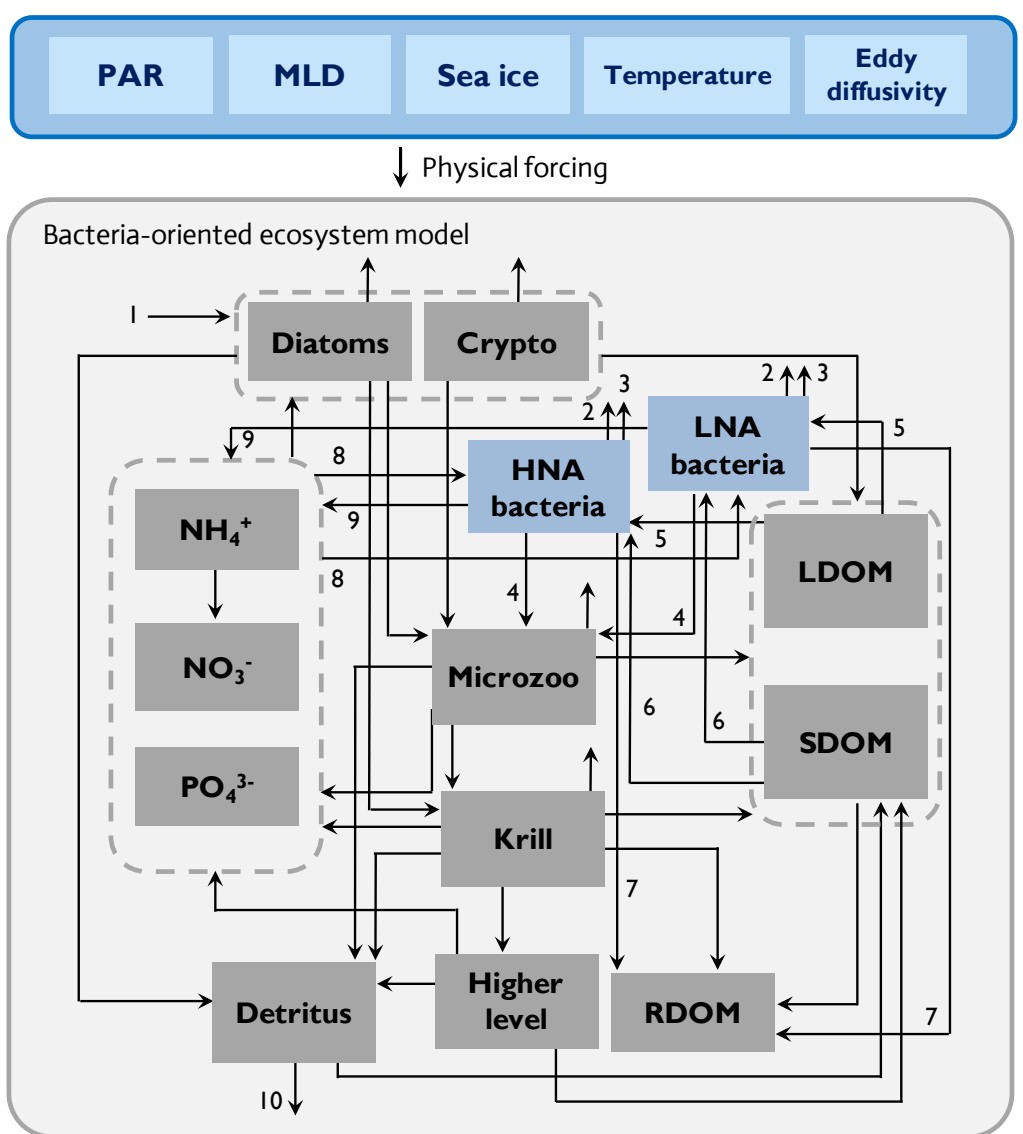

PAR: photosynthetically active radiation, MLD: mixed layer depth,
Crypto: cryptophytes, Microzoo: microzooplankton, LDOM: labile dissolved organic matter,
SDOM: semi-labile dissolved organic matter, RDOM: refractory dissolved organic matter,

1: Primary production, 2: Respiration, 3: Viral mortality, 4: Grazing , 5: LDOM uptake,
6: SDOM uptake, 7: Excretion, 8: Nutrient uptake, 9: Regeneration, 10: Particle export

**Figure 1: Model structure.** The model is forced by five different physical forcings, denoted as a horizontal row across the top of the schematic. As the ecosystem component, heterotrophic bacteria are divided into two groups of differing physiological states, high nucleic acid (HNA) and low nucleic acid (LNA) bacterial compartments. The flows between the prognostic state variables with the name of the numbered flows in the legend only represent for these two bacterial compartments.

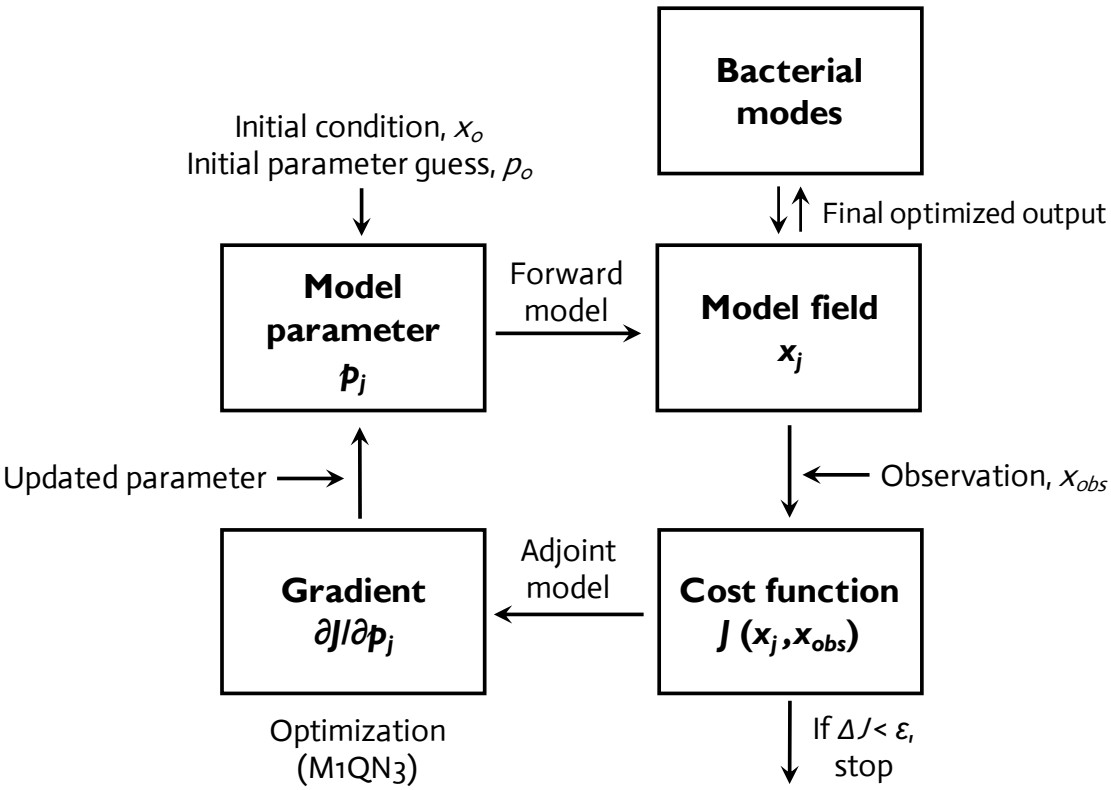

**Figure 2: Data assimilation scheme.** A variational adjoint method is employed for the parameter optimization and data assimilation processes (adapted from Glover et al. 2011). Gradient: the sensitivity of the total cost function with respect to model parameter from optimization. Optimized model output was interpreted as a function of bacterial taxonomic and physiological traits.

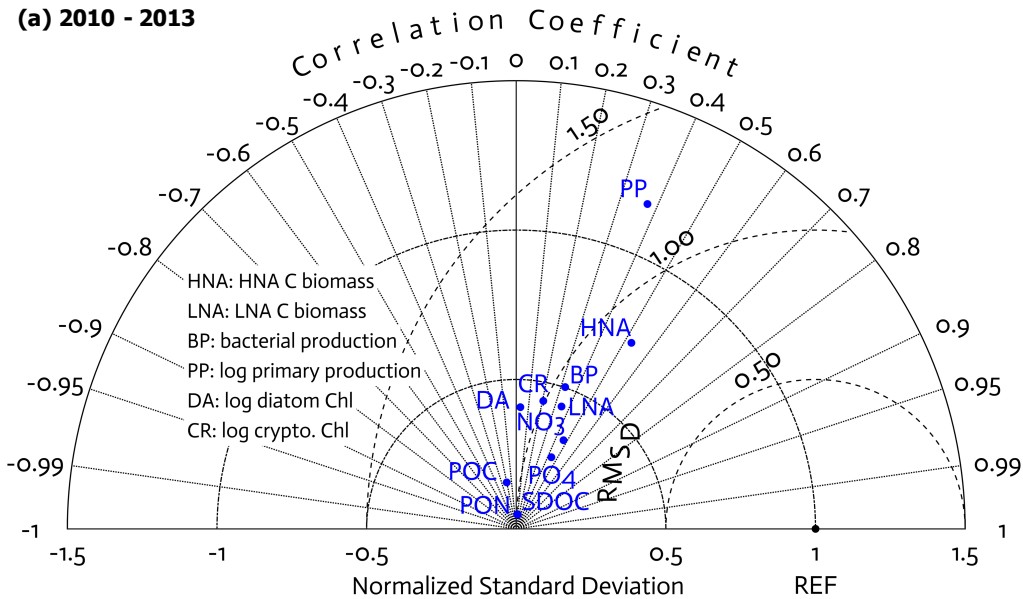

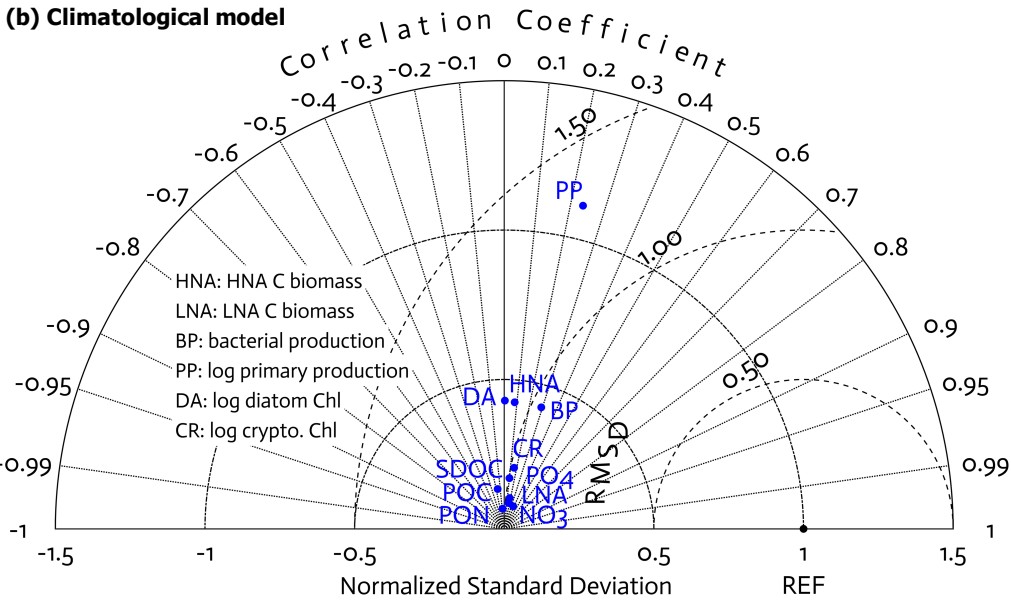

**Figure 3: Model skill assessment.** A Taylor diagram using a polar-coordinate system summarizing the model-observational correspondence for each model stock and flow for individual annual simulations for the four select modelled years together (2010-11 to 2013-14; a) and for the climatological year (b). The angular coordinate denotes the Pearson correlation coefficient (r), the distance from the origin denotes the normalized standard deviation, and the distance from point (1,0), marked as REF on x-axis, describes the centred (bias removed) root-mean-square difference (RMSD) between model results and observations. Note different x-axis scales are used for the normalized standard deviation in each panel.

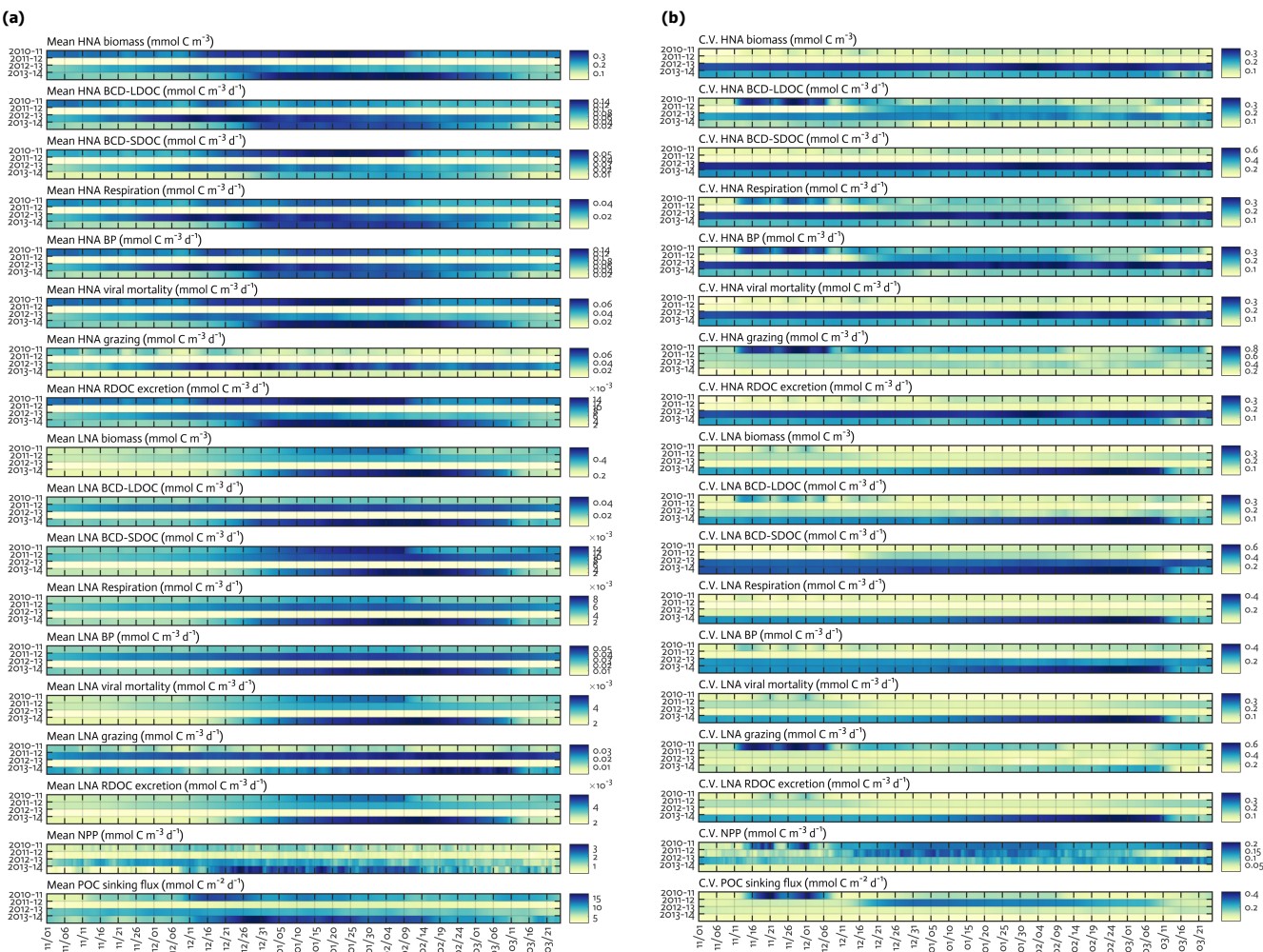

**Figure 4: Seasonal progression of modelled HNA and LNA bacterial carbon stocks and rates and key ecosystem functions across years.** Seasonal patterns of HNA and LNA bacterial carbon stocks and flows, NPP and POC sinking flux at 10-m depth over the growth season (November-March) for each of the 4 simulation years (a), and coefficient of variation (Monte Carlo-derived standard deviation divided by each data point from Figure 4A) from 1,000 Monte Carlo experiments (b).

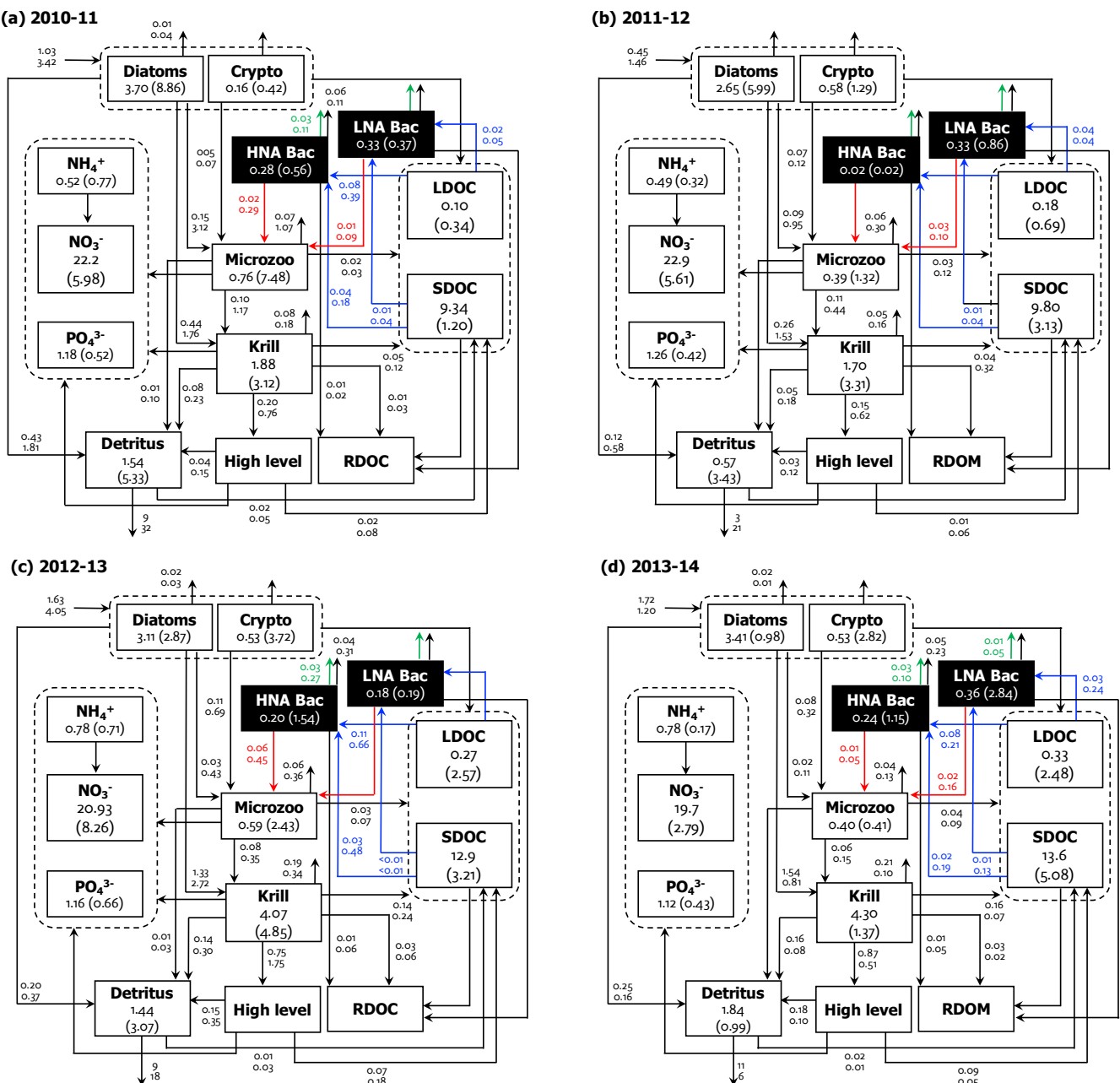

**Figure 5: Annual mean carbon stocks and flows.** Carbon stocks (mmol C m$^{-3}$), flows (mmol C m$^{-3}$ d$^{-1}$), and particle sinking flux (mmol C m$^{-2}$ d$^{-1}$) averaged over the growth season in each year are denoted as the numbers on the first row, while the numbers in the parentheses are the standard deviation propagated from averaging over the growth season and the Monte Carlo experiment-derived uncertainties. Flows do not necessarily balance to zero due to the build-up or loss in a compartment over the growth season. N and P flows, as well as the flows smaller than 0.01 mmol C m$^{-3}$ d$^{-1}$, are omitted.

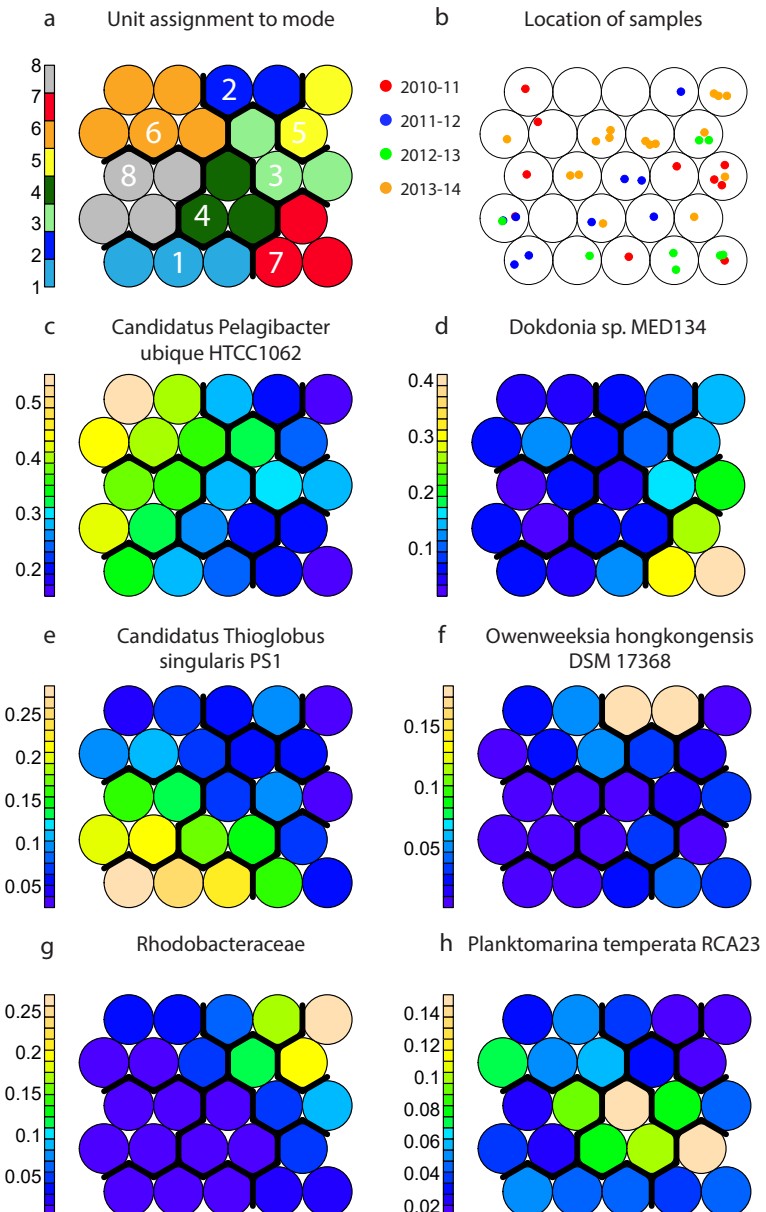

**Figure 6: Properties of the emergent self-organizing map for bacterial community structure shown as taxonomic modes (modified from Bowman et al. 2017).** Map units are colored and numbered according to taxonomic mode membership (a). Location of samples used in this study within the map (b). The map was trained with a larger set of samples, here, only those samples for which BP and flow cytometry data were available (those samples used in this study) are shown. Mode boundaries are the same as in (a). Each sample was placed within the map unit that had the most similar community structure, however, the position of each samples within the map unit is random. Relative abundance of the most abundant taxa in the microbial community structure dataset in each map unit after training (c-h). For example, *P. ubique* HTCC1062 (c) dominated samples associated with Mode 6, while *Ca*. Thioglobus singularis PS1 (e) dominated samples associated with Mode 1. The boundaries across all panels are as in (a).

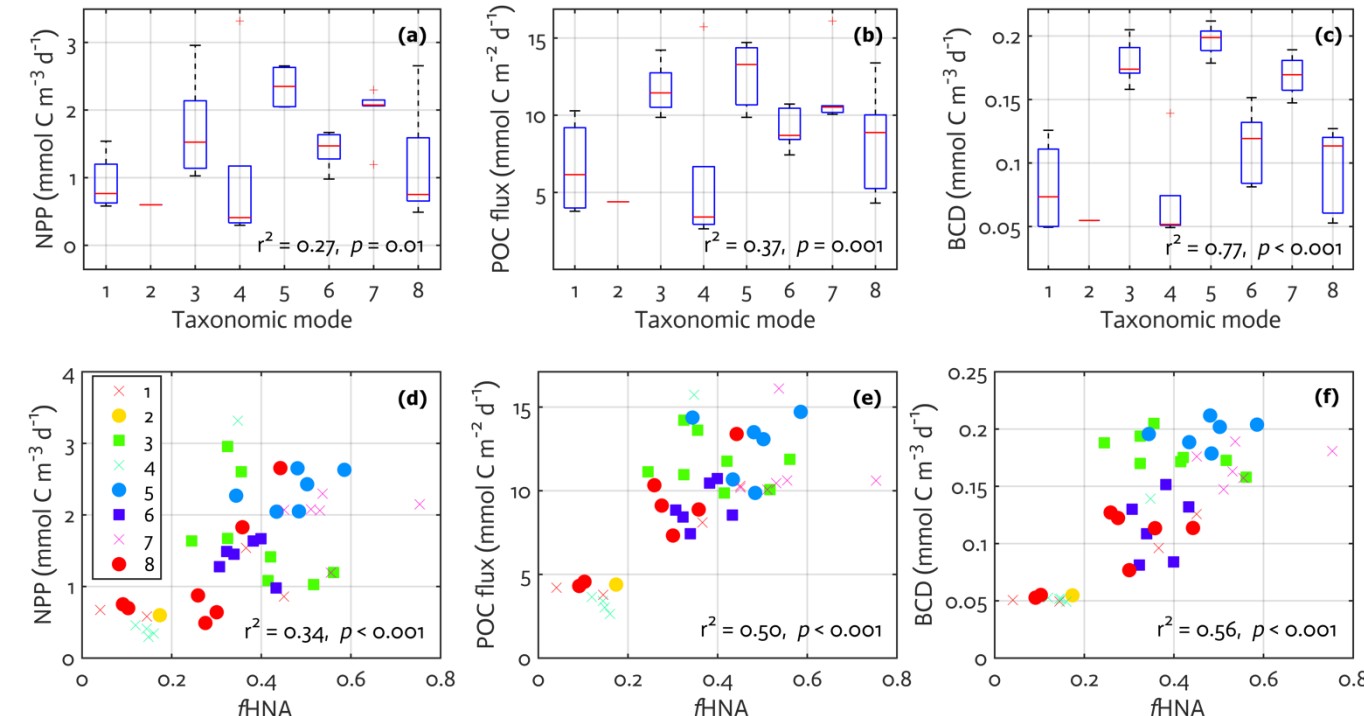

**Figure 7: Bacterial physiological and taxonomic association with ecosystem functions.** The results of linear regression of key modelled ecosystem functions on a categorical predictor of the observed mode (a-c) and on the observed fraction of HNA cells (d-f). Regression statistics: (a) number of observations ($N$) = 43, error degrees of freedom ($df$) = 35; , root mean square error (RMSE) = 0.68 $r^2$ = 0.39, adjusted $r^2$ = 0.27, $F$-statistic value = 3.22, $p$-value = 0.01; (b) $N$ = 43, $df$ = 35, RMSE = 2.88, $r^2$ = 0.48, adjusted $r^2$ = 0.37, $F$-statistic value = 4.55, $p$-value = 0.001; (c) $N$ = 43, $df$ = 35, RMSE = 0.03, $r^2$ = 0.81, adjusted $r^2$ = 0.77, $F$-statistic value = 20.7, $p$-value < 0.001; (d) $N$ = 43, $df$ = 41, RMSE = 0.65, $r^2$ = 0.36, adjusted $r^2$ = 0.34, $F$-statistic value = 22.8, $p$-value < 0.001; (e) $N$ = 43, $df$ = 41, RMSE = 0.13, $r^2$ = 0.51, adjusted $r^2$ = 0.50, $F$-statistic value = 43.0, $p$-value < 0.001; (f) $N$ = 43, $df$ = 41, RMSE = 0.04, $r^2$ = 0.57, adjusted $r^2$ = 0.56, $F$-statistic value = 53.5, $p$-value < 0.001.

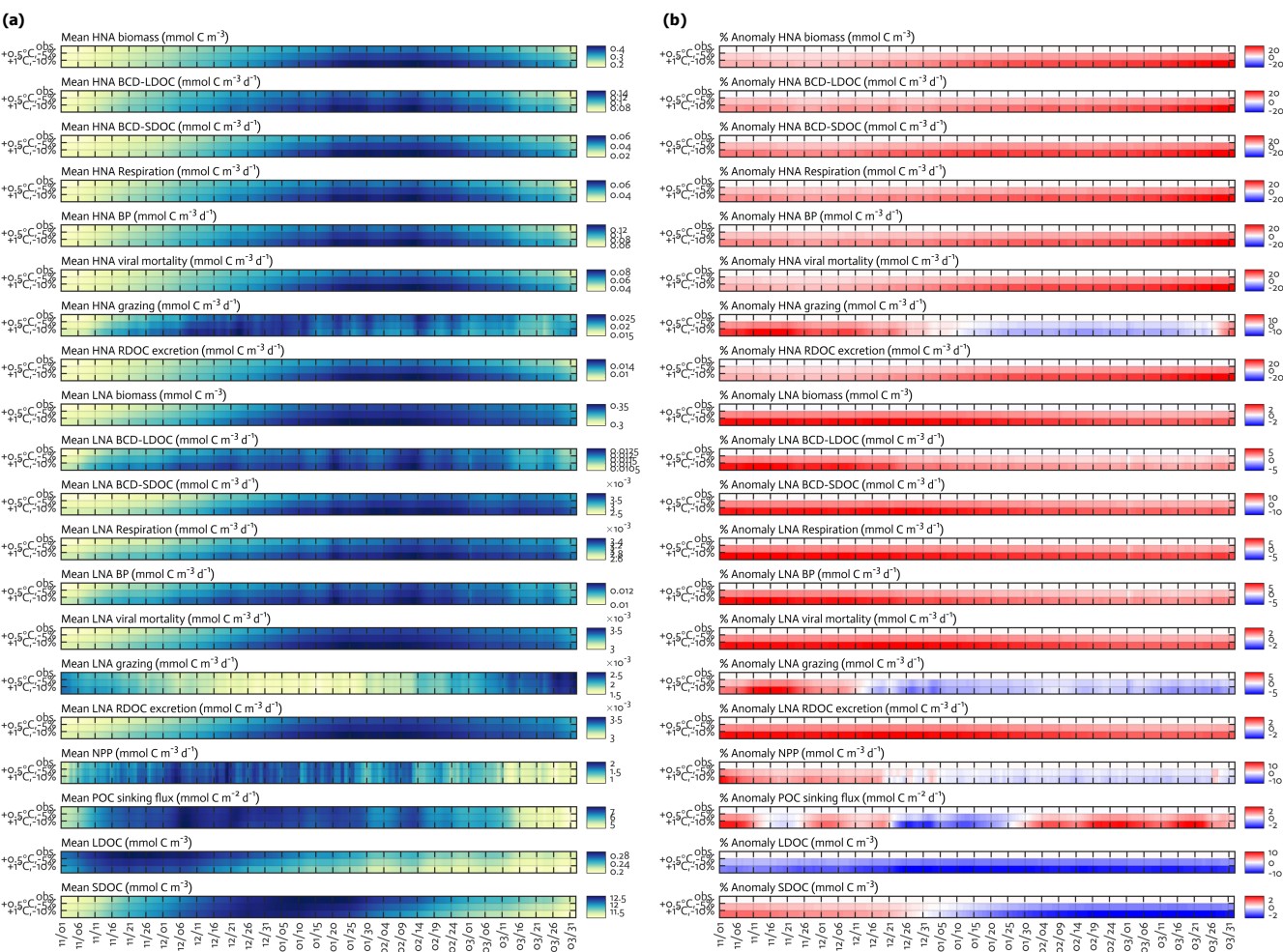

**Figure 8: Climate change experiments.** Seasonal progression of the modelled HNA and LNA bacterial carbon stocks and rates and key ecosystem functions under observed physical forcing and climate change experiments (a) and the percent change of the corresponding variable compared to observed fields in the second and third row of each panel, with the first row of each panel as zero to represent base states (b). For example, percent anomaly of the HNA biomass in (b) = (HNA biomass under +1°C/-10% − HNA biomass under observed forcing) × 100/HNA biomass under observed forcing.

1135

 **Tables**

| Parameter | Definition | HNA | LNA |
|---|---|---|---|
| $k^{DOC,BAC}$ | DOC half-saturation concentration for bacterial uptake, mmol C m$^{-3}$ | 0.5 | 0.2 |
| $\mu_{BAC}$ | Maximum bacterial growth rate, d$^{-1}$ | 2.0 | 1.0 |
| $b_{R,BAC}$ | Parameter control bacterial active respiration versus production, (mmol C m$^{-3}$ d$^{-1}$)$^{-1}$ | 0.08 | 0.2 |
| $remi_{BAC}$ | Bacterial nutrient regeneration rate, d$^{-1}$ | 8.0 | 2.0 |
| $ex_{REFR,BAC}$ | Bacterial RDOC production rate, d$^{-1}$ | 0.04 | 0.01 |
| $f_{S,BAC}$ | Bacterial selection strength on SDOM | 0.1 | 0.7 |
| $r^{B}_{BAC}$ | Bacterial basal respiration rate, d$^{-1}$ | 0.04 | 0.01 |
| $r^{A}_{min,BAC}$ | Bacterial minimum active respiration rate, d$^{-1}$ | 0.08 | 0.04 |
| $r^{A}_{max,BAC}$ | Bacterial maximum active respiration rate, d$^{-1}$ | 0.4 | 0.1 |
| $mort_{BAC}$ | Bacterial mortality rate, d$^{-1}$ | 0.2 | 0.01 |
| $g_{BAC}$ | Bacterial half-saturation concentration in microzooplankton grazing, mmol C m$^{-3}$ | 0.55 | 0.55 |

**Table 1: Initial guess values of bacterial model parameters.** Different values are assigned to the model parameters of the HNA and LNA groups to simulate distinct physiological processes and trophic interactions for each group.

| Data types | $\bar{\bar{a}}$ | CV | σ | 2010-11 model parameter set | | 2011-12 model parameter set | | 2012-13 model parameter set | | 2013-14 model parameter set | | Climatological model parameter set | |
|---|---|---|---|---|---|---|---|---|---|---|---|---|---|
| | | | | $J'_0$ | $J'_f$ | $J'_0$ | $J'_f$ | $J'_0$ | $J'_0$ | $J'_0$ | $J'_f$ | $J'_0$ | $J'_f$ |
| NO$_3$ | 19.70 | 0.04 | 0.76 | - | - | 8.04 | 5.23 | 11.74 | 8.88 | 27.82 | 13.52 | 10.41 | 9.62 |
| PO$_4$ | 1.31 | 0.03 | 0.04 | 9.20 | 7.08 | 86.26 | 21.03 | 41.41 | 6.64 | 2.70 | 7.05 | 45.76 | 10.47 |
| log$_{10}$(Chl$_{DA}$) | 0.16 | 0.08 | 0.09 | 12.94 | 5.69 | 6.55 | 9.49 | 12.19 | 12.66 | 10.57 | 7.76 | 6.57 | 8.52 |
| log$_{10}$(Chl$_{CR}$) | -0.90 | 0.10 | 0.10 | 8.75 | 6.41 | 11.04 | 7.33 | 10.02 | 8.37 | 9.92 | 8.23 | 11.10 | 6.95 |
| log$_{10}$(PP) | 1.32 | 0.21 | 0.21 | 4.50 | 4.71 | 4.51 | 2.69 | 9.81 | 6.26 | 9.86 | 7.61 | 7.19 | 3.83 |
| HNA biomass | 0.21 | 0.08 | 0.02 | 20.39 | 2.08 | 0.15 | 0.20 | 24.86 | 8.49 | 36.34 | 10.28 | 23.78 | 10.87 |
| LNA biomass | 0.33 | 0.08 | 0.02 | 4.26 | 3.06 | 673.73 | 21.05 | 860.14 | 1.99 | 10.65 | 6.15 | 590.29 | 9.27 |
| BP | 0.11 | 0.16 | 0.02 | 3.54 | 3.83 | 16.72 | 0.65 | 24.20 | 13.23 | 5.65 | 5.05 | 12.82 | 3.50 |
| SDOC | 10.52 | 0.20 | 2.13 | 3.88 | 3.96 | 1.38 | 1.42 | - | - | - | - | 2.76 | 2.68 |
| POC | 11.24 | 0.13 | 0.78 | 12.03 | 12.68 | 33.39 | 23.19 | - | - | - | - | 39.23 | 16.26 |
| PON | 2.40 | 0.12 | 0.43 | 48.44 | 48.26 | 42.77 | 26.19 | - | - | - | - | 43.86 | 27.30 |
| Total cost | | | | 127.94 | 97.77 | 884.53 | 118.46 | 994.38 | 66.51 | 113.51 | 65.65 | 793.77 | 109.29 |

**Table 2: Data types, observed means, coefficient of variation, target errors, and costs before and after optimization.** The observed mean ($\bar{\bar{a}}$), coefficient of variation (CV), and target error (σ) of each assimilated data type used for calculating the normalized cost function (unitless; Eq. 5) before ($J'_0$) and after optimization ($J'_f$). Data type unit: mmol m$^{-3}$ for nitrate (NO$_3$), phosphate (PO$_4$); mmol C m$^{-3}$ for diatom chlorophyll (Chl$_{DA}$), cryptophyte chlorophyll (Chl$_{CR}$), HNA and LNA bacterial biomass, SDOC, and POC; mmol N m$^{-3}$ for PON; and mmol C m$^{-3}$ d$^{-1}$ for primary production (PP) and bacterial production (BP). NO$_3$ was not assimilated in 2010-11, while SDOC, POC, and PON were not assimilated in 2012-13 and 2013-14 (denoted as '-' in the table).

| Data types | $J'_c$ | 2010-11 model parameter set | | | 2011-12 model parameter set | | | 2012-13 model parameter set | | | 2013-14 model parameter set | | |
|---|---|---|---|---|---|---|---|---|---|---|---|---|---|
| | | $J'_{x, 2011-12}$ | $J'_{x, 2012-13}$ | $J'_{x, 2013-14}$ | $J'_{x, 2010-11}$ | $J'_{x, 2012-13}$ | $J'_{x, 2013-14}$ | $J'_{x, 2010-11}$ | $J'_{x, 2011-12}$ | $J'_{x, 2013-14}$ | $J'_{x, 2010-11}$ | $J'_{x, 2011-12}$ | $J'_{x, 2012-13}$ |
| $NO_3$ | 9.62 | 4.88 | 10.35 | 30.82 | N/A | 10.61 | 31.96 | N/A | 4.68 | 20.32 | N/A | 6.54 | 10.26 |
| $PO_4$ | 10.47 | 24.36 | 5.42 | 1.46 | 9.15 | 5.47 | 0.88 | 8.33 | 28.54 | 2.70 | 7.56 | 37.74 | 10.70 |
| $log_{10}(Chl_{DA})$ | 8.52 | 8.66 | 13.93 | 8.45 | 7.42 | 13.92 | 8.67 | 5.95 | 7.37 | 7.91 | 5.64 | 7.34 | 12.45 |
| $log_{10}(Chl_{CR})$ | 6.95 | 12.00 | 17.38 | 19.62 | 8.47 | 8.44 | 9.50 | 10.22 | 7.24 | 8.99 | 9.50 | 7.16 | 7.54 |
| $log_{10}(PP)$ | 3.83 | 1.86 | 8.45 | 10.71 | 6.08 | 10.38 | 12.94 | 3.87 | 1.70 | 8.18 | 4.68 | 2.28 | 5.88 |
| HNA biomass | 10.87 | 22.93 | 11.57 | 12.57 | 26.90 | 25.86 | 43.08 | 6.11 | 9.44 | 16.01 | 2.75 | 27.71 | 14.95 |
| LNA biomass | 9.27 | 22.54 | 4.64 | 16.39 | 4.60 | 17.14 | 7.57 | 12.91 | 28.98 | 27.43 | 7.03 | 25.40 | 28.47 |
| BP | 3.50 | 3.39 | 14.14 | 5.48 | 6.76 | 16.32 | 10.02 | 3.80 | 1.66 | 5.86 | 3.02 | 3.21 | 13.69 |
| SDOC | 2.68 | 1.40 | N/A | N/A | 3.90 | N/A | N/A | 3.53 | 1.85 | N/A | 3.47 | 2.46 | N/A |
| POC | 16.26 | 23.70 | N/A | N/A | 12.02 | N/A | N/A | 12.63 | 21.51 | N/A | 14.35 | 20.23 | N/A |
| PON | 27.30 | 26.04 | N/A | N/A | 47.53 | N/A | N/A | 47.92 | 27.48 | N/A | 49.58 | 29.47 | N/A |
| Total cost | 109.29 | 151.77 | 85.86 | 105.51 | 132.85 | 108.15 | 124.62 | 115.27 | 140.47 | 97.39 | 107.57 | 169.54 | 103.93 |
| Portability index | | 0.68 ± 0.08 | | | 0.61 ± 0.12 | | | 0.76 ± 0.11 | | | 0.73 ± 0.17 | | |

⌊160

**Table 3: Cross-validation cost and portability index.** $J'_c$ as the normalized optimized cost from the climatological model (equivalent to $J'_f$ under the climatological model parameter set in Table 1) and $J'_x$ as the normalized cross-validation cost (Eq. 7) where, for example, $J'_{x, 2011-12}$ under 2010-11 model parameter set indicates the normalized cross-validation cost from simulating 2010-11 model parameter set against 2011-12.