# Peer review of "Modelling polar marine ecosystem functions guided by bacterial physiological and taxonomic traits"

_Biogeosciences, 2020_

## Referee Comment (RC1) · Anonymous Referee #1 · 2 Oct 2020

This ms presents a bacteria-oriented ecosystem model, calibrated with a data-assimilation scheme, with two explicit bacteria types, fast growing (HNA) and slow growing (LNA). The authors find that properties of the bacterial community are strong predictors of bacterial carbon (C) demand, primary production (PP), and export (EP). The calibrated model is used to make predictions for a warming ocean.

At first I was quite intrigued by the approach of this study. But after going through the ms, it now appears fraught with too many problems to make it worthwhile. The problems start already with the title. I consider "microbial diversity-informed modelling ..." a gross overstatement of the authors' approach, which is more correctly describes

in the abstract as "bacteria-oriented". Below I will outline why I consider this a failed attempt and how it might be modified into a useful contribution. Because I have the strong impression that essential information about the model and the data-assimilation method is missing, I will not go into much detail, though.

My first major problem was understanding the design of the model. The authors refer to one published work (Luo et al., 2010) regarding the model equations (besides unpublished manuscripts, which may or may not eventually be published), and present only the equations for the two bacteria groups. The model of Luo et al. (2010) is much more complex, totalling 30 state variables, than this one (with 12 states), so this reference does not really help much. Without access to the model equations, any attempt to understand the model code will be futile. In consequence, it also remains unclear what the model currency is. According to Fig. 1 and the description in the text (which is not very clear in this respect, except that the number of states is 12) the model employs a fixed stoichiometry approach but it remains unclear whether the fluxes are based on nitrogen (N) or phosphorus (P). Also according to Fig. 1, it appears that inorganic nutrient have no effect on and are not utilised by phytoplankton, leaving open the question what drives PP in this model. Since only very little information about the model is provided in the text of the ms and the supplement, the model design remains very much opaque. From what little information is presented I can see clearly only that the model is 0-D and employs a rather simplistic physiology (fixed stoichiometry).

The model has 84 parameters, of which 22 (inferred from l. 219 of the ms) are calibrated via data assimilation. What is missing here is a description of how these 22 parameters have been selected in the first place. For example, was the selection based on a preliminary sensitivity analysis or a-priori knowledge or assumptions of the model equations? Also, 22 is, in my experience, a very large number of parameters to constrain given the kinds and amount of data employed here. Thus, it is not very surprising that only a subset of 7–10 of these could be constrained well.

Related to this, the next problem is the description regarding overfitting and portability.

I agree that these are essential concepts all too often neglected in modelling studies and so was happy to see that these are addressed here. Nevertheless, I question the quantification of overfitting (lines 175–179) by comparing the residual error with the (undefined in the ms) "target error" of the observations. Overfitting has very little to do with the noisiness of the observations. It is a consequence of the fact that every model is a simplification of the system it describes, and it is also tightly related to portability. The connection is that overfitting can compromise portability, and this is a good way of assessing overfitting. Overfitting often results from attempting to constrain too many parameters, which is revealed here by several parameters being not well constrained (Tables S2–S6). The different estimates of portability for the different year are another indication of overfitting.

I must admit that the concept of the bacterial modes was new to me, so I was happy to see the clear definition in Section 2.2 (first para). However, I could not figure out the main characteristic of these modes, since only very cursory information is presented in the text and Fig. 6. A table listing the modes and their properties and composition might be very helpful here. As it stands, the concept remains rather confusing. For example, the authors state that (l. 276) each mode is dominated by unique bacterial taxa. But considering Fig. 6, it appears that Candidatus Pelagibacter dominates both modes 6 and 1, although it appears that mode 1 is supposed to be dominated by Candidatus Thioglobus.

The above may be viewed as more technical problems, which could possibly be dealt with by, e.g., a detailed model description with all equations, or a recalibration of the model etc. However, I also see a major conceptual problem regarding the design of the study.

The problem lies in the way the authors use the model to make predictions for a warmer ocean. The main assumption behind the presented approach is that bacterial community composition is strongly correlated ("strong predictor", Abstract) to PP and EP. The functional bacterial community composition is represented in the model and its calibration by assigning higher growth rates to HNA than LNA. Nevertheless, bacteria process the DOM produced during PP, so the behaviour of the bacterial community must be viewed as a response, not a driver, of PP. If bacterial community composition is in fact strongly correlated with PP and EP, that is in itself a very significant finding and I would very much like to see this substantiated. It could become a very useful diagnostic tool. However, here the authors treat the bacteria as the driving force determining PP and EP, which is wrong for several reasons. First, it reverses the cause-effect relation between bacterial activity and PP. Second, even if the cause-effect relation was OK, the data do not cover sufficient interannual temperature variability to allow predicting the response to a warmer ocean.

---

## Referee Comment (RC2) · Anonymous Referee #2 · 6 Nov 2020

General

The manuscript provides an analysis of the bacteria dynamics and ecosystem functioning in the surface ocean layer of the coastal West Antarctic Peninsula, based on in situ measurements and an ecosystem model. The authors develop and validate an existing model and apply it in a 0-dimensional configuration to analyze the bacteria dynamics and the link between the bacterial characteristics and the ecosystem functions. To investigate the impact of climate change on the ecosystem, they use the model to assess changes on bacteria fluxes and ecosystem functions under temperature increase and sea-ice melting conditions. The manuscript makes novel contribution with respect to

ecosystem modelling, in which bacteria are often under-studied despite the fact that they play a crucial role in the ecosystem. The manuscript is well written and organized. However I have the following main concerns that should be addressed before I can recommend its publication: some aspects on the description of the model and its specific implementation for this study should be justified and clarified; the authors carried out a validation effort but, as the validation performed in the study of Kim et al. (in review) is not accessible and the specific implementation is unclear, this effort should be fleshed out to gain additional confidence in the model results; also discussions on model performance, on limitations and weaknesses of the modelling should be included.

Main comments

1/ Description of the ecosystem model and its implementation for this study

One of the objectives of the work is to extend an existing model (Luo et al., 2010) applied by Kim et al. (in review) in the study area, by refining the bacteria compartment of this model. The fact that the manuscript of Kim et al. (In review) is not yet accessible makes it difficult to understand the extension and the specific implementation (period, 1-d/0-d, boundary conditions) performed for the study presented here. The modelling is 0-dimensional. The authors should justify the choice of the 0-dimensional study instead of a 1-dimensional study as performed by Luo et al. (2010) and that is usually done for ecosystem modelling at a water-column measurement station. Is 0-dimensional (and even 1-dimensional) modelling appropriate for this coastal site? Is there a significant influence of lateral transport of organic carbon or nutrients on the ecosystem in this region? If a 0-dimensional is justified here the limitations of this 0-d modelling should be clearly discussed. In the Supplementary Material, the authors specified the boundary conditions of the model during the growth season for the nutrient and dissolved organic matter: "The boundary conditions of nitrate, phosphate, SDOC, SDON, and SDOP are set to 30.9 mmol m-3, 2.4 mmol m-3, 6.5 mmol m-3, 0.6 mmol m-3, and 0.03 mmol m-3, respectively". I find unclear the description of the boundary conditions for

this 0-dimensional modelling. What are the boundary conditions for phytoplankton, zooplankton and particulate organic carbon? Are the given conditions at the base of the 10m depth layer? A constant concentration of variables over time at 10m depth does not seem appropriate for representing a seasonal cycle of the ecosystem. The vertical fluxes of the different model variables at the base of the modelled layer should be better specified in the case when the MLD is greater than 10m or at least references to a similar 0-dimensional study describing this should be included (Luo et al. (2010) study is a 1-dimensional modelling study). The authors should clarify and justify the forcing and boundary conditions of the modelled surface layer and specify the depths of the euphotic and mixed layer here.

2/ Data assimilation

The authors show that data assimilation and parameter optimization can reduce model/observation errors, especially for bacterial stocks and flows. However, the simultaneous assimilation of climatological data and data corresponding to the given year raises questions, notably given the strong link between nutrients and phytoplankton time evolutions (Kim et al 2016) and the possibility of a time lag in phytoplankton growth from one year to another. The authors should justify the choice to assimilate climatological data for chlorophyll and microzooplankton instead of not assimilating these data if they were not measured in the simulated year as is done for nitrate and POC? Does this choice lead to some inconsistencies?

3/ Validation of the model results

The authors present a comparison of the model results with the available in situ data. First, the description and discussion of these comparisons should be a little more substantial. For instance the error on primary production appears significant in some years (e.g. 2012-2013) and data assimilation does not seem to bring an improvement in modelled primary production for all periods and years (e.g. January/February 2012) (Figures S1-S5). Also a negative correlation is obtained in some years for phosphate. The

authors should mention and discuss these points. Second, a description with a figure of the comparison of the modelled and observed climatological or 4-year seasonal cycles of nutrients, phytoplankton, zooplankton and bacteria (in addition to the error that is presented) in the main text or in the Supplementary Material would increase confidence in the capacity of the model to represent the seasonal cycle of the ecosystem. The authors use their model to explore the impact of an increase in temperature and a decrease in sea ice concentration, predicted as a result of climate change, on the WAP ecosystem, in particular on bacterial fluxes, primary production and POC export flux. A validation of the model's capacity to reproduce the already observed climate trends of the ecosystem as mentioned in the introduction L49-52, over a longer period for some of the model's variables (POC, chlorophyll), fluxes (primary production) and/or indicators (for instance time and magnitude of the maximum concentration of bacteria, phytoplankton, DOM or POM or primary production or annual averages) would strengthen confidence in the model for the study of the impact of future changes. This is perhaps presented in the study by Kim et al (in review) but could be redone with this new version of the model and added in this manuscript, perhaps in the Supplementary Material. Another possibility would be to compare the interannual variability obtained with a modelling without data assimilation and with the climatological model parameter set of the 4 simulated years (2010-2011 to 2013-2014) to the observed interannual variability (by specifying the potential anomaly in temperature and sea-ice concentration for those years).

4/ Interannual variability

The following comment is in line with the previous one. The ecosystem model is applied to 4 consecutive years, 2010-11 to 2013-14. The results show interannual variability in bacterial carbon stocks and fluxes. As the changes of primary production and POC export flux are analyzed under varying temperature and sea-ice concentration conditions, those fluxes could also be presented (in Figure 4) and discussed for the 4 modeled years. The authors do not discuss the link between the interannual variability of bacterial C flux and that of meteorological and physical forcing. The authors should consider adding a short description of the meteorological and physical forcing for these 4 years. A figure of the forcing in the Supplementary Material would also be helpful. The authors could specify the potential anomaly in temperature and sea-ice concentration for these 4 years and consider adding a discussion on the interannual variability in ecosystem functioning and in particular bacteria dynamics in response to the interannual variability of forcing.

5/ Discussion on modelling limitations and results

Section 4 should be flushed out with discussions on weaknesses and limitations of the model such as 0-dimensional modelling, short duration of the simulation to explore impact of climate change, errors in some variables or fluxes, data assimilation. The authors mention a "microzooplankton model-observation misfits" in their model outputs (L 163). A discussion on the potential impact of this discrepancy on major results of the study, for instance on distribution of loss terms (including grazing) of the BCD presented in 3.2 and discussed in 4.1, seems to me to be necessary. The authors refine the bacterial compartment of the model. It would be interesting that the authors specify if they have compared the results of this new version with those of the basic version and if so, if they obtain a significant improvement of the modeling of bacteria concentration and production, primary production and POC stock with this new version. It would be relevant to know the potential positive contributions of this complexification of the ecosystem model to guide future works on ecosystem modelling. The use of the term "POC export flux" for the calculated sinking flux of particulate organic matter at 10m depth does not necessarily seem appropriate to me. The term POC export generally refers to POC export under the euphotic layer or the mixed layer. What are the depths of the mixed layer and the euphotic layer in this region? This term could be replaced by a more appropriate term at least in the introduction, discussion and abstract sections.

Minor comments / technical corrections

L55-58 : Could you be more specific and indicate the growth of what, the depth of the Palmer Station, the period of comparisons with observations? L118-119 : Could you specify the depth of the site modelling? L 299-301 : The authors should justify the choice of perturbations applied on temperature and sea-ice concentration by citing previous studies on climate change in the study area. L 406 : Replace "phytoplankton account" by "POM production by phytoplankton accounts" ? L 415 : "experiments" seems more appropriate than "scenarios" considering the duration of the simulation. L 418, 431 : Warming temperature/ temperature warming : Remove temperature or replace warming by increasing. Figure 4 : black titles written over blue colours are difficult to read in Figure 4b, the values in the colour bar overlap. The comparison of HNA and LNA bacteria biomass and fluxes would be easier with an identical range in the colour bar of both panels. Figure 5 : The reading of this figure could be simplified by a colour code for the different compartments. Figure 8b : via instead of vi on the fourth panel.

In Supplements: Figure S1-S5: Some labels of y-axis on figures S1 S5 S6 are cut and grey lines are visible. How are the model outputs and observations normalized? Are they both divided the observed means? Please indicate units of the model/observations errors. L 117: "Kim et al. in prep" : Is it the same article as Kim et al. in review? L 209 : Do you mean June 1, 2012 instead of June 1, 2011? Tables S2-S6 : Please indicate units of the parameters.

---

## Author Comment (AC1) · 31 Mar 2021

Thank you very much for your constructive comments and suggestions on our manuscript. Please note four major elements as our effort to fully address your and other reviewer's concerns together, which served as the basis for updated results in the revised manuscript as well as our response to each comment in this file. After these major changes summarized below, most of the results and main conclusions remained similar compared to the previous version: 1) significant associations of the observed bacterial mode with the modelled NPP, POC flux, and BCD; 2) significant associations of the observed fHNA with the modelled NPP, POC flux, and BCD 3) larger

increases of HNA stocks and functions under climate change conditions than those of LNA cells; and 4) larger cell-specific BP and SDOC uptake rates of HNA cells than those of LNA cells. This suggests the robustness of our model study.

In response to the reviews, we made a number of substantial revisions to the modeling study and manuscript:

1) Modification of modeling framework: We re-built, re-optimized, and re-analyzed the model by completely changing the previous version's 0-D (fixed surface layer) formulation to a 1-D (vertical profile) framework.

2) Additional data assimilation: We added diatom and cryptophyte Chl observations for 2010-2011, 2012-2013, and 2013-2014 in that data assimilation; this new data became available for use during the period of revision.

3) Model equations and GMD manuscript: We included a complete set of model equations (line 92-104, Appendix A) and other details about the model set-up (Text S1-4), as well as attached our Geoscientific Model Development (GMD) manuscript on the original WAP model that served as the basis for our study's bacteria-oriented model (Kim, H. H., Luo, Y.-W., Ducklow, H. W., Schofield, O. M., Steinberg, D. K., and Doney, S. C.: WAP-1D-VAR v1.0: Development and Evaluation of a One-Dimensional Variational Data Assimilation Model for the Marine Ecosystem Along the West Antarctic Peninsula, Geosci. Model Dev. Discuss. [preprint], https://doi.org/10.5194/gmd-2020-375, in review, 2021).

4) Climate change simulations: We updated error estimates in the climate change experiments (Results 3.4) after fixing an error in the Monte Carlo simulation code. Temperature and sea-ice perturbations were also replaced by $+0.5°C$ and $+1.0°C$ of warming and 5% and 10% of melting, from $+1.0°C$ and $+2.0°C$ of warming and 10% and 20% of melting in the previous version, in order to reflect better the trends and changes relevant to the WAP.

5) Others: We 1) added the summary of the climatological model optimization in Table 2 (missing in the previous version), 2) combined the 4-modelled years together (Figure 3a; each year presented in the previous version) and included the Taylor diagram of the climatological model (Figure 3b; missing in the previous version) for model skill assessment, 3) removed discussion on microzooplankton model fits from Table 1 for consistency (presented in Table 1 in the previous version but never discussed), and 4) removed the discussion on the fate of BCD as it did not add new information to the study.

In particular, modification of the modeling framework to 1-D vertical profile and additional data assimilation were both labor-intensive and time-consuming, which caused a long delay in providing our Final Author's comments. Thank you again for your patience and for willingness to re-review the revised manuscript in advance. Below are our responses to each of your specific comments that are highlighted throughout the revised manuscript file.

This ms presents a bacteria-oriented ecosystem model, calibrated with a data assimilation scheme, with two explicit bacteria types, fast growing (HNA) and slow growing (LNA). The authors find that properties of the bacterial community are strong predictors of bacterial carbon (C) demand, primary production (PP), and export (EP). The calibrated model is used to make predictions for a warming ocean. At first I was quite intrigued by the approach of this study. But after going through the ms, it now appears fraught with too many problems to make it worthwhile. The problems start already with the title. I consider "microbial diversity-informed modelling..." a gross overstatement of the authors' approach, which is more correctly describes in the abstract as "bacteria-oriented". Below I will outline why I consider this a failed attempt and how it might be modified into a useful contribution. Because I have the strong impression that essential information about the model and the data-assimilation method is missing, I will not go into much detail, though.

Answer: We hope that the GMD manuscript and added details about the model in the
revised version clear up these concerns. We agree with your assessment on the title of the study. With only part of the microbial diversity data directly informing the model (bacterial physiology data) we modified the title to "Modelling polar marine ecosystem functions guided by bacterial physiological and taxonomic traits" in the revised version.

My first major problem was understanding the design of the model. The authors refer to one published work (Luo et al., 2010) regarding the model equations (besides unpublished manuscripts, which may or may not eventually be published), and present only the equations for the two bacteria groups. The model of Luo et al. (2010) is much more complex, totalling 30 state variables, than this one (with 12 states), so this reference does not really help much. Without access to the model equations, any attempt to understand the model code will be futile. In consequence, it also remains unclear what the model currency is. According to Fig. 1 and the description in the text (which is not very clear in this respect, except that the number of states is 12) the model employs a fixed stoichiometry approach but it remains unclear whether the fluxes are based on nitrogen (N) or phosphorus (P). Also according to Fig. 1, it appears that inorganic nutrient have no effect on and are not utilised by phytoplankton, leaving open the question what drives PP in this model. Since only very little information about the model is provided in the text of the ms and the supplement, the model design remains very much opaque. From what little information is presented I can see clearly only that the model is 0-D and employs a rather simplistic physiology (fixed stoichiometry).

Answer: It is correct that the model was applied as a 0-D framework in the previous version (i.e., a 0-D box model of the surface layer at 10 m), but as mentioned above, the revised version is now based on the new model results from a 1-D vertical profile framework (line 126-129). The model tracks and simulates C, N, and P stocks as concentrations in different inorganic and organic pools (e.g., C, N, P for all living model groups, N for nitrate and ammonium, and P for phosphate). The model has flexible stoichiometry, in which phytoplankton store more C under high light and nutrient-depleted conditions and more N and P under low light conditions. If the N or P cellular quota is lower

than the predefined reference (Redfield) ratio, plankton excrete DOM to adjust their stoichiometry close to the reference ratio. To make these points clear, we 1) revised Figure 1 to show N and P uptake by phytoplankton and 2) added a section demonstrating the model's variable stoichiometry (Text S1, line 224-230). The model PP is driven by photosynthetic active radiation and nutrient uptake by phytoplankton, but given the abundance of NO3, PO4, and SiO3 (Kim et al., 2016, doi: 10.1002/2015JG003311) and iron (Annett et al., 2017, doi: 10.1016/j.marchem.2017.06.004) at the study site, it is the light level that primarily limits PP. To make this pint clear, we added the model coupled ordinary differential equations (line 92-103, Appendix A) and other details (Text S1-4).

The model has 84 parameters, of which 22 (inferred from l. 219 of the ms) are calibrated via data assimilation. What is missing here is a description of how these 22 parameters have been selected in the first place. For example, was the selection based on a preliminary sensitivity analysis or a-priori knowledge or assumptions of the model equations? Also, 22 is, in my experience, a very large number of parameters to constrain given the kinds and amount of data employed here. Thus, it is not very surprising that only a subset of 7–10 of these could be constrained well.

Answer: We revised the Material and Methods 2.3 to demonstrate how we chose an initial subset of model parameters submitted to optimization (line 156-165). With regard to your concern on too many optimized parameters, the total number of optimized parameters changed from 12-15 in the previous version to 3-6 in the revised version, while the total number of constrained (optimized with low uncertainties) parameters changed from 7-10 in the previous version to 5-7 in the revised version, therefore, the revised version has a larger fraction of well-constrained parameters.

Related to this, the next problem is the description regarding overfitting and portability. I agree that these are essential concepts all too often neglected in modelling studies and so was happy to see that these are addressed here. Nevertheless, I question the quantification of overfitting (lines 175–179) by comparing the residual error with the
(undefined in the ms) "target error" of the observations. Overfitting has very little to do with the noisiness of the observations. It is a consequence of the fact that every model is a simplification of the system it describes, and it is also tightly related to portability. The connection is that overfitting can compromise portability, and this is a good way of assessing overfitting. Overfitting often results from attempting to constrain too many parameters, which is revealed here by several parameters being not well constrained (Tables S2–S6). The different estimates of portability for the different year are another indication of overfitting.

Answer: This is an excellent point. The portability analysis showed that the optimized model parameter set for 2012-13 was the most portable while the parameter set for 2011-12 was least portable (Table 2), in which the most (n = 7 out of total 11) and the least numbers (n = 5 out of total 11) of parameters were constrained (i.e., optimized with low uncertainties), respectively (Tables S3-4). The other two years exhibited intermediate levels of portability, with similar portability index values characterized by the same number of constrained parameters (n = 6 out of total 10 for 2010-11 and n = 6 out of total 12 for 2013-14; Tables S2, S5). In other words, it was the number of well-constrained parameters that mattered most in driving high model portability, suggesting the connection between overfitting and portability of optimized models, as you suggested. It is commonly thought that more tuned parameters indicate a higher possibility of overfitting and less portability. However, our analysis showed that if the tuned parameters were well-constrained by observations, they would not compromise portability. We added these points in Discussion 4.1 (line 370-376). However, it is our understanding that a reduced chi-square estimate of model fit is a reasonable metric for assessing overfitting of data (Glover et al., 2011, doi: 10.1017/CBO9780511975721). Target errors in our study reflect both the observational errors and seasonal and interannual variations of the observations (which has been intensively defined and discussed in Section 2.5), and larger target errors compared to model-observation misfits, or the noisiness of the observations, would be an indication of overfitting.

I must admit that the concept of the bacterial modes was new to me, so I was happy to see the clear definition in Section 2.2 (first para). However, I could not figure out the main characteristic of these modes, since only very cursory information is presented in the text and Fig. 6. A table listing the modes and their properties and composition might be very helpful here. As it stands, the concept remains rather confusing. For example, the authors state that (l. 276) each mode is dominated by unique bacterial taxa. But considering Fig. 6, it appears that Candidatus Pelagibacter dominates both modes 6 and 1, although it appears that mode 1 is supposed to be dominated by Candidatus Thioglobus.

Answer: Thank you for this comment. The modes are entirely taxonomic, although they can be related statistically to different physiological or ecophysiological parameters. A complete discussion of the modes is given in Bowman et al. (2017; Bowman, J., Amaral-Zettler, L., J Rich, J. et al. Bacterial community segmentation facilitates the prediction of ecosystem function along the coast of the western Antarctic Peninsula. ISME J 11, 1460–1471, doi: 10.1038/ismej.2016.204). Because of this we have opted not to give a more in-depth description here, but have modified the caption for Fig. 6 to try and make this concept more clear. In direct response to your question Mode 6 is dominated by P. ubique (comprising over 50 % of the community for some map units in Mode 6) while Mode 1 is dominated by T. singularis (also reaching over 50 % of the community).

The above may be viewed as more technical problems, which could possibly be dealt with by, e.g., a detailed model description with all equations, or a recalibration of the model, etc. However, I also see a major conceptual problem regarding the design of the study. The problem lies in the way the authors use the model to make predictions for a warmer ocean. The main assumption behind the presented approach is that bacterial community composition is strongly correlated ("strong predictor", Abstract) to PP and EP. The functional bacterial community composition is represented in the model and its calibration by assigning higher growth rates to HNA than LNA. Nevertheless, bacteria

process the DOM produced during PP, so the behaviour of the bacterial community must be viewed as a response, not a driver, of PP. If bacterial community composition is in fact strongly correlated with PP and EP, that is in itself a very significant finding and I would very much like to see this substantiated. It could become a very useful diagnostic tool. However, here the authors treat the bacteria as the driving force determining PP and EP, which is wrong for several reasons. First, it reverses the cause-effect relation between bacterial activity and PP. Second, even if the cause-effect relation was OK, the data do not cover sufficient interannual temperature variability to allow predicting the response to a warmer ocean.

Answer: Thank you for this insightful comment. It is well understood that there can be significant feedback between bacteria and PP, in particular in systems where large PP rates are supported by fast/efficient recycling of nutrients by bacterial remineralization of organic matter to inorganic nutrients. By contrast, the fact that macro- and micronutrients are abundant at the study site (Kim et al., 2016, doi: 10.1002/2015JG003311; Annett et al., 2017, doi: 10.1016/j.marchem.2017.06.004) makes the limitation of light, not of nutrients (Eq. A.2.4-5, B.2.4-5) determine PP rates (Eq. A.2.7, B.2.7). PP rates are also proportional to phytoplankton biomass that is grazed by microzooplankton which follows preferential selection on phytoplankton (i.e., gDA, gCR) versus bacterial food sources (i.e., gHNA, gLNA) as well as bacterial biomass (Eq. A.2.33, A.3.33). This step-wise, indirect connection makes the bacterial-microzooplankton grazing influence on PP rather remote compared to nutrient recycling by bacteria. The same applies to EP. Thus, we agree with your argument that bacteria should be regarded as a responder, rather than a driver, of PP and EP, and therefore changed the wording from "predict/predictor" to "indicate/indicator", "respond", "reflect", or "associated" in the revised version. Regarding your second comment, our climate change experiments identified functional relationships between modeled processes and temperature. Although the real response may not follow the same functions when temperature increases beyond the existing observed range, this is the best we can do for future projections as in other modeling studies. In our experiments,

the "climate change" condition is equivalent to simultaneous warming and melting, which have a more solid physical basis as well as more profound impacts on bacterial processes than the warming alone condition does. Nevertheless, we reduced the range of warming to +0.5°C and +1.0°C, which are relatively minor increases that are not expected to cause very different trends from the existing observed patterns.

Please also note the supplement to this comment:
https://bg.copernicus.org/preprints/bg-2020-302/bg-2020-302-AC1-supplement.pdf

---

## Author Comment (AC2) · 31 Mar 2021

Thank you very much for your constructive comments and suggestions on our manuscript. Please note four major elements as our effort to fully address your and other reviewer's concerns together, which served as the basis for updated results in the revised manuscript as well as our response to each comment in this file. After these major changes summarized below, most of the results and main conclusions remained similar compared to the previous version: 1) significant associations of the observed bacterial mode with the modelled NPP, POC flux, and BCD; 2) significant associations of the observed fHNA with the modelled NPP, POC flux, and BCD 3) larger

increases of HNA stocks and functions under climate change conditions than those of LNA cells; and 4) larger cell-specific BP and SDOC uptake rates of HNA cells than those of LNA cells. This suggests the robustness of our model study.

In response to the reviews, we made a number of substantial revisions to the modeling study and manuscript:

1) Modification of modeling framework: We re-built, re-optimized, and re-analyzed the model by completely changing the previous version's 0-D (fixed surface layer) formulation to a 1-D (vertical profile) framework.

2) Additional data assimilation: We added diatom and cryptophyte Chl observations for 2010-2011, 2012-2013, and 2013-2014 in that data assimilation; this new data became available for use during the period of revision.

3) Model equations and GMD manuscript: We included a complete set of model equations (line 92-104, Appendix A) and other details about the model set-up (Text S1-4), as well as attached our Geoscientific Model Development (GMD) manuscript on the original WAP model that served as the basis for our study's bacteria-oriented model (Kim, H. H., Luo, Y.-W., Ducklow, H. W., Schofield, O. M., Steinberg, D. K., and Doney, S. C.: WAP-1D-VAR v1.0: Development and Evaluation of a One-Dimensional Variational Data Assimilation Model for the Marine Ecosystem Along the West Antarctic Peninsula, Geosci. Model Dev. Discuss. [preprint], https://doi.org/10.5194/gmd-2020-375, in review, 2021).

4) Climate change simulations: We updated error estimates in the climate change experiments (Results 3.4) after fixing an error in the Monte Carlo simulation code. Temperature and sea-ice perturbations were also replaced by +0.5°C and +1.0°C of warming and 5% and 10% of melting, from +1.0°C and +2.0°C of warming and 10% and 20% of melting in the previous version, in order to reflect better the trends and changes relevant to the WAP.

5) Others: We 1) added the summary of the climatological model optimization in Table 2 (missing in the previous version), 2) combined the 4-modelled years together (Figure 3a; each year presented in the previous version) and included the Taylor diagram of the climatological model (Figure 3b; missing in the previous version) for model skill assessment, 3) removed discussion on microzooplankton model fits from Table 1 for consistency (presented in Table 1 in the previous version but never discussed), and 4) removed the discussion on the fate of BCD as it did not add new information to the study.

In particular, modification of the modeling framework to 1-D vertical profile and additional data assimilation were both labor-intensive and time-consuming, which caused a long delay in providing our Final Author's comments. Thank you again for your patience and for your willingness to re-review the revised manuscript in advance. Below are our responses to each of your specific comments that are highlighted throughout the revised manuscript file.

The manuscript provides an analysis of the bacteria dynamics and ecosystem functioning in the surface ocean layer of the coastal West Antarctic Peninsula, based on in situ measurements and an ecosystem model. The authors develop and validate an existing model and apply it in a 0-dimensional configuration to analyze the bacteria dynamics and the link between the bacterial characteristics and the ecosystem functions. To investigate the impact of climate change on the ecosystem, they use the model to assess changes on bacteria fluxes and ecosystem functions under temperature increase and sea-ice melting conditions. The manuscript makes novel contribution with respect to ecosystem modelling, in which bacteria are often under-studied despite the fact that they play a crucial role in the ecosystem. The manuscript is well written and organized. However I have the following main concerns that should be addressed before I can recommend its publication: some aspects on the description of the model and its specific implementation for this study should be justified and clarified; the authors carried out a validation effort but, as the validation performed in the study of Kim et

al. (in review) is not accessible and the specific implementation is unclear, this effort should be fleshed out to gain additional confidence in the model results; also discussions on model performance, on limitations and weaknesses of the modelling should be included.

Answer: We hope that the GMD manuscript and added details about the model in the revised version clear up these concerns. For other significant issues that you listed, please see our response below specific to each of your comments.

1/ Description of the ecosystem model and its implementation for this study One of the objectives of the work is to extend an existing model (Luo et al., 2010) applied by Kim et al. (in review) in the study area, by refining the bacteria compartment of this model. The fact that the manuscript of Kim et al. (In review) is not yet accessible makes it difficult to understand the extension and the specific implementation (period, 1-d/0-d, boundary conditions) performed for the study presented here. The modelling is 0-dimensional. The authors should justify the choice of the 0-dimensional study instead of a 1-dimensional study as performed by Luo et al. (2010) and that is usually done for ecosystem modelling at a water column measurement station. Is 0-dimensional (and even 1-dimensional) modelling appropriate for this coastal site? Is there a significant influence of lateral transport of organic carbon or nutrients on the ecosystem in this region? If a 0-dimensional is justified here the limitations of this 0-d modelling should be clearly discussed.

Answer: The model was applied as a 0-D framework in the previous version (i.e., a 0-D box model of the surface layer at 10 m), but as mentioned above, the revised version is now based on the new model results from a 1-D profile simulation (line 126-129). We also added the justification for the 1-D modeling of the WAP study site (line 129-134).

In the Supplementary Material, the authors specified the boundary conditions of the model during the growth season for the nutrient and dissolved organic matter: "The boundary conditions of nitrate, phosphate, SDOC, SDON, and SDOP are set to 30.9

mmol m-3, 2.4 mmol m-3, 6.5 mmol m-3, 0.6 mmol m-3, and 0.03 mmol m-3, respectively". I find unclear the description of the boundary conditions for this 0-dimensional modelling. What are the boundary conditions for phytoplankton, zooplankton and particulate organic carbon? Are the given conditions at the base of the 10m depth layer? A constant concentration of variables over time at 10m depth does not seem appropriate for representing a seasonal cycle of the ecosystem. The vertical fluxes of the different model variables at the base of the modelled layer should be better specified in the case when the MLD is greater than 10m or at least references to a similar 0-dimensional study describing this should be included (Luo et al. (2010) study is a 1-dimensional modelling study). The authors should clarify and justify the forcing and boundary conditions of the modelled surface layer and specify the depths of the euphotic and mixed layer here.

Answer: We appreciate your concern about several issues regarding the 0-D modeling of the system in the previous version. As mentioned above, the revised version is now based on the new model results from 1-D modeling, whose framework is directly comparable to the 1-D models in Luo et al. (2010; Luo, Y.W., Friedrichs, M.A., Doney, S.C., Church, M.J. and Ducklow, H.W., 2010. Oceanic heterotrophic bacterial nutrition by semilabile DOM as revealed by data assimilative modeling. Aquatic Microbial Ecology, 60(3), pp.273-287. doi: 10.3354/ame01427) and in the attached GMD manuscript (Kim et al., 2021). In these two studies climatological observations of the deep water values were used for bottom boundary conditions of NO3, PO4, and SDOM, while bottom boundary conditions of other variables (i.e., C, N, and P components of bacteria, phytoplankton, zooplankton, LDOM, and detritus as well as NH4) were set as zero – an approach valid for the vertical domain down to the deep, (near) bottom depth of the study sites. For this particular bacteria-oriented study, however, we think it is best to minimize the number of depth levels that do not have bacterial traits observations, yet to include an adequate number of depth levels required for simulating seasonally-varying MLD and light impacts (mostly < 20 m of MLD and ~ 25 +/- 8 m of euphotic zone depth using hydro-light data for 2010-2011 and 2013-2014 at our study site). Thus, we chose

to model 0, 10, and 20 m, which required non-zero bottom boundary values from 20 m, because of high biological and biogeochemical activities there. There are, however, no available observations of HNA, LNA, LDOM, microzooplankton, and krill biomass, LDOM, detritus, and NH4 at 20 m at the study site. Instead, we 1) estimated the climatological (2010-2013) HNA and LNA biomass at 20 m using the ratio of bacterial production to group-specific biomass observations at 10 and 20 m, 2) extracted the climatological (2002-2011) modelled values of microzooplankton, krill, detritus, LDOM, and NH4 at 20 m from our GMD manuscript (Kim et al., 2021), and 3) used the climatological (2010-2013) observations of diatom- and cryptophyte biomass at 20 m for their bottom boundary conditions. We revised the section detailing these procedures (Text S3, line 345-351).

2/ Data assimilation The authors show that data assimilation and parameter optimization can reduce model/observation errors, especially for bacterial stocks and flows. However, the simultaneous assimilation of climatological data and data corresponding to the given year raises questions, notably given the strong link between nutrients and phytoplankton time evolutions (Kim et al 2016) and the possibility of a time lag in phytoplankton growth from one year to another. The authors should justify the choice to assimilate climatological data for Chl and microzooplankton instead of not assimilating these data if they were not measured in the simulated year as is done for nitrate and POC? Does this choice lead to some inconsistencies?

Answer: In the previous version, the decision of assimilating the climatological Chl data had been made based upon our initial attempt of model optimization without assimilating Chl data type at all. However, model cost functions had failed to reach local minima, suggesting the necessity of the Chl data type to constrain key model parameters sensitive to lowering total cost functions (e.g., $\Theta$). Similarly, the single-year observation of microzooplankton data had been assimilated to better constrain grazing loss terms of phytoplankton and bacteria, which otherwise also led to failed optimization. We had not encountered the same issue when not assimilating nitrate and POM. In the

revised version, thanks to the updated data sets we were able to assimilate diatom-
and cryptophyte-specific Chl for 2010-2011, 2012-2013, and 2013-2014, resolving your
concern about the potential mismatch between the time evolutions of Chl and of nu-
trients (or PP, also) in those years. However, Chl data were still missing in 2011-12,
for which we assimilated the climatological observation due to the above reasons. We
revised the Results 2.4. section to include these points (line 176-182).

3/ Validation of the model results The authors present a comparison of the model re-
sults with the available in situ data. First, the description and discussion of these com-
parisons should be a little more substantial. For instance, the error on primary produc-
tion appears significant in some years (e.g. 2012-2013) and data assimilation does not
seem to bring an improvement in modelled primary production for all periods and years
(e.g. January/February 2012) (Figures S1-S5). Also, a negative correlation is obtained
in some years for phosphate. The authors should mention and discuss these points.
Second, a description with a figure of the comparison of the modelled and observed
climatological or 4-year seasonal cycles of nutrients, phytoplankton, zooplankton, and
bacteria (in addition to the error that is presented) in the main text or in the Supple-
mentary Material would increase confidence in the capacity of the model to represent
the seasonal cycle of the ecosystem.

Answer: We updated the Results 3.1 on model skill assessment based on the new 1-D
model results and mentioned several data types whose correlations were negative and
variability was not well captured (line 255-259). We also included the seasonal cycle
of the ecosystem, as suggested, and included it in the discussion of model fits (Figure
S7).

The authors use their model to explore the impact of an increase in temperature and a
decrease in sea ice concentration, predicted as a result of climate change, on the WAP
ecosystem, in particular on bacterial fluxes, primary production and POC export flux. A
validation of the model's capacity to reproduce the already observed climate trends of
the ecosystem as mentioned in the introduction L49-52, over a longer period for some

of the model's variables (POC, Chl), fluxes (primary production) and/or indicators (for instance time and magnitude of the maximum concentration of bacteria, phytoplankton, DOM or POM or primary production or annual averages) would strengthen confidence in the model for the study of the impact of future changes. This is perhaps presented in the study by Kim et al (in review) but could be redone with this new version of the model and added in this manuscript, perhaps in the Supplementary Material. Another possibility would be to compare the interannual variability obtained with a modelling without data assimilation and with the climatological model parameter set of the 4 simulated years (2010-2011 to 2013-2014) to the observed interannual variability (by specifying the potential anomaly in temperature and sea-ice concentration for those years).

Answer: This is a great suggestion. As suggested, we conducted a modified set of the cross-validation analysis to examine the climatological model's capacity to reproduce the observed climate trends of the WAP ecosystem variables. In this analysis, we compared the observed interannual variability to the modelled interannual variability using the climatological parameter set and each year's forcing (without further data assimilation). The climatological model yielded overestimated BP and HNA biomass in 2011-12 and underestimated PP in 2012-13 and 2013-14, compared to other variables whose interannual variability was captured comparatively well to their observed interannual variability (Table S7). Notably, 2011-12 was characterized by the negative temperature anomaly (-0.13 +/- 0.83°C versus 0.03 +/- 0.84°C for the 4-year climatology) and the positive sea-ice anomaly (24 +/- 38% versus 21 +/- 29% for the 4-year climatology), with significantly lower temperature and higher sea-ice cover than other three years (all $p < 0.05$, two-sample t-test). This coldest year had the lowest values of BP, HNA biomass, and PP observations (Table S7), consistent with increases in the modelled BP, HNA biomass, and PP under the combined warming/melting scenario, adding confidence in using the climatological model for the climate change experiments. We revised the Results 4.4 to discuss these points (line 461-474) and added a supplementary figure (Table S7).

4/ Interannual variability The following comment is in line with the previous one. The ecosystem model is applied to 4 consecutive years, 2010-11 to 2013-14. The results show interannual variability in bacterial carbon stocks and fluxes. As the changes of primary production and POC export flux are analyzed under varying temperature and sea-ice concentration conditions, those fluxes could also be presented (in Figure 4) and discussed for the 4 modeled years. The authors do not discuss the link between the interannual variability of bacterial C flux and that of meteorological and physical forcing. The authors should consider adding a short description of the meteorological and physical forcing for these 4 years. A figure of the forcing in the Supplementary Material would also be helpful. The authors could specify the potential anomaly in temperature and sea-ice concentration for these 4 years and consider adding a discussion on the interannual variability in ecosystem functioning and in particular bacteria dynamics in response to the interannual variability of forcing.

Answer: We added a figure summarizing each year's physical forcing (Figure S1) and discussed the interannual variability of the ecosystem variables in relation to the inter-annual variability of temperature and sea ice (line 474-477), in addition to our response above under section 3/ validation of the model results (line 461-474). We also added the modelled PP and POC sinking flux in Figure 4 and added a supplementary table summarizing the annual maximum and minimum values of the modelled variables in different sets of temperature/sea ice anomalies (Table S7).

5/ Discussion on modelling limitations and results Section 4 should be flushed out with discussions on weaknesses and limitations of the model such as 0-dimensional modelling, short duration of the simulation to explore impact of climate change, errors in some variables or fluxes, data assimilation. The authors mention a "microzooplankton model-observation misfits" in their model outputs (L 163). A discussion on the potential impact of this discrepancy on major results of the study, for instance on the distribution of loss terms (including grazing) of the BCD presented in 3.2 and discussed in 4.1, seems to me to be necessary. The authors refine the bacterial compartment of the

model.

Answer: Thank you for reminding us to discuss the weakness and limitations of the study that were poorly demonstrated in the previous version. As suggested, we added the discussions on the duration of the climate change experiments (line 461-464) and error/mismatch of zooplankton variables in data assimilation aspects (line 401-408). As mentioned, we no longer include the fate of the BCD as it does not add much new information to the findings and discussion points of the study.

It would be interesting that the authors specify if they have compared the results of this new version with those of the basic version and if so, if they obtain a significant improvement of the modeling of bacteria concentration and production, primary production and POC stock with this new version. It would be relevant to know the potential positive contributions of this complexification of the ecosystem model to guide future works on ecosystem modelling.

Answer: As suggested, we added a paragraph to discuss the suggested points and contributions to WAP ecosystem modeling brought by our bacteria-oriented modeling (line 478-490).

The use of the term "POC export flux" for the calculated sinking flux of particulate organic matter at 10m depth does not necessarily seem appropriate to me. The term POC export generally refers to POC export under the euphotic layer or the mixed layer. What are the depths of the mixed layer and the euphotic layer in this region? This term could be replaced by a more appropriate term at least in the introduction, discussion and abstract sections.

Answer: As suggested, POC, C, or particle export flux was replaced by POC, C, or particle "sinking flux" throughout.

L55-58: Could you be more specific and indicate the growth of what, the depth of the Palmer Station, the period of comparisons with observations? Answer: Revised (line

56-69)

L118-119: Could you specify the depth of the site modelling? Answer: Added (line 122)

L 299-301: The authors should justify the choice of perturbations applied on temperature and sea-ice concentration by citing previous studies on climate change in the study area. Answer: Added (line 438-445)

L 406: Replace "phytoplankton account" by "POM production by phytoplankton accounts"? Answer: This sentence was deleted as no longer valid.

L 415: "experiments" seems more appropriate than "scenarios" considering the duration of the simulation. Answer: Replaced by "experiments", "simulations", or "conditions" throughout

L 418, 431: Warming temperature/ temperature warming : Remove temperature or replace warming by increasing. Answer: Fixed

Figure 4: black titles written over blue colours are difficult to read in Figure 4b, the values in the colour bar overlap. The comparison of HNA and LNA bacteria biomass and fluxes would be easier with an identical range in the colour bar of both panels. Answer: We tried, but the figure was illegible due to significant intergroup variability.

Figure 5: The reading of this figure could be simplified by a colour code for the different compartments. Answer: Color-coded the flows mentioned (Figure 5).

Figure 8b: via instead of vi on the fourth panel. Answer: Fixed

Figure S1-S5: Some labels of y-axis on figures S1 S5 S6 are cut and grey lines are visible. How are the model outputs and observations normalized? Are they both divided the observed means? Please indicate units of the model/observations errors. Answer: Fixed the figures and legends (Figures S2-6)

L 117: "Kim et al. in prep" : Is it the same article as Kim et al. in review? Answer:

Yes, it is the same article as the attached GMD manuscript (Kim et al. 2021). Fixed throughout the manuscript.

L 209: Do you mean June 1, 2012 instead of June 1, 2011? Answer:Fixed

Tables S2-S6: Please indicate units of the parameters. Answer: Fixed

Please also note the supplement to this comment:
https://bg.copernicus.org/preprints/bg-2020-302/bg-2020-302-AC2-supplement.pdf

**Supplement:**

[Figure]

[Figure]

**WAP-1D-VAR v1.0: Development and Evaluation of a One-Dimensional Variational Data Assimilation Model for the Marine Ecosystem Along the West Antarctic Peninsula**

Hyewon Heather Kim[1,2], Ya-Wei Luo[3], Hugh W. Ducklow[4], Oscar M. Schofield[5], Deborah K. Steinberg[6], Scott C. Doney[1]

[1]Woods Hole Oceanographic Institution, Woods Hole, MA 02543, United States
[2]University of Virginia, Charlottesville, VA 22904, United States
[3]Xiamen University, Xiamen, Fujian 361102, China
[4]Lamont-Doherty Earth Observatory, Columbia University, Palisades, NY 10964, United States
[5]Rutgers University, New Brunswick, NJ 80901, United States
[6]Virginia Institute of Marine Science, Gloucester Point, VA 23062, United States

*Correspondence to*: Hyewon Heather Kim (hkim@whoi.edu)

**Abstract**

The West Antarctic Peninsula (WAP) is a rapidly warming region, with substantial ecological and biogeochemical responses to climate change and variability for the past decades, revealed by multi-decadal observations from the Palmer Antarctica Long-Term Ecological Research (LTER) program. The wealth of these long-term observations provides an important resource for ecosystem modelling, but there has been a lack of focus on the development of numerical models that simulate time-evolving plankton dynamics over the Austral growth season along the coastal WAP. Here we developed a one-dimensional, data assimilation planktonic ecosystem model (i.e., the WAP-1D-VAR model v1.0) equipped with a variational adjoint and model parameter optimization scheme. We first demonstrate the modified and newly added model schemes to the pre-existing food-web and biogeochemical components of the WAP-1D-VAR model, including diagnostic sea-ice forcing and trophic interactions specific to the WAP region. We then conducted model experiments by assimilating eleven different data types from an example Palmer LTER growth season (October 2002 - March 2003) directly related to corresponding model state variables and intercompartmental flows. The iterative, data assimilation procedure reduced by 80% the misfits between observations and model results, compared to before optimization, via an optimized set of 14 parameters out of total 72 free parameters. The optimized model results captured key WAP ecological features, such as blooms during seasonal sea-ice retreat, the lack of macronutrient limitation, and comparable values of the assimilated and non-assimilated model state variables and flows to other studies, as well as several important ecosystem metrics. One exception was slightly underestimated particle export flux, for which we discuss fully potential underlying reasons. The data assimilation scheme of the WAP-1D-VAR model enabled the available observational data to constrain previously poorly understood processes, including the partitioning of primary production by different phytoplankton groups, the optimal chlorophyll to carbon ratio of the WAP phytoplankton community, and the partitioning of dissolved organic carbon pools with different lability. The WAP-1D-VAR model was successfully employed to glue the snapshots from a range of the available data sets together to explain and understand the observed dynamics along the coastal WAP.

[Figure]

[Figure]

**1 Introduction**

The West Antarctic Peninsula (WAP) has experienced significant atmospheric and surface ocean warming since the 1950s, resulting in decreased winter sea-ice duration, retreat of glaciers, and changes in upper ocean dynamics (Clarke et al., 2009; Cook et al., 2005; Henley et al., 2019; King, 1994; Meredith & King, 2005; Stammerjohn et al., 2008; D. Vaughan et al., 2003; D. G. Vaughan, 2006; Whitehouse et al., 2008). These climate-driven changes propagate through marine food-webs

40 by affecting physiology of individual organisms and the whole communities (Hugh W Ducklow et al., 2007). Long-term observational efforts through the Palmer Antarctica Long-Term Ecological Research program (LTER; since 1991) have demonstrated a range of ecological and biogeochemical responses to changing environments, including phytoplankton (Montes-Hugo et al., 2009; Saba et al., 2014; Schofield et al., 2017), marine heterotrophic bacteria (Bowman & Ducklow, 2015; Hugh W. Ducklow et al., 2012; Kim & Ducklow, 2016; Luria et al., 2014, 2017), nutrient drawdown (Kim et al., 2016),

45 and micro- and macrozooplankton (L. Garzio & Steinberg, 2013; Steinberg et al., 2015; Thibodeau et al., 2019).

The wealth of Palmer LTER time-series observations provides an important resource for ecological modelling, and different types of ecological modelling approaches have been developed to explore the WAP responses to climate change and variability. For example, an inverse modelling study estimated the steady-state dynamics of the WAP food-web by deriving snapshots of intercompartmental flows among different plankton functional types and higher trophic levels (Sailley et al.,

50 2013). However, there has been a less focus on numerical ecosystem models that simulate time-evolving plankton dynamics over the full Austral growth season along the WAP. Numerical ecosystem models provide estimates of key rate processes for which observations have been less frequently or seldom made compared to frequently measured stocks and rates. Despite its strength, constructing an ecosystem model is a challenge due to the lack of *a priori* knowledge on model parameters and incomplete understanding of ecological processes that should be explicitly presented in the model structure (H. W. Ducklow

55 et al., 2008; Murphy et al., 2012). Owing to many observational studies, a more robust, yet still incomplete, data-based picture is emerging of WAP food-web interactions and ecosystem dynamics, which could guide a development of the WAP-specific numerical ecosystem model.

Here we introduce a one-dimensional variational data assimilation model specific to the coastal WAP (i.e., the WAP-1D-VAR model v1.0) that we developed by adapting an existing biogeochemical-planktonic model of different ocean basins

60 (Friedrichs, 2001; Friedrichs et al., 2006, 2007; Luo et al., 2010). The WAP-1D-VAR model is compared against the roughly semi-weekly, bio-physical observations over the Austral growth season near Palmer Station on Anvers Island, Antarctica (64.77°S, 64.05°W). The field data record the seasonal variations in phytoplankton bloom initiation, peak and termination and biogeochemistry modulated by variations in surface light, mixed layer depth, and sea-ice cover. In the present study, we 1) describe the structure and schemes of the WAP-1D-VAR in great detail, 2) evaluate the model performance and robustness

65 using a variety of quantitative metrics, and 3) discuss the model applicability with regard to capturing the key WAP ecological and biogeochemical features using the data from an example growth season.

**2 Model development and implementation**

**2.1 Model state variables**

70

The WAP-1D-VAR model v1.0 (Figure 1) is originally derived and modified from data-assimilative, ocean regional test-bed models of the Arabian Sea, the Equatorial Pacific, and the Hawaii Ocean Time-series Station ALOHA (Friedrichs, 2001; Friedrichs et al., 2006, 2007; Luo et al., 2010). The WAP-1D-VAR model simulates stocks and flows of C, N, and P through 12 different model prognostic state variables. The two size-fractionated phytoplankton compartments, diatoms and cryptophyte, and the two different zooplankton compartments, microzooplankton and krill are separately simulated following

the plankton functional types as in Sailley et al (2013). Functional grazing relationships are defined in which diatoms are
75   consumed by both krill (*Euphausia superba*) and microzooplankton (mostly ciliates and other protozoa), cryptophytes are
consumed by microzooplankton, and microzooplankton are grazed by krill. Other abundant zooplankton taxa in the WAP (e.g.,
salps, pteropods, and copepods; Steinberg et al. 2015) are not explicitly simulated in the WAP-1D-VAR model, in part to limit
the model complexity and in part because of limited data constraints on these groups, especially feeding. Higher trophic levels
are implicitly represented to close the model. The WAP-1D-VAR model allows for the partitioning between labile DOM
80   (LDOM) and semi-labile DOM (SDOM) such that the entire LDOM pool is available but only a limited portion of the SDOM
is available for bacterial utilization to account for lower lability of SDOM. Refractory DOM (RDOM) is not explicitly modelled
due to its much longer turnover time than labile and semi-labile pools, but some mass flows are included to RDOM from other
prognostic model compartments, such as bacteria, krill, and SDOM, to account for loss terms for those state variables. Detritus
represents an average particulate organic matter (POM) pool after removing living phytoplankton and bacterial biomasses, and
85   sinking of the detritus pool contributes to export flux. The WAP-1D-VAR model explicitly simulates $NO_3$, $NH_4$, and $PO_4$ for
inorganic (macro)nutrient compartments, but there is not a separate Fe model compartment or Fe uptake processes, given that
iron limitation is absent or occurs only minimally and seasonally in the nearshore Palmer Station area (Sherrell et al., 2018).

**2.2 Model equations**

Here we describe key model processes that are either based on existing schemes or built as new, modified schemes
90   for the WAP. The original model schemes are detailed in Supplementary Material of Luo et al (2010). The WAP-1D-VAR
model simulates biological-physical model processes for a 1-D vertical water column, solving numerically for a discretized
version of the time-rate of change for each model state variable. For a generic tracer variable $C$ the time-rate of change equation
takes the form (Glover et al., 2011):

$$\frac{\partial C}{\partial t} = -\frac{\partial}{\partial z}\left(wC\right) + \frac{\partial}{\partial z}\left(K_z \frac{\partial C}{\partial z}\right) + J_C \qquad (1)$$

95   where $z$ is the depth, $w$ is the vertical velocity (the sum of water motion and gravitational particle sinking), $K_z$ is the turbulent
eddy diffusivity (Eq. 1), and $J_C$ is the biological and biogeochemical net source and sink term for $C$ (Appendix Equations Eq.
A.2.41-44, A.3.37-40, A.4.53-55, A.5.24-26, A.6.27-29, A.7.4-6, A.8.4-9, A.9.2-4). The physical advection and mixing terms
are discussed below in section 2.3 and are applied sequentially following the computation of the biological and biogeochemical
terms $J_C$ using a constant time step $\Delta t$ of 1 hour. The contributions of the source sink terms $J_C$ to the full time rate of change
100   equations are constructed as a series of coupled ordinary differential equations, detailed in Appendix (sections A1-9), and are
solved using a second-order Runge-Kutta numerical integration scheme. The WAP-1D-VAR model simulates the dynamics
of C, N, and P, but here we only focus on the presentation of the model C dynamics. The cellular molar (N/C or P/C) quota
parameters of most state variables are fixed and not submitted to the optimization and data assimilation procedure. To first
order, a variety of the model physiological processes are affected by water temperature, including the maximum growth rates
105   of phytoplankton, bacteria, and zooplankton and basal respiration rates of bacteria and zooplankton. The Arrhenius function
is implemented to change these physiological rates as a function of water temperature (Eq. A.1.1).

The net change of phytoplankton (both diatoms and cryptophytes) C biomass is driven by their gross growth, DOC
excretion, particulate organic C (POC) production via aggregation, respiration, and grazing (Eq. A.2.41, A.3.37), the net
change of their N and P biomass by gross growth, dissolved organic N(P) (DON(P)) excretion, particulate organic N(P)
110   (PON(P)) production, and grazing (Eq. A.2.42-43, A.3.38-39), and the net change of their chlorophyll *a* (Chl) by gross growth,
DOM excretion, and grazing (Eq. A.2.44, A.3.40). The WAP-1D-VAR model adapts a phytoplankton growth scheme with
flexible stoichiometry, in which phytoplankton cells are allowed to accumulate and store more nutrients under light stress
(Bertilsson et al., 2003; Droop, 1974, 1983; McCarthy, 1980). The phytoplankton C growth rate is limited by their cellular
nutrient contents (quota) (Eq. A.2.1-2, A.3.1-2). Modified from Geider et al. (1998), phytoplankton N uptake decreases when

their cellular N/C quota is higher than their reference (Redfield) ratio, but not limited when lower than their reference ratio (Eq. A.2.4, A.2.8, A.3.4, A.3.8). The N consumption completely ceases when the phytoplankton cellular quota reaches their maximum allowable ratios and additionally limited by the ambient nutrient concentrations with a Monod function (Eq. A.2.10-11, A.3.10-11). $NH_4$ inhibition on $NO_3$ uptake is modelled by assigning lower $k^{NH4}$ compared to $k^{NO3}$ (Table 2). The inhibition term does not exist for $PO_4$. The uptake scheme is similar for P (Eq. A.2.13, A.3.13), but P can be consumed in great excess of current needs (Armstrong 2006). Such luxury uptake is modelled by assigning smaller maximum and minimum P quota (Table 2), which acts to alleviate P limitation. The maximum photosynthesis rate decreases when the phytoplankton cellular quota is lower than their reference ratio, and approaches zero near their minimum ratio (Eq. A.2.6, A.3.6). The Chl production decreases with lowering photosynthetic active radiation (PAR) and completely ceases under no light, i.e., PAR = 0 (Eq. A.2.14, A.3.14). Phytoplankton release LDOM via passive diffusion of the low molecular weights DOM (e.g., neutral sugars and dissolved free amino acids) with the same cellular elemental ratio as that of phytoplankton (Fogg 1966, Bjørnsen 1988, Biddanda and Benner 1997) (Eq. A.2.16-18, A.3.16-18). Phytoplankton also release L- and SDOM actively, such as carbohydrate, as 75% of the labile (Eq. A.2.19, A.2.23, A.3.19, A.3.23) and 25% of the semi-labile pools (Eq. A.2.20, A.2.26, A.3.20, A.3.26). This active DOM production enables phytoplankton to adjust their stoichiometry to approach their reference ratio. If cellular organic C is in excess, DOC is released on a time scale of 2 days, and if excess N(P), DON(P) is released on a time scale of 8 days (Eq. A.2.21-22, A.3.21-22). Diatoms are grazed by both microzooplankton and krill (Eq. A.2.33-40), while cryptophytes are only grazed by microzooplankton (Eq. A.3.33-36). Microzooplankton grazing functions are altered by assigning grazing limitation terms ($\epsilon$) to provide a limit on diatom grazing and route more cryptophytes to microzooplankton (Eq. A.2.33, A.3.33), based on initial modelling attempts where elevated diatom Chl were not simulated due to their much stronger removal by microzooplankton than cryptophytes, even with a varying model initial parameter set and adjusted baseline parameters.

The net change of bacterial biomass is driven by their gross growth (via L- and SDOM uptake; Eq. A.4.12-14, A.4.15-16, A.4.21-22), respiration (Eq. A.4.25), S- and RDOM excretion (Eq. A.4.26-28, A.4.35-43), grazing (Eq. A.4.44-46), and mortality due to viral attack (Eq. A.4.47-49). The WAP-1D-VAR model allows both L- and SDOM as the sources for bacteria, and bacterial nutrient quota let the lability of SDOM variable for their selective utilization. All the LDOM pool is available, while only a limited portion of the SDOM pool is allowed for bacterial utilization, the degree of which is controlled by an optimizable parameter controlling the relative utilization of SDOM to LDOM, or SDOM lability (Eq. A.4.11, Table 2). Bacterial C growth is determined by their cellular quota and available L- and SDOM concentration (Eq. A.4.12-13), where the growth would be limited if bacterial cellular N(P) quota is smaller than their reference ratios (Eq. A.4.8-9). Bacteria take up LDOM in the way that the ratio of LDON(P) to LDOC uptake is the same as the bulk N(P)/C ratio of the LDOM (Eq. A.4.15, Eq. A.4.21). Bacteria take up SDOM with higher N/P ratios to reflect that SDOM with higher N/P ratios is more labile (Eq. A.4.13). The ratio of SDON to SDOC uptake by bacteria would vary between the bulk N/C of SDOM and the bacterial reference cellular quota (Eq. A.4.16, Eq. A.4.22). Bacteria are modelled to either take up or release $NH_4$ and $PO_4$ to maintain their rather stable and consistent stoichiometry (Kirchman, 2000). Bacteria take up $NO_3$ only if their cellular N/C ratio is smaller than their reference ratio (i.e., when bacteria are in short of N), in order to reflect higher energetic cost of $NO_3$ uptake than $NH_4$, but the amount of $NO_3$ uptake is modelled to be no more than 10% of N-specific bulk L- and SDOM uptake, and the $NO_3$ plus $NH_4$ uptake is modelled to be no more than N-specific bulk L- and SDOM uptake (Eq. A.4.17-20). These limit the maximum $NO_3$ uptake rate and set the inhibition of $NH_4$ uptake on $NO_3$ uptake. Bacteria excrete RDOM by transforming LDOM to RDOM (A4.26-28). Bacteria adjust their cellular stoichiometry by remineralizing $NH_4$ and $PO_4$ if C in short (i.e., N and P in excess; Eq. A.4.29-30) and by excreting SDOC if C is in excess (i.e., N and P are in short; Eq. A.4.38-43). Bacteria are grazed by microzooplankton (Eq. A.4.44-46), and a certain percentage of bacteria gets lost to LDOC pool due to viral attack (Eq. A.4.47-49).

The net change of zooplankton (both microzooplankton and krill) biomass is driven by their gross growth (via grazing on preys; Eq. A.5.3-5, A.6.3-5), L- and SDOM excretion (Eq. A.5.6-14, A.6.6-14), respiration (Eq. A.5.17, A.6.17), POM

production (Eq. A.5.18-20, A.6.18-20), and grazing (Eq. A.5.21-23, A.6.24-26). Microzooplankton C growth is supported by
160 grazing on cryptophytes and bacteria (Eq. A.5.3-5), while krill C growth is supported by grazing on diatoms and
microzooplankton (Eq. A.6.3-5). Both zooplankton compartments follow the Holling Type 2 density-dependent grazing
function with a preferential selection on different prey species (Eq. A.2.33, A.2.37, A.3.33, A.4.44, A.5.21). Both zooplankton
groups release a portion of the organic matter that they ingest as DOM via sloppy feeding and excretion (Eq. A.5.6-8, A.5.9-
11, A.6.6-8, A.6.9-11) such that the ratio of the released DON(P) to LDOC is equivalent to the N(P)/C ratio of zooplankton.
165 The amount of SDOC excretion is a function of the total C growth (Eq. A.5.9, A.6.9), while the amount of SDON(P) excretion
is also a function of the zooplankton cellular N(P)/C ratio relative to their reference ratio (Eq. A.5.10-11, A.6.10-11).
Zooplankton adjust their body cellular quota by either releasing SDOM if C is in excess, or by regenerating $NH_4$ or $PO_4$ if N
or P is in excess (Eq. A.5.12-16, A.6.12-16), similar to the bacterial scheme. Respiration is formulated such that basal
respiration is based on a portion of zooplankton biomass, while active respiration is based on a portion of their grazed C (Eq.
170 A.5.17, A.6.17). Both zooplankton egest fecal matter as POM (Eq. A.5.18-20, A.6.18-20), but only krill additionally excrete
RDOM with N/C and P/C similar to bacteria (Eq. A.6.21-23). Microzooplankton are grazed by krill (Eq. A.5.21-23), while
krill are removed by implicit higher trophic levels (Eq. A.6.24-26), similarly calculated as a bacterial mortality term, rather
than as an explicit grazing process.

The net change of detritus is driven by POM produced by all phyto- and zooplankton compartments that is routed to
175 detrital pool (Eq. A.2.29-32, A.3.29-32, A.5.18-20, A.6.18-20) and its dissolution (Eq. A.7.1-3). An optimizable vertical
sinking speed is assigned to detritus to derive particle export fluxes (i.e., particle export flux = detrital concentration × particle
sinking velocity, *wnsv* in Table 2). The detritus that is lost due to dissolution is incorporated to SDOM pool when it sinks (Eq.
A.8.7-9), before regenerated to inorganic nutrients, rather than directly regenerated from as a particulate form. The net change
of LDOM is driven by LDOM excretion by all phyto- and zooplankton compartments (Eq. A.2.16-19, A.3.16-19, A.5.6-8,
180 A.6.6-8) and the amount of bacterial mortality that is incorporated to LDOM due to viral attack (Eq. A.4.47-49) and its uptake
by bacteria (Eq. A.4.12). The net change of SDOM is driven by SDOM excretion by all organisms (Eq. A.2.16-19, A.3.16-19,
A.5.6-8, A.6.6-8) and the amount of detrital dissolution (Eq. A.7.1-3), its uptake by bacteria (Eq. A.4.13-14), and its conversion
to RDOM pool (Eq. A.8.1-3). The conversion of SDOM to RDOM pool is a function of the stoichiometry of SDOM, in which
the conversion process is slower for higher N/C and P/C of the SDOM, to reflect that N- and P-enriched SDOM are more
185 likely labile. A certain percentage of $NH_4$ is converted to $NO_3$ on a daily basis to represent a simple nitrification process in the
model (Eq. A.9.1).

**2.3 Physical forcing**

The WAP-1D-VAR model v1.0 is forced by mixed layer depth (MLD), photosynthetically active radiation (PAR) at
the ocean surface, sea-ice concentration, water-column temperature, vertical velocity, and vertical eddy diffusivity, at a
190 temporal resolution of 1 day. Temperature, sea ice, and vertical eddy diffusivity are set up at every vertical grid (depth) point.
MLD is determined based on a finite difference density criterion with a threshold value of $\Delta\sigma_\theta = 0.03$ kg m$^{-3}$ (Montégut
et al., 2004), after calculating potential density of water mass from temperature and salinity Conductivity-Temperature-Density
(CTD) observations. Vertical velocity, *w*, is assigned as zero because it is very weak in the surface waters of the study site (see
3.1), and materials are transported vertically mostly by diffusion. The vertical eddy diffusivity scheme treats the rapid vertical
195 mixing in the surface boundary layer by homogenizing model state variables instantaneously in the mixed layer by averaging
their values at every time step. Thus, $K_z$ value above MLD is not required, and only $K_z$ below MLD is calculated as follows:

$$K_z(z) = K_{z0} \times \exp\{-\alpha \times (z - MLD)\} \quad\quad\quad (2)$$

where z is depth (m) below MLD, $K_{Z0}$ is the vertical eddy diffusivity at the bottom of the mixed layer ($1.1 \times 10^{-4}$ m$^2$ s$^{-1}$) (Klinck,
1998; D. A. Smith et al., 1999), and $\alpha$ is 0.01 (m$^{-1}$).

[Figure]

[Figure]

200    Daily surface downward solar radiation flux (National Centers for Environmental Prediction reanalysis daily averages) is used to calculate sea surface PAR. PAR is estimated as 46% of the total solar radiation (Pinker & Laszlo 1992, Kirk 1994). The attenuation of PAR as a function of depth is calculated as follows:

$$PAR\ (z) = PAR_0 \times \exp\{ -(k_w + k_c \times CHL) \times z \} \qquad (3)$$

where $z$ is depth (m), $PAR_0$ is PAR level at sea surface (W m$^{-2}$), $k_w$ is the attenuation coefficient for seawater (m$^{-1}$), $k_c$ is the
205    attenuation coefficient for Chl ((mg Chl)$^{-1}$ m$^2$), and $CHL$ is the Chl concentration (mg Chl m$^{-3}$).

    Sea ice conditions in the coastal WAP do not necessarily represent solely local temperature and climate conditions, given that sea-ice concentration can be impacted by temperature, mixed layer, heat fluxes, regional winds, and other physical processes (Saenz et al. *in review*). We implement a sea-ice model scheme to account for light transmission through sea ice (5% of incident irradiance, a typical transmittance value used in the Community Earth System Model) and non-linearities in the
210    photosynthesis-irradiance (P-I) response under partial ice concentration (Long et al. 2015), using percent daily sea-ice concentration data (GSFC Bootstrap versions 2/3, derived from SMMR/SSMI satellite temperature brightness data binned into 25 by 25 km grid cells). In many previous models, the light-limitation term $\mathcal{L}(I)$ is calculated as a function of mean irradiance $\bar{I}$ averaged over both ice-covered and open-water conditions, so $\mathcal{L}(\bar{I})$; instead we compute the mean of light-limitation term $(\overline{\mathcal{L}(I)})$ as a function of fractional sea ice and open water and incident irradiance:

215    $$\mathcal{L}(I) = P^C/P^C_{MAX} = 1 - \exp(-I/I_k) \qquad (4)$$

$$\overline{\mathcal{L}(I)} = f_i \times \mathcal{L}(I_i) + f_o \times \mathcal{L}(I_o) \qquad (5)$$

where $P^C$ is the C-specific photosynthetic rate (d$^{-1}$), $P^C_{MAX}$ is the maximum photosynthetic rate (d$^{-1}$), $I_k$ is the parameter describing the light-saturation behaviour of the PI-curve (W m$^{-2}$), $I_o$ is the open-water irradiance, $I_i$ is the under-ice irradiance (i.e., $I_i = 0.05 \times I_o$), $f_i$ is the fraction of area covered with sea ice, and $f_o$ is the fraction of open water (i.e., $f_o = 1 - f_i$).

**2.4 Variational data assimilation**

    The WAP-1D-VAR model v1.0 is equipped with a built-in data assimilation scheme based on a variational adjoint method (Lawson et al., 1995). By objectively optimizing model parameters, the assimilation method generates optimal solutions of the model that minimize model-observation misfit (Friedrichs, 2001; Spitz et al., 2001; Ward et al., 2010). Initial guesses of the model parameters (total of 72 free or optimizable parameters, Table 2) are assigned based on literature values,
225    or estimated directly using a subset of the observations. The data assimilation scheme (Figure 2) consists of four steps (Glover et al., 2011): 1) starting with an initial guess of the model parameter set, the model is integrated forward in time from specified initial conditions to calculate the difference between the model simulation and the field data, the model-observation misfit (i.e., cost function; see 2.5); 2) an adjoint model constructed using the Tangent linear and Adjoint Model Compiler (TAPENADE) is integrated backward in time to compute the gradients of the total cost with respect to the model parameters;
230    3) the computed gradients are then passed to a limited-memory quasi-Newton optimization software M1QN3 3.1 (Gilbert & Lemaréchal, 1989) to determine the direction and optimal step size by which the model parameters need to be modified in order to reduce the total cost; and 4) a new forward in time simulation is conducted using the new set of modified parameters. These procedures are conducted in an iterative manner until the pre-set convergence criteria are satisfied to ensure that the optimized parameters converge and the total cost reaches a local minimum. The pre-set convergence criteria include the low
235    gradients (sensitivity) of the total cost function with respect to model parameter from optimization and positive eigenvalues of the Hessian matrix. See Supplementary Material of Luo et al. (2010) for more details.

    When computed at the minimum of the cost function value, the inverse of the Hessian matrix provides the uncertainties of optimized parameters, cross-correlations among parameters and sensitivities of the total cost to each parameter (Matear, 1996; Tziperman & Thacker, 1989). High off-diagonal values in the inversed Hessian matrix indicate highly cross-
240    correlated model parameters, and one of the highly cross-correlated parameters is therefore removed from the optimization. The square root of a diagonal element in the inversed Hessian matrix is the logarithm of the relative uncertainty of the

corresponding optimized parameter (i.e., $\sigma_f$). An optimized parameter with $\sigma_f$ larger than 50% is updated but removed from the next optimization cycle, while the optimized parameter with $\sigma_f$ smaller than 50% is updated and kept for the next optimization cycle. We define the optimized parameters with $\sigma_f$ smaller than 50% as 'optimized' parameters and the optimized parameters

245 with $\sigma_f$ larger than 50% as 'updated parameters', with the former and the latter reported with and without their error ranges, respectively (Table 2). In other words, both changed parameters consist of an optimized model parameter set, but the parameters reported with their error ranges are the ones optimized with relatively small errors and therefore considered constrained. The absolute uncertainty of the constrained parameter is calculated as $p_f \times \exp(\pm\sigma_f)$ where $p_f$ is the value of the optimized parameter. (Table 20).

250 **2.5 Cost function**

To represent a misfit between observations and model output, a total cost function is defined as follows (Luo et al., 2010):

$$J = \sum_{m=1}^{M} \frac{1}{N_m} \sum_{n=1}^{N_m} \left( \frac{\hat{a}_{m,n} - a_{m,n}}{\sigma_m} \right)^2 \tag{5}$$

where $m$ and $n$ represent assimilated data types and data points, respectively, $M$ and $N_m$ are the total number of assimilated
255 data types and data points for data type $m$, respectively, $\sigma_m$ is the target error for data type $m$, $a_{m,n}$ is observations, and $\hat{a}_{m,n}$ is model output. Given the high biological productivity of the WAP waters and the approximate log-normal distribution of many marine biological variables, the base-10 logarithms of Chl and primary production (PP) are used in the cost function calculation to capture phytoplankton dynamics (Campbell, 1995; Glover et al., 2018). The target error is calculated for each data type as:

$$\sigma_m = \overline{a_{m,n}} \times \mathrm{CV}_m \tag{6}$$

260 where $\overline{a_{m,n}}$ is the climatological mean of the observations and $\mathrm{CV}_m$ is the coefficient of variation (CV) of the observations of each data type in the mixed layer (due to observational error and seasonal and interannual variations) over 2002-2003, 2003-2004, 2004-2005, 2005-2006, 2006-2007, 2008-2009, 2009-2010, 2010-2011, and 2011-2012 period (for growth seasons' relatively complete data coverage for modelling purposes). The standard deviations are used as target errors of the log-converted data types. The CV of the log-converted data type is estimated as the average of ± 1 standard deviation in log space
265 converted back into normal space (Doney et al., 2003; Glover et al., 2018). Hereafter, we present the total cost normalized by $M$ ($J$ equivalent to $J/M$ hereafter) as it indicates the model-observation misfit equivalent to a reduced Chi-square estimate of model goodness of fit. We report the normalized total cost $J$ along with normalized costs of individual data types ($J_m$) throughout this article. $J = 1$ indicates a good fit from optimization, $J \gg 1$ indicates a poor fit or underestimation of the error variance, and $J \ll 1$ indicates an overfitting of the data, fitting the noise, or overestimation of the error variance.

270 **3 Model experiments**

**3.1 Modelling framework**

To examine the applicability of the WAP-1D-VAR model v1.0 to the coastal WAP region, we selected a nearshore Palmer LTER water-column time-series station, Station E (64.77°S, 64.05°W), as the modelling site, that is situated approximately 3 km south of Palmer Station and 6.5 km northeast of the head of Palmer Deep (Sherrell et al., 2018). Physical
275 forcing (Figure 3) and data types assimilated were derived from roughly semi-weekly physical, chemical and biological profiles collected from small boat via a profiling CTD and discrete water samples at Station E. When weather and ice conditions permitted, water column sampling at the station was conducted twice a week over the growth season since 1991. Seven upper-ocean layer depths (2.5, 10, 20, 30, 40, 50 and 60 m) were chosen for the model vertical grids. Given that routine observations

of Palmer LTER were conducted over the growth season (October - March), we simulated one example growth season with
the most complete data coverage, from October 2002 to March 2003 (2002-2003 growth season hereafter), instead of a series
of different growth seasons in a continuous manner. The example growth season simulations utilized that year's specific
observed physical forcing fields and assimilated biological and biogeochemical observations. In other words, each Palmer
LTER growth season would have its own unique optimized parameter set, as well as initial conditions and physical forcing
that together determine the model solution for that year; however, only 2002-2003 growth season simulations are modelled in
the present study for model analysis and evaluation.

**3.2 Initial and boundary conditions**

Model initial conditions were prescribed 135 days before the model start date for the growth season (October 15,
2002), so on June 1, 2002. This 135-day spin up was conducted to minimize the impact of initial conditions on the model
output over the growth season. Initial conditions were prepared by first generating an optimized model simulation of the full
growth seasonal cycle forced by climatological physics and assimilated with climatological observations (i.e., climatological
model; using climatological physics and observations averaged over 2002-2003, 2003-2004, 2004-2005, 2005-2006, 2006-
2007, 2008-2009, 2009-2010, 2010-2011, and 2011-2012 period). Due to strong interannual variability in the phytoplankton
bloom phenology at Palmer Station, averaging across all these 9 years did not reflect distinct seasonal phytoplankton peaks,
leading to underestimated phytoplankton values (not shown). To capture this non-linear aspect of the coastal WAP system, we
constructed the climatological year by applying a single time shift to all variables so that a seasonal PP peak of each year lined
up with an average date of seasonal PP peaks from all years. Most biological initial conditions on June 1 were close to zero
given the lack of active physiological processes in the very low light and the presence of sea ice during wintertime before the
model growth season started. All the data types were set to zero at the lower boundary (bottom) except for $NO_3$, $PO_4$, SDOC,
SDON, and SDOP in which the climatological values at 65 m were used for lower boundary values (25.9 mmol m$^{-3}$, 1.9 mmol
m$^{-3}$, 6.5 mmol m$^{-3}$, 0.6 mmol m$^{-3}$, and 0.03 mmol m$^{-3}$, respectively).

**3.3 Assimilated data**

We included data types directly related to corresponding model outputs, including a mix of ecosystem stocks or state
variables – $NO_3$, $PO_4$, Chl for diatoms and cryptophytes, bacterial biomass, microzooplankton biomass, SDOC, POC, and PON
as well as C flows among model stocks – bulk net PP and bacterial production (BP). The data were sampled semi-weekly at
Palmer Station E (64.77°S, 64.05°W), the same location where our model was set up, and were downloaded from the Palmer
LTER data website (see Availability of Data and Model Simulations). The distinction between diatoms and cryptophytes was
established by assimilating phytoplankton taxonomic-specific Chl data for diatoms and non-diatom species derived from a
High-Performance Liquid Chromatography (HPLC) and CHEMTAX analysis (Schofield et al., 2017), but given cryptophytes
being the second dominant species in the water samples they are considered to represent non-diatom species. Given that
POC(N) from bottle filtration might capture both living biomass and detrital material, we adjusted the observed POC(N) by
subtracting phytoplankton and bacterial C(N) biomass to estimate the detrital pool, in order to only include non-living particles
to detrital pool. When phytoplankton or bacterial biomass data were not available, we assigned climatological (2002-2003 to
2011-2012) fractions of POC(N) to detrital pool. Phytoplankton- and bacterial biomass accountd for 74% of total POC and
71% of total PON. In converting Chl to phytoplankton C(N) biomass, the maximum Chl/C(N) ratio submitted for optimization
was used along with other reference ratios (Table 2). Microzooplankton biomass data were not available for the full time-
series, so microzooplankton biomass data from grazing experiments at Palmer Station (L. M. Garzio et al., 2013) were
assimilated to at least provide constraints on bacterial and cryptophyte grazing processes. However, due to the discrepancy in
the timing and location from model simulations of this study, the microzooplankton model-observation misfits were not

examined in the present study. Krill biomass data were not assimilated due to the strong patchiness of their distribution that
320  might hinder proper model optimization. The vertical profiles of most of the data types were assimilated, whereas average $NO_3$
and $PO_4$ concentrations in the mixed layer were assimilated due to the difficulty of simulating depth-dependent nutrient
concentrations and the fact that net PP was mostly determined by surface nutrient concentrations (Luo et al., 2010). BP (mmol
C m$^{-3}$ d$^{-1}$) was derived from the $^3$H-leucine incorporation rate (pmol l$^{-1}$ h$^{-1}$) data using the conversion factor of 1.5 kg C (mol
leucine)$^{-1}$ incorporated (H. W. Ducklow, 2000). Bacterial biomass (mmol C m$^{-3}$) was estimated from bacterial abundance
325  measured by flow cytometry with the conversion factor of 10 fgC cell$^{-1}$ (Fukuda et al., 1998). SDOC was calculated by
subtracting the background concentration (41.2 mmol m$^{-3}$ for the modelling site) from total DOC concentration.

**3.4 Uncertainty analysis**

Uncertainties of the optimized parameters were computed from a finite difference approximation of the complete
Hessian matrix (i.e., second derivatives of the cost function with respect to the model parameters) during the iterative
330  optimization process. We then conducted Monte Carlo experiments to calculate the impact of the optimized parameter
uncertainties on the model results. We created an ensemble of parameter sets ($N = 1,000$) by randomly sampling values within
the uncertainty ranges of the constrained parameters, and then performed a model simulation using each parameter set. 1,000
Monte Carlo experiments were shown to be adequate from a series of tests with different numbers of Monte Carlo sampling
($N = 500$-$2,000$), where standard deviations of model simulated values converged after >1,000 Monte Carlo sampling (not
335  shown). All uncertainty estimates were calculated following standard error propagation rules and presented as ± 1 standard
deviation in the article.

**4 Results and discussion**

**4.1 Model skill assessment**

In the case of the example growth season modelled in the present study (2002-2003 growth season), the iterative data
340  assimilation-parameter optimization procedure reduced by 80% the misfits between observations and model output compared
to the misfits with the initial guess parameters (Table 1). The optimized model solution also satisfied the pre-set convergence
criteria, with the low gradients of the total cost with respect to the optimized parameters and positive eigenvalues of the Hessian
matrix. Overall, there was a good model-data fit with largely decreased cost for each data type after optimization (Table 1).
There were three data types ($NO_3$, diatom Chl, and bacterial biomass) that indicated a good fit from optimization, with their $J_f$
345  near 1, while none of the data types indicated a poor fit, or underestimation of the error variance. The average error or bias
($\varepsilon_{bias}$) was calculated to evaluate the model over- or underestimation of the observations (Doney et al., 2009; Stow et al., 2009).
$\varepsilon_{bias}$ indicated negative biases (i.e., model underestimation of the observations) for $NO_3$, diatom Chl, cryptophyte Chl, bacterial
biomass, SDOC, POC, and PON, while positive biases (i.e., model overestimation of the observations) for $PO_4$, PP, and BP
(Table 1). Some of these model biases cases were evidently shown on a point-to-point basis, especially for POC and PON
350  (Figure B1). Model skill was further evaluated with a Taylor diagram (Taylor, 2001) summarizing the statistics of the
correlation coefficient, normalized standard deviation (by the standard deviation of the observations), and centred (bias
removed) root-mean-square difference (RMSD) for each data type (Figure 4). There were five data types with relatively high
correlation coefficients (r > 0.5) and five data types with relatively low correlation coefficients (r < 0.5). The data types with
relatively high correlation coefficients tended to have relatively low centred RMSD and vice versa. Together, the model fitted
355  average observations slightly better than the seasonal- and spatial (vertical) variability, given that the changes in the phase and

relative amplitude of the modelled variables relative to observations were not always strongly similar, despite small absolute model-observation differences for most data types.

**4.2 Optimized parameters**

Among the total of 72 optimizable model parameters, subsets of 14 (5 optimized and 9 updated) parameters changed
360  compared to their initial guess values (Table 2). The number of the optimized parameters in the present study was small and comparable to the results from optimization of other regional test-bed models (Friedrichs, 2001; Friedrichs et al., 2006, 2007; Luo et al., 2010). This is consistent with the general behaviour of marine plankton ecosystem models, in which well-posed model solutions would be found with only a subset of independent model parameters due to many cross-correlated parameters inherent in nonlinear model equations (Fennel et al., 2001; Harmon & Challenor, 1997; Matear, 1996; Prunet, Minster,
365  Echevin, et al., 1996; Prunet, Minster, Ruiz-Pino, et al., 1996). Ecosystem models with a relatively large number of unconstrained parameters (i.e., optimized parameters with high uncertainties, or updated parameters in the present study) might reduce total costs to a greater extent, but could possess low predictive skill as a result of being over-tuned to fit noise in the observations (Friedrichs et al., 2007).

The optimized model results at each model time step and grid were associated with generally small errors derived
370  from the Monte Carlo experiments (Figure B2). To examine the translation of optimized parameters to altered functioning of the WAP biogeochemical processes, we compared two different sets of the model simulation results – one based on the initial guess parameters (i.e., before optimization; Figure B3) and the other based on the optimized parameters (i.e., after optimization; Figure 5, Figure B2). However, due to the non-linearities in the model it was not straightforward to identify what caused the parameter variations, except for a few cases in which the changes in the parameter values were clearly
375  linked to the difference in the model results. One case was the relation of optimized $\Theta$ (maximum Chl/N ratio, g Chl $a$ (mol N)$^{-1}$) value to phytoplankton accumulations slightly more dominated by diatoms. The variations of $\Theta$ are fundamentally driven by an imbalance between the rate of light absorption and energy demands for photosynthesis and biosynthesis in phytoplankton cells (R. J. Geider et al., 1997). $\Theta$ can also change because of variations in phytoplankton photo-acclimation or physiological differences across phytoplankton groups, from a lower $\Theta$ value for smaller species to a higher $\Theta$ value for
380  larger diatom cells (Richard J. Geider, 1987). In our case, $\Theta$ was optimized to a value that is 10% higher than its initial guess value (Table 2), in order to simulate a relatively larger (smaller) proportion of diatoms (cryptophytes) in total phytoplankton biomass. $\mu_{KR}$ (maximum krill C-specific growth rate, d$^{-1}$) was optimized to about 2 times larger value than its initial guess value (Table 2), which resulted in larger krill accumulations in the optimized model results (Figure 5), compared to the unoptimized model results (Figure B3). By contrast, the remaining cases of the other parameters is less clear because the
385  first-order impact of parameter variations on model results is less direct and more nuanced. Compared to the unoptimized results (Figure B3), $diss$ (detrital dissolution rate, d$^{-1}$) was optimized to 5 times smaller value relative to its initial guess value (Table 2), but overall there was decreased detrital accumulation in the optimized results (Figure 5) due to increased POC production by cryptophytes, microzooplankton, and krill in the optimized results (not shown). Similarly, compared to the unoptimized results, increased $\alpha_{DA}$ (initial slope of P-I curve of diatoms, mol C (g Chl)$^{-1}$ d$^{-1}$ (W m$^{-2}$)$^{-1}$), $g_{DA}$ (diatom half-
390  saturation concentration in microzooplankton grazing, mmol C m$^{-3}$), and $g'_{DA}$ (diatom half-saturation concentration in krill grazing, mmol C m$^{-3}$) did not necessarily lead to increased diatom accumulations, due to increased $\mu_{MZ}$ (microzooplankton C-specific maximum growth rate, d$^{-1}$) and $\mu_{KR}$ (maximum krill C-specific growth rate, d$^{-1}$). The three other cases were similarly interpreted: 1) The decreased $\alpha_{CR}$ (initial slope of P-I curve of cryptophytes mol C (g Chl)$^{-1}$ d$^{-1}$ (W m$^{-2}$)$^{-1}$) and increased $\mu_{MZ}$ (microzooplankton C-specific maximum growth rate, d$^{-1}$), despite increased $g_{CR}$ (cryptophytes half-saturation
395  concentration in microzooplankton grazing, mmol C m$^{-3}$), generated smaller cryptophyte accumulations in the optimized results; 2) the increased $r^A_{MAX,BAC}$ (bacterial maximum active respiration rate, d$^{-1}$) and $\mu_{MZ}$ (microzooplankton C-specific maximum growth rate, d$^{-1}$), despite increased $\mu_{BAC}$ (maximum bacterial growth rate, d$^{-1}$) and $g_{BAC}$ (bacterial half-saturation

concentration in microzooplankton grazing, mmol C m$^{-3}$), generated smaller bacterial accumulations in the optimized results; and 3) the increased $\mu_{MZ}$ (microzooplankton C-specific maximum growth rate, d$^{-1}$) and $g_{MZ}$ (microzooplankton half-

400 saturation concentration in krill grazing, mmol C m$^{-3}$), despite increased $\mu_{KR}$ (maximum krill C-specific growth rate, d$^{-1}$) and $r^{A}_{MZ}$ (microzooplankton active respiration rate, d$^{-1}$), generated larger microzooplankton accumulations in the optimized results.

**4.3 Ecosystem indices**

We examined key ecosystem indices for the modelled year, including NPP (directly comparable to [14]C-PP
405 observations), net community production (NCP; i.e., NCP = NPP – bacterial-, microzooplankton-, and krill respiration), BP integrated over the 60-m water column, and POC export (sinking) flux at 60-m (Figure 6, Figure B5). NPP increased after complete sea-ice retreat, but a brief ice-edge bloom was simulated under sea ice at the beginning of the growth season (Figures 3, 6). Seasonal patterns of NCP resembled those of NPP and occasionally fell below zero (Figure 6), indicating the net heterotrophy in subsurface waters. The depth-integrated NCP time series was moderately correlated to that of NPP over the
410 growth season (r = 0.78, *p* <0.001). The POC export flux increased over time and reached the maximum values at the end of the growth season (Figure 6). The depth-integrated POC export flux was also moderately correlated to that of NCP, with the POC export flux lagged by 15 days (r = 0.51, *p* <0.001). The depth-integrated BP showed a similar seasonal pattern with and strongly correlated to that of NPP (Figure 6), with BP lagged by 2 days (r = 0.94, *p* <0.001). The growth-season mean of the depth-integrated NPP, NCP, BP, and POC export flux at 60 m were 62 ± 17 mmol C m$^{-2}$ d$^{-1}$, 17.3 ± 19.6 mmol C m$^{-2}$ d$^{-1}$, 1.7
415 ± 0.45 mmol C m$^{-2}$ d$^{-1}$, and 4.7 ± 4.6 mmol C m$^{-2}$ d$^{-1}$ (errors propagated from season-averaging in Figure 6 and Monte Carlo errors in Figure B5). The mean *e*-ratio, defined as the growth-season mean of the time-lagged (by 15 days) POC export flux divided by the annual mean NPP (i.e., particle export efficiency), was 0.08 ± 0.06. The mean *f*-ratio, defined as the amount of NO$_3$ uptake divided by the amount of NO$_3$ and NH$_4$ uptake both, was 0.66 ± 0.18 (errors propagated from season-averaging in Figure 7 and Monte Carlo errors in Figure B5). The mean BP/NPP ratio (also with time lagged BP by 2 days) was 0.03 ± 0.002
420 (errors propagated from season-averaging in Figure 6 and Monte Carlo errors in Figure B5).

Setting an upper limit for lateral or vertical carbon export from the euphotic zone (Dugdale & Goering, 1967), over appropriate time and space scales, NCP is quantitatively equivalent to new production that is supported via external sources of nitrogen (Hugh W. Ducklow & Doney, 2013). The higher mean *f*-ratio relative to the mean *e*-ratio in the present study implied an imbalance between production and export at the study site, at least during the modelled period. Excess new
425 production relative to export production (as derived from sediment traps and [234]Th disequilibrium; see Ducklow et al. 2018) was previously observed in the WAP, presumably due to diel vertical migration, DOM export, lateral export, and diffusive loss of PON via diapycnal mixing (Stukel et al., 2015). Stukel et al (2015) reported up to 5 times larger new production via NO$_3$ uptake than export production via Th-based N export along the coastal WAP. There would be several additional mechanisms driving the discrepancy between production and export. First, given that the assimilated pool of suspended POC
430 in the model formulation would not be a good indicator of a rapidly sinking detrital pool dominating particle export, our WAP-1D-VAR model did not capture large, short-lived particle flux events (e.g., fecal pellets produced by a large swarm of krill), underestimating POC export flux. Second, the WAP-1D-VAR model export scheme did not consider DOC export that would have lowered the production-export discrepancy. Finally, it should be noted that RDOC was not explicitly modelled in the WAP-1D-VAR, due to its much longer time scale than the model time scale, so accumulated and not-exportable RDOC pool
435 would have contributed to derivation of the modelled *e*-ratio from the modelled *f*-ratio. Indeed, the modelled mean *e*-ratio in the present study situated at the lower end of the range of the *e*-ratios measured or estimated in the WAP waters (Hugh W. Ducklow et al., 2018; Sailley et al., 2013; Stukel et al., 2015; Weston et al., 2013). By contrast, the modelled mean BP/NPP ratio corresponded well to the estimates from other measurement- and observation-based studies (Hugh W. Ducklow et al., 2012; Kim & Ducklow, 2016). Relatively low bacterial activity in productive Antarctic waters, typically reflected as a low

440  BP/PP ratio, has been attributed to low LDOM availability for bacterial growth (Kirchman et al., 2009), low temperature (Pomeroy & Wiebe, 2001), or top-down control via grazing and viral lysis (Bird & Karl, 1999).

**4.4 Average C stocks and flows**

We summarized the growth-season means of the C stocks and flows in the entire food web (Figure 7). The WAP-1D-VAR model captured several key WAP ecological and biogeochemical features, including the lack of macronutrient limitation

445  ($NO_3$ and $PO_4$ drawdown by phytoplankton utilization but remaining well above their half-saturation constants, Table 2) and comparable values of the assimilated and non-assimilated model state variables (Hugh W Ducklow et al., 2007; Hugh W. Ducklow et al., 2012, 2018; Kim et al., 2016; Moline et al., 2008; R. C. Smith et al., 2008), providing confidence in the model simulations even for a non-assimilated data type such as krill. For instance, growth-season measurements in 2017-2018 at Palmer Station showed a strongly patchy krill distribution, with the mean biomass of $0.12 \pm 0.04$ mmol C m$^{-3}$ and maximum

450  biomass of 0.57 mmol C m$^{-3}$ when krill were present (unpublished data provided by J. Conroy and D. Steinberg), falling in the range of the modelled krill biomass values ($0.13 \pm 0.06$ mmol C m$^{-3}$; calculated from Figure 7). The WAP-1D-VAR model also simulated several important ecosystem metrics comparable to other statistical modelling studies. For instance, the modelled phytoplankton seasonal patterns in the present study were consistent with physicochemical attributes revealed by a distinct ecological seascape pattern in the coastal WAP (Bowman et al., 2018), including low Chl and high nutrients in the

455  first half of the growth season followed by high Chl and low nutrients in the second half of the growth season. A steady-state solution based, inverse modelling study quantified different food-web states using ecosystem network indices from Palmer LTER annual summer cruises along the WAP shelf region (Sailley et al., 2013). Their network indices included the ratio of C export to total PP (i.e., equivalent to *e*-ratio in the present study) and the ratio of recycling (the sum of flows into respiration and DOC pool) to total PP, where more (less) recycling favourable microbial food-webs were characterized by greater (smaller)

460  ratios of recycling to total PP and smaller (greater) ratios of total C export to total PP (Legendre & Rassoulzadegan, 1996). As discussed above, the modelled mean *e*-ratio in the present study was smaller than the estimates in the inverse modelling study for the WAP shelf region (Sailley et al., 2013), but consistent with their conclusion on the food-web status of the modelled growth season (2002-2003) positioned on the microbial food-web side. The discrepancy in the *e*-ratio values between the present study and Sailley et al. (2013) might be attributed to fundamentally different model formulation (i.e., time-evolving

465  modelling versus steady-state modelling) and optimization approach, or due to relatively strong microbial food-web activity at our coastal site compared to the shelf region. Microbial food-web activity can be approximated by quantifying the amount of fixed C flowing through heterotrophic bacteria (Carlson et al., 1999; del Giorgio & Cole, 1998; H. W. Ducklow, 2000; Hugh W. Ducklow et al., 2012). According to this approach, microbial food-web activity from the optimized model results was around $26 \pm 9\%$, calculated as the ratio of bacterial L- and SDOC uptake to PP (mean ± error from season-averaging and

470  Monte Carlo errors in Figure 7). SDOC appeared be an important bacterial C source when LDOC became scarce as the growth season progressed (Figure 5), and on average, supported $19 \pm 7\%$ of the total bacterial C uptake, or C demand (mean ± error from season-averaging and Monte Carlo errors in Figure 7). Indeed, several observational studies speculated that the WAP bacteria utilize SDOM in short of LDOM (Hugh W. Ducklow et al., 2011; Kim & Ducklow, 2016; Luria et al., 2017).

**5 Summary**

475  We developed the WAP-1D-VAR v1.0 model, a one-dimensional variational data assimilation model specific to the coastal WAP region, evaluated the model performance and robustness using a variety of quantitative metrics, and discussed the model applicability with regard to capturing the key WAP ecological and biogeochemical features using the data from an example growth season, 2002-2003. The data assimilation scheme significantly reduced the model-observation misfits via the

optimized model parameter set that adjusted the simulation to better match the Palmer LTER observations in 2002-2003. We
480   also explored the nuanced question of how the observations influenced the data assimilation process, drove the variations in
optimized parameter values relative to their corresponding initial guess values, and affected the resulting model simulations.
The WAP-1D-VAR model successfully simulated the variables and flows that were not assimilated in the model simulations,
with the values consistent with other measurement- and observation-based studies in the WAP. Importantly, the data
assimilation scheme enabled the available observational data to constrain processes that were poorly understood, such as the
485   partitioning of NPP by different phytoplankton groups, the optimal Chl/C ratio of the WAP phytoplankton community, and
the partitioning of DOC pools with different lability. Up to this point, a range of observational studies has provided snapshots
of ecosystem and biogeochemical processes in the WAP. Yet, we have little understanding of the driving processes that
underlie the connections between each component in complex food-web interactions. We used data-assimilative modelling to
glue these snapshots together to explain better the observed dynamics and further understand the previously poorly constrained
490   processes in the coastal WAP system.

**Acknowledgements**

We thank the many Palmer LTER team members and Palmer Station support staff for collecting, measuring, and
analyzing the field samples and for providing the datasets used in the present study. We also thank Le Xie for providing
495   protocols for the parameter optimization and adjoint models, Cristina Schultz for providing the initial guess values of the
model parameters, Nicole Waite for compiling HPLC data, and Joe Cope and Lori Garzio for compiling the zooplankton data.
Computing resources for model simulations were provided on the Rivanna high-performance computing system by the
Advanced Research Computing Services of the University of Virginia. Kim and Doney were supported by the National
Aeronautics and Space Administration Ocean Biology and Biogeochemistry Program (grant NNX14AL86G) and US National
500   Science Foundation Office of Polar Programs (grant PLR-1440435 to Hugh Ducklow at Columbia University; Palmer LTER).
Schofield and Steinberg were supported by US NSF grant PLR-1440435. Kim was also supported by the Investment in Science
Fund and the Reuben F. and Elizabeth B. Richards Endowed Fund from Woods Hole Oceanographic Institution. Luo was
supported by National Natural Science Foundation of China project 41890802.

505   **Data availability**

Complete Palmer LTER time-series data used for data assimilation are available online (http://pal.lternet.edu/data).
Surface downward solar radiation flux data used for physical forcing of the model simulations are found in the National Centers
for Environmental Prediction website (https://www.esrl.noaa.gov/psd/data/gridded/data.ncep.reanalysis.surface.html).

510   **Code availability**

The Tangent linear and Adjoint Model Compiler (TAPENADE) used to construct an adjoint model is available online
(http://www-sop.inria.fr/tropics/). The current version of WAP-1D-VAR (v1.0) is available from the project website:
https://zenodo.org/record/4470034 under the Creative Commons Attribution 4.0 International license. The exact version of the
model used to produce the results, input data, and scripts to run the model and produce the plots for all the simulations presented
515   in this paper are archived on Zenodo (Kim, H. Heather, Luo, Ya-Wei, Ducklow, Hugh W., Schofield, Oscar M., Steinberg,
Deborah K., & Doney, Scott C. (2021, January 26). WAP-1D-VAR v1.0: A One-Dimensional Variational Data Assimilation
Model for the West Antarctic Peninsula (Version v1.0). Zenodo. http://doi.org/10.5281/zenodo.4470034). A user manual is
available as a separate supplement to this article.

520   **Author contributions**

[Figure]

[Figure]

HHK developed the model, performed model simulations, and wrote the manuscript. YWL contributed to model simulations. HWD, OMS, and DKS provided observational data. SCD supervised the research and significantly the manuscript.

**Competing interest**

525      The authors declare that they have no conflict of interest.

[Figure]

**Figure 1. Ecosystem model.** The model is forced by five different physical forcing, denoted as a horizontal row across the top of the schematic. The ecosystem component incorporates eleven different prognostic state variables.

PAR: photosynthetically active radiation, MLD: mixed layer depth,
Crypto: cryptophytes, Microzoo: microzooplankton, LDOM: labile dissolved oragnic matter,
SDOM: semi-labile dissolved organic matter, RDOM: refractory dissolved organic matter

530

[Figure]

[Figure]

**Figure 2. Variational data assimilation.** A variational adjoint scheme is employed for the parameter optimization and data assimilation processes (adapted from Glover et al., 2011). Gradient: the sensitivity of the total cost function with respect to model parameter from optimization.

[Figure]

535

[Figure]

[Figure]

**Figure 3. Physical forcing.** Physical forcing used in the model, including surface PAR (a), sea ice concentration (b), water temperature (c), and vertical eddy diffusivity (d) overlaid with MLD (dash line) in the modelled growth season.

[Figure]

[Figure]

[Figure]

**Figure 4. Model validation.** A Taylor diagram using a polar-coordinate system summarizing the model-observational correspondence for each model stock and flow for the modelled growth season. The angular coordinate for the Pearson correlation coefficient (r), the distance from the origin for the standard deviation normalized by the standard deviation of the observation, and the distance from point (1,0), marked as REF on x-axis, for the centred (bias removed) root-mean-square difference (RMSD) between model results and observations.

[Figure]

[Figure]

[Figure]

**Figure 5. Model state variables.** The model state variables for the modelled growth season (x-axis; month/day). The error (standard deviation) of each model state variable from the Monte Carlo experiments ($N = 1,000$) is available in Figure SX.

550   Note different contour scales among panels. opt. as the optimized model output.

[Figure]

**Figure 6. Model ecosystem indices.** The key ecosystem indices for the modelled growth season (x-axis; month/day). NPP: net primary production, NCP: net community production, C export flux: particulate organic carbon (POC) export flux, and
555    BP: bacterial production. The error (standard deviation) of each rate process from the Monte Carlo experiments ($N = 1,000$) is available in Figure B5. Note different contour scales among panels. opt. as the optimized model output.

[Figure]

[Figure]

**Figure 7. Mean carbon stocks and flows.** Depth-integrated (0-60 m) C stocks (mmol C m$^{-3}$), flows (mmol C m$^{-3}$ d$^{-1}$), and
560  POC export flux at 60-m (mmol C m$^{-2}$ d$^{-1}$) averaged over the modelled Austral growth season (October-March). Values in
parentheses as errors propagated from season-averaging and depth-integration of the Monte Carlo errors.

[Figure]

[Figure]

**Table 1. Data types, observed means, coefficient of variation, target errors, and costs before and after optimization.**
The observed mean ($\bar{a}$), coefficient of variation (CV), and target error ($\sigma$) of each assimilated data type used for calculating the cost function before and after optimization. $J_0$ is the normalized cost function before optimization and $J_f$ is the normalized cost function after optimization (Eq. 5-6). Data type units: mmol m$^{-3}$ for NO$_3$, PO$_4$; mmol C m$^{-3}$ for microzooplankton biomass, diatom Chl, cryptophyte Chl, bacterial biomass, SDOC, and POC; mmol N m$^{-3}$ for PON; and mmol C m$^{-3}$ d$^{-1}$ for PP and BP. The average error or bias ($\varepsilon_{bias}$) is calculated from Stow et al. (2009), where a positive value indicates the model overestimation of the observation and vice versa.

| Data types | $n$ | $\bar{a}$ | CV | $\sigma$ | $J_0$ | $J_f$ | $\varepsilon_{bias}$ |
|---|---|---|---|---|---|---|---|
| NO$_3$ | 75 | 21.54 | 0.04 | 0.80 | 0.76 | 1.01 | -35.86 |
| PO$_4$ | 75 | 1.43 | 0.03 | 0.05 | 0.54 | 0.39 | 0.36 |
| log$_{10}$ diatom Chl | 86 | -0.07 | 0.20 | 0.08 | 5.47 | 1.05 | -0.75 |
| log$_{10}$ crypto. Chl | 86 | -0.27 | 0.24 | 0.10 | 4.34 | 0.43 | -0.59 |
| log$_{10}$PP | 92 | 1.30 | 0.50 | 0.21 | 0.88 | 0.42 | 14.16 |
| Bacterial biomass | 55 | 0.44 | 0.08 | 0.04 | 9.03 | 1.17 | -0.96 |
| BP | 55 | 0.05 | 0.16 | 0.01 | 0.35 | 0.26 | 0.65 |
| SDOC | 55 | 11.39 | 0.20 | 2.30 | 0.68 | 0.75 | -45.23 |
| POC | 91 | 19.78 | 0.13 | 2.58 | 0.50 | 0.45 | -22.17 |
| PON | 91 | 2.66 | 0.12 | 0.32 | 0.67 | 0.64 | -3.49 |
| Total cost function | | | | | 23.21 | 6.57 | |

[Figure]

[Figure]

**Table 2. Summary of model parameters.** Summary of the model parameter symbol and definition, initial guess ($p_0$) and optimized values ($p_f$) for optimizable parameters, the cost function gradient with regard to the optimized parameter ($\partial J/\partial p$), and prescribed values for fixed model parameters over the course of simulations. The parameter with 'n/a' in the parenthesis is an updated parameter, while the parameter with values in the parenthesis is an optimized parameter with its upper and lower bounds in the parenthesis. The uncertainties for these upper and lower bounds are calculated as: $p_f \times e^{\pm \sigma_f}$ where $p_f$ is the value of the optimized parameter and $\sigma_f$ is the square roots of diagonal elements of the inverse of the Hessian matrix. The cost function gradient with regard to the optimized parameter ($\partial J/\partial p$) after data assimilation defined as: $\Delta J/e^{\Delta p}$ where $e^{\Delta p} \approx \Delta p$ for an infinitely small $\Delta p$. For example, a 10% change of a parameter ($\Delta p = 10\%$) leads to a total cost change equivalent to 10% of the corresponding gradient.

[revised manuscript text omitted]

**Appendix B**

935 **Figure B1.** Comparison of the observations to model results from initial guess values and optimized model parameters. Errors represent how much larger model output is compared to observations. Normalized observation: observations normalized by the mean of each model state variable.

940

[Figure]

**Figure B2.** The errors (standard deviation) of the model state variables for the modelled growth season (x-axis; month/day) from the Monte Carlo experiments ($N = 1,000$). Note different contour scales among panels.

[Figure]

**Figure B3.** The model state variables for the modelled growth season (x-axis; month/day) using the initial parameter guess values (i.e., before optimization). Note different contour scales among panels.

950

[Figure]

[Figure]

**Figure B4.** The ecosystem indices for the modelled growth season (x-axis; month/day) using the initial parameter guess values (i.e., before optimization). Note different contour scales among panels.

[Figure]

[Figure]

[Figure]

955 **Figure B5.** The errors (standard deviation) of the ecosystem indices for the modelled growth season (x-axis; month/day) from the Monte Carlo experiments ($N = 1,000$). Note different contour scales among panels.

[Figure]

[Figure]

[Figure]

---

## Referee Report (RR1)

**Review of "Modelling polar marine ecosystem functions guided by bacterial physiological and taxonomic traits" by Hyewon Heather Kim et al.**

**General**

The authors have made substantial modifications to the modeling configuration and substantial revisions that have significantly improved the manuscript and addressed my previous comments.

I have included additional minor comments on specific sections of the text below. Overall, this manuscript makes a valuable contribution to the community's understanding of bacteria dynamics and to the modeling of the WAP ecosystem and I look forward to seeing the final manuscript published after these final issues are addressed.

**Minor comments / technical corrections**

Line 252 "with relatively high correlations" and line 255 "with lower correlations": Specify if the correlations are significant.

Lines 261, 264, 371, caption of Table 3: The references to the tables in the text and caption appear incorrect. Consider changing "Tables 1-2" to "Tables 2-3" in line 261, and "Table 2" to "Table 3" in lines 264 and 371. In the caption of Table 3, consider changing "Table 1" in "Table 2".

Line 264: Uniform the portability index between Table 3 and the text for 2011-2012.

Lines 266-267: Change "Figure S6" to "Figure S8"

Figure S8: The time evolution for February and March is missing. The time series should cover the whole growth season, i.e. until March 31 as in Figure 4. The multiplication factor is hidden under the title for 2 sub-figures and parentheses are missing in titles.

Line 333: Change "HHA" to "HNA"

Line 373: Change "wth similar" to "with similar"

Lines 383-384: "Assimilating each bacterial group's biomass allows for the partitioning of [...] that were never measured in this study." The sentence is unclear. Do you mean "in this study area"?

Lines 399-400: "The WAP typically exhibits strong interannual variability (Ducklow et al. 2007)" Could you be more specific? Do you mean the WAP meteorological or hydrodynamic conditions, or ecosystem?

Line 476: Change "export" to "sinking flux"

Figure 5: Add arrows for flows 8 and 9 "nutrient uptake' and "regeneration" or remove arrows for the other nutrient flows on the 4 sub-figures.

---

## Author Response (AR2)

**Response to Reviewer #1**

The authors have made substantial modifications to the modeling configuration and substantial revisions that have significantly improved the manuscript and addressed my previous comments. I have included additional minor comments on specific sections of the text below. Overall, this manuscript makes a valuable contribution to the community's understanding of bacteria dynamics and to the modeling of the WAP ecosystem and I look forward to seeing the final manuscript published after these final issues are addressed.

Thank you for your positive feedback. We have revised the manuscript to address each of your specific comments below. We also should note that the related paper on the modelling and data assimilation framework, which we shared with the previous round of revisions, has now been accepted in *Geoscientific Model Development*:

Kim, H. H., Luo, Y.-W., Ducklow, H. W., Schofield, O. M., Steinberg, D. K., and Doney, S. C.: WAP-1D-VAR v1.0: Development and Evaluation of a One-Dimensional Variational Data Assimilation Model for the Marine Ecosystem Along the West Antarctic Peninsula, Geosci. Model Dev., https://doi.org/10.5194/gmd-2020-375, Accepted, 2021

Minor comments / technical corrections
Line 252 "with relatively high correlations" and line 255 "with lower correlations": Specify if the correlations are significant.

Included (line 294).

Lines 261, 264, 371, caption of Table 3: The references to the tables in the text and caption appear incorrect.

All fixed.

Consider changing "Tables 1-2" to "Tables 2-3" in line 261, and "Table 2" to "Table 3" in lines 264 and 371.

All fixed.

In the caption of Table 3, consider changing "Table 1" in "Table 2".

Fixed.

Line 264: Uniform the portability index between Table 3 and the text for 2011-2012.

Fixed.

Lines 266-267: Change "Figure S6" to "Figure S8" Figure S8: The time evolution for February and March is missing. The time series should cover the whole growth season, i.e. until March 31

as in Figure 4. The multiplication factor is hidden under the title for 2 sub-figures and parentheses are missing in titles.

We updated Figure S8 and the figure caption.

Line 333: Change "HHA" to "HNA"

Fixed.

Line 373: Change "wth similar" to "with similar"

Fixed.

Lines 383-384: "Assimilating each bacterial group's biomass allows for the partitioning of [...] that were never measured in this study." The sentence is unclear. Do you mean "in this study area"?

Yes, and we fixed the sentence to "… never measured for each bacterial group in this study area" because the bulk BP has been measured but not the group-specific BP (line 424).

Lines 399-400: "The WAP typically exhibits strong interannual variability (Ducklow et al. 2007)" Could you be more specific? Do you mean the WAP meteorological or hydrodynamic conditions, or ecosystem?

All of them; so we have listed those now (line 448-449).

Line 476: Change "export" to "sinking flux"

Fixed.

Figure 5: Add arrows for flows 8 and 9 "nutrient uptake' and "regeneration" or remove arrows for the other nutrient flows on the 4 sub-figures.

We have removed the arrows for the nutrient flows to only present C stocks and flows and updated the figure captions accordingly for both Figure 5 and Figure S9.

**Response to Reviewer #2**

For the revised ms the authors expanded their model from 0D to 1D (3 layers). The ms has greatly improved, mostly by providing the model equations (although apparently incomplete, see below), whose omission had made it impossible for me to understand the model structure in the previous round. The parameter estimation has much improved and its description has become OK now. But even after the long time it has taken the authors to prepare the revised ms, it still leaves a strong impression of sloppiness. Several sentences are simply incomprehensible and little attention seems to have been paid to the readability, correctness, and design of some of the figures. Only some of the changes to the previous ms are highlighted. The model description in the main text is still very much unclear and this applies also to the mode concept. Nevertheless, having seen the equations, the study seems to be much better than I had feared based on the original ms. After another major revision or two, I now think it could become a useful contribution.

Thank you for your time thoroughly reviewing our manuscript again. Please see below our response specific to each of your comment. We also should note that the related paper on the modelling and data assimilation framework, which we shared with the previous round of revisions, has now been accepted in *Geoscientific Model Development*:

Kim, H. H., Luo, Y.-W., Ducklow, H. W., Schofield, O. M., Steinberg, D. K., and Doney, S. C.: WAP-1D-VAR v1.0: Development and Evaluation of a One-Dimensional Variational Data Assimilation Model for the Marine Ecosystem Along the West Antarctic Peninsula, Geosci. Model Dev., https://doi.org/10.5194/gmd-2020-375, Accepted, 2021

One of the remaining problems is the confusion of assumptions and results. The authors mention in the response letter (introductory para, point 4) that the "larger cell-specific BP and SDOC uptake rates of HNA cells than those of LNA cells" indicate the robustness of their analysis. This finding is also referred to in the results and discussion sections (lines 276, 384–385). But this is a model assumption, not a result: "maximum bacterial growth rate of the HNA group ($\mu$HNA, d-1) was ensured to be optimized to be higher than that of the LNA group ($\mu$LNA, d-1)" (lines 168–169).

While it is true that the maximum HNA growth rate was kept higher than the maximum LNA growth rate over the course of simulations, there are additional model stock and flow variables that determine bacterial SDOC uptake, respiration, and production. One of them is the bacterial stock ($C_{BAC}$, here $_{BAC}$ can be applied to HNA and LNA both; Eq. A.4.53) whose time rate of change is determined prognostically in the model by multiple source (total DOC uptake) and sink terms (DOC excretion, respiration, grazing, and viral mortality). Bacterial SDOC uptake is proportional to the bacterial stock (Eq. A.4.13), and bacterial respiration is affected by the bacterial stock and total DOC uptake both (Eq. A.4.25). By definition, cell-specific BP is the biomass-normalized difference between total DOC uptake and respiration (i.e., cell-specific BP = ($G^C_{BAC,DOC}$ − $R^C_{BAC}$)/ $C_{BAC}$ , Eq. A.4.14, A.4.25, A.4.53). Because of these intertwined processes whose magnitudes are continuously adjusted and determined during optimization, keeping the high

maximum HNA growth rate does not simply guarantee higher SDOC uptake and cell-specific BP of HNA than those of LNA. One simple counterexample case would be: despite higher maximum growth rate of HNA and that of LNA ($\mu_{HNA} > \mu_{LNA}$) assigned initially and kept during optimization, if LNA biomass is larger than HNA biomass ($C_{HNA} < C_{LNA}$, due to the relative magnitude of sink and source terms for each bacterial group, Eq. A.4.53), this could potentially lead to larger SDOC uptake rate of LNA than that of HNA (Eq. A.4.13). Thus, these findings are the result of optimization in conjunction with the model assumption about class growth rates. We have added this explanation in Section 4.2 (line 428-435).

The authors have now clarified that their model considers flexible (Chl:)C:N:P stoichiometry, but this is mentioned only in the equations and the supplement. This information must be provided in the main text, e.g., under Sections 2.1 or 2.2 or a new 2.x section, as this information is quite crucial for understanding the model design. The statement that the model has 12 state variables (line 81) is simply wrong (I counted 32). This misinformation had led me to conclude that the model was based on a fixed-stoichiometry approach in my previous review. Fig. 1 has been amended regarding the flows of inorganic nutrients to phytoplankton. But it still remains a source of confusion. Fig. 1 shows two compartments, "Higher level" and "RDOM", which do not have corresponding differential equations, so the authors should either add the missing equations or modify Fig. 1 to clarify what these are (this applies also to Fig. 5).

We have expanded on Section 2.1 to detail the model's flexible stoichiometry and referred the rest information to Text S1 (line 91-102).

By 12 prognostic model state variables, we are referring to the subset of carbon stocks of biological compartments and dissolved inorganic nutrient compartments analyzed and presented in the study. These include diatoms (Eq. A.2.41), cryptophytes (Eq. A.3.37), HNA bacteria (Eq. A.4.53), LNA bacteria (Eq. A.4.53), microzooplankton (Eq. A.5.24), krill (Eq. A.6.27), detritus (Eq. A.7.4), LDOC (Eq. A.8.4), SDOC (Eq. A.8.7), $NH_4$ (Eq. A.9.2), $NO_3$ (Eq. A.9.3), and $PO_4$ (Eq. A.9.4). We have made this point clear in Section 2.1 (line 81-85).

In the previous version we mentioned that both higher levels and RDOM are implicitly represented as model closure terms in Section 2.1. Higher levels and RDOM play a role as source or sink terms of other explicit model state variables (i.e., krill removal by higher level, detrital production by higher level, SDOM production by higher level in Eq. A.6.24, bacterial RDOM excretion in Eq. A.4.26, krill RDOM excretion in Eq. A.6.21, SDOM to RDOM conversion in Eq. A.8.2) but the model does not calculate time derivatives of their concentrations. As suggested, we have modified both Figure 1 and Figure 5 captions to clarify that these two compartments are implicit as well as included the explanation above in Section 2.1 (line 85-90).

The sentence "Total (bulk) bacterial production (BP; BP = BPHNA + BPLNA) was constrained by observations, and therefore, the group-specific production (BPHNA and BPLNA, mmol C m-3 d-1) was determined during optimization:" (lines 99–100) is unclear. Does this mean that Eqs.

(3) and (4) apply only during the optimization? How do you calculate BP_HNA and BP_LNA when not optimizing?

Our apologies for the confusion. To answer your question first, Eq. (3) and (4) apply both before and during optimization. Before optimization the assigned initial parameter values are used to calculate LDOC uptake ($G^C_{HNA,LDOC}$, Eq. A.4.12), SDOC uptake ($G^C_{HNA,SDOC}$, Eq. A.4.13), and respiration ($R^C_{HNA}$, Eq. A.4.25) and the resulting $BP_{HNA}$. These parameter values are adjusted during the optimization resulting in new updated $BP_{HNA}$ values (based on optimized parameters) presented throughout our manuscript (the same applies to LNA). We have added this explanation in Section 2.1 (line 118-123).

On line 104, you state that "The modelling framework consisted of a dynamic (mechanistic) part and a data-driven part (Figure 2)" but Fig. 2 is about the data assimilation scheme and does not show or mention dynamic and data-driven parts.

We have modified Figure 2 to show the mechanistic and the data-driven parts. Also, by its design, the data assimilation methodology is a fusion of model dynamics and data constraints.

On lines 109–110, you introduce fmodes as functional modes, but even after reading the whole ms several times, it remains unclear what these are, e.g., which functions the fmodes describe. Since the fmodes are used later on in the statistical analysis, they should be explained clearly.

The fmode constructs are described on line 133-134: "functional modes (*f*modes hereafter) were derived from predicted community metabolic structure." For further details we refer the reader to Bowman et al. (2017). In brief the functional modes are derived in exactly the same way as the taxonomic modes, except that the SOM is trained on the abundance of predicted metabolic pathways rather than taxa.

On lines 310–311, "These results suggest a clear link between the modelled ecosystem functions and observed bacterial taxonomic (modes) and physiological (fHNA) traits observations." This, together with the absence of any significant relations for fmodes, seems to indicate that the functions (not decribed in the ms) of the functional modes were chosen inappropriately.

We appreciate the comment but disagree with the interpretation. In general, we find that functional modes are poorer predictors of ecological processes than taxonomic modes. This falls in part from the very different distribution of the underlying data in the marine environment. Taxa are much more sensitive to ecological processes (as drivers and responders) whereas many metabolic pathways – which may be widely distributed across taxa – are not.

The authors moved from a 0D to a 1D setup for the model but do not provide any information about the 1D setup except the depth levels. No indication about vertical mixing is given in the equations either, so they must be considered incomplete. Also, I am not convinced that, given the shallow model domain (20 m), a 1D design provides a significant advantage over 0D. But again, essential information is missing to allow a firm judgement, e.g., the depth of the mixed layer and

its seasonal variations. If the mixed layer is usually deeper than 20 m at the modelled site, then a 1D model offers no advantage over a 0D model. Also, no information is provided regarding the vertical geometry (are the three layers of the same height?) or the mixing scheme (implicit, explicit, positive definite, etc.). The authors should also indicate how the mixing coefficients were obtained or calculated. The reference to Kim et al. (2021) is insufficient, as this has not been published. The authors should just add a short section describing the vertical configuration and modify the equations accordingly.

Please note that in the previous version we detailed physical forcings of the model that takes the vertical structure and configuration into account (Figure S1, Text S2). The model implements the mixing scheme where vertical advection and detrital sinking are demonstrated with a third-order direct space-time upwind-biased scheme (Hundsorfer & Trompert 1994) and the Sweby flux limiter (Sweby 1984) but simplified to work for 1-D vertical advection only. Vertical diffusion is applied using a Crank-Nicholson vertically variable diffusion operation (Press et al. 1986), with a closed upper boundary and an open bottom boundary. We have added this information on the mixing scheme in Text S2 (line 304-311). We modeled 3 layers, 0, 10, and 20 m, which has 2, 16, and 4 m layer thickness, respectively, so that the center of each layer corresponded to the depths (surface, 10, and 20 m) from observations as closely as possible (added in Section 2.2 line 150-151). More importantly, the original model article (Kim et al. 2021) has been accepted and will be published soon in *Geoscientific Model Development* with which we hope to navigate potential readers for more details about the model framework. A preprint of the paper is now publicly available from https://gmd.copernicus.org/preprints/gmd-2020-375/

In the previous version we switched to the 1D framework to simulate seasonally-varying MLD and light impacts on the model stocks and flows with more realistic, observation-based boundary conditions at the base of the layer of interest (10 m, full description in Text S3 line 354-360) that were previously fixed to constant concentrations over time in the 0D setup. These constant vertical fluxes of the variables at 10 m are not appropriate for representing a seasonal cycle of the system and especially troublesome when MLD is deeper than 10 m. In the previously added Figure S1, we show that MLD is mostly deeper than 10 m but frequently shallower than 20 m over the seasonal cycle, so the 1D setup the model now correctly introduces the vertical fluxes from the base of the 10 m by mixing. On a minor note, despite the advantage of simulating the full water-column layers, we judged that it would be best to exclude depth levels without bacterial traits observations, yet to include an adequate number of depth levels for seasonal MLD and light impacts, and ultimately chose to model 0, 10, and 20 m. We have added this explanation in Section 2.2 (line 153-155).

The concept of the bacterial modes remains rather confusing. Since this is one of the main foundations of the present study, this must be clarified. My main problem with Fig. 6 is still that the explanation of the modes (also in the authors' response letter) does not seem to match what is shown in the panels. For example, Candidatus Pelagibacter ubique is supposed to dominate mode 6 (Fig. 6c) and C. Thioglobus singularis should dominate mode 1 (Fig. 6e). However, the relative

abundance of C. T. singularis in mode 1 never exceeds 0.25, whereas C. P. ubique has relative abundances between 0.25 and 0.35 in mode 1, according to panel c, so it appears that both modes 1 and 6 are dominated by C. P. ubique. I did go through Bowman et al. (2017) but could not find an explanation there either.

Thank you for noting the discrepancy between the figure and the caption. We have modified the example to *Dokdonia* sp. MED134 as that is a clearer example (line 1178-1180).

The description of the data assimilation and parameter optimisation has become much more accessible by the added explanations. Still, several points remain unclear. On lines 166–167, you write "... group-specific bacterial model parameters were optimized in the direction to properly represent the dynamics associated with each group ..." I do not understand what this means, even with the explanation in the next sentence, which describes a constraint imposed in the maximum bacterial growth rates.

We have significantly elaborated and rearranged elements in Section 2.3 to demonstrate details on the parameter optimization process (line 183-185, 193-209). Please see if the added paragraphs make sense and help you understand the optimization process better.

On lines 187–188, "When converting Chl to phytoplankton C (N) biomass, the maximum Chl to N ratio was used along with other reference ratios ..." it remains unclear why you use the maximum (rather than, e.g., an average) Chl:N ratio and what the other reference ratios are. This must be clarified. Also, throughout the ms, you mostly refer to C biomass, so it is unclear where, when, or why you convert to N biomass here.

We needed the Chl:C ratio to convert Chl to C biomass to calculate the fractional contribution of phytoplankton to the total (observed) POC, which also includes zooplankton and detritus carbon. However, the Chl:C ratio of phytoplankton has not been measured at the study site. The maximum Chl/N ratio ($\Theta$, g Chl *a* (mol N)$^{-1}$, Table S2-6) is the parameter that the model directly requires to calculate phytoplankton C growth and Chl production, the value of which is available from other data assimilation studies (Luo et al. 2010), so we used it and multiplied $\Theta$ by the Redfield C/N ratio of 0.15 to obtain phytoplankton C biomass. We acknowledge that by using the maximum Chl/C ratio we applied a minimum estimate of C and N biomass converted from Chl that the model needs to match. We also have updated the percentage of total POC and PON that living biomass account for that was mistakenly written in the previous version (Section 2.4, line 224-227).

On lines 198–199, you refer to "... normalized costs of individual data types (J'm) ..." The J'm seem to be indicated in Table 2 but they are never defined.

Fixed (line 237).

On lines 205–213, you refer to the depth of the mixed layer as affecting the calculation of the target error. Besides the lack of information on the mixed-layer depth, it is also unclear whether the

mixed layer was always deeper than 20 m, or whether you always applied the same CV throughout the whole model domain. Please explain clearly.

Apologies for the insufficient explanation about target errors in the previous version. In Kim et al. (2021) we calculated the climatological mean and standard deviation of each variable in the mixed layer per observation (vertical profile) over an extended year period (2002-03 to 2011-12) to get a more generalized picture with large sample size. We then used the climatological CV (from the same climatological mean and standard deviation) for target errors of the most data types and the same climatological standard deviation for target errors of the log-transformed data types. In other words, each data type was assigned with its own but non-time varying CV.

In the present study though, MLD is mostly deeper than 10 m but frequently shallower than 20 m (Figure S1), and following the methodology as in Kim et al. (2021) would throw out most vertical profiles, decrease the sample size, and make inadequate cases for representing the overall observational errors and seasonal-interannual variations (e.g., for all data types the depth levels measured typically span surface, 10 m, 20 m, etc, so shallow MLD would leave vertical profiles with only one or at most two data points within MLD). As stated in the previous version, we instead calculated the climatological standard deviation, and CV in the upper 20 m (i.e., 0, 10, and 20 m) per profile over the four study years of the present study, which we adjusted to similar values in Kim et al. (2021) as they were, of course, higher than those in Kim et al. (2021), largely due to the inclusion of the observations at 20 m when MLD < 20 m. For adjustment, we derived the ratios of the climatological CV (for most data types) and standard deviation (for the log-converted variables) between our study and Kim et al. (2021), averaged the ratio for the same categorical data types (e.g., nutrients ($NO_3$ and $PO_4$), phytoplankton (diatoms, cryptophytes, and primary production), bacteria (HNA and LNA biomass and production)), and multiplied this ratio to what we calculated for the present study to reduce to the level in the "mixed layer" to avoid an overestimated target error of each data type. Though complicated, we chose to do this way of combining target errors in Kim et al. (2021) and error adjustment hoping that it would more realistically represent the dynamics at Palmer Station B and for the 4 study years in the present study because target errors in Kim et al. (2021) were calculated for the 11-year period of the data from slightly offshore Palmer Station E. We have created a new Text S4 section (line 361-381) to include this explanation (added also to refer in main text, line 245-249).

On lines 240–241, "... 5-7 constrained parameters and 3-6 optimized parameters ..." Please clearly explain how you define and determine constrained and optimized parameters and how they differ. Also, explain CS and OP in Tables S2–S6.

We have made this part clear in Section 2.3 (line 194-199).

On lines 254–155 "However, the model skill for HNA biomass slightly degraded in the climatological model (Figure 3b), with lower correlations and normalized standard deviation and higher RMSD than the four years together (Figure 3a)." Since the individual years have been optimized individually, this result was to be expected. What could be more informative is a comparison with simulations for the individual years but with the same parameter set, e.g., the

most portable one. This could provide insight into the influence of parameter differences compared to that of different boundary conditions and forcings between the different years.

Thanks for your suggestion. Please note that the rationale for the climatological model skill assessment in Taylor diagrams is because we use the climatological model parameters for the climate change simulations, not the model parameters from any specific year. The portability of the most portable model (PI = 0.76 ± 0.11, 2012-13) is still quite similar with the next portable year's model (PI = 0.76 ± 0.11 for 2013-14, Table 2) thereby making it hard to choose one specific year's model parameter set over others to represent the overall "mean" ecosystem required for climate change simulations. We have updated Section 3.1 (line 301-304) and highlighted again the parts on model portability relevant to your comment (line 354-357, line 416-420).

These sentences are incomprehensible and must be corrected. I could not figure out what you wanted to say here: Lines 273–275: "C stocks and flows averaged over the growth (Figure 5) and normalized by NPP (normalized by NPP in 1-day for C stocks; Figure S9) season for each year summarized an annual snapshot of the group-specific bacterial dynamics."

Fixed (line 313-315).

Lines 283–284: "NO3, POC, and SDOC in unassimilated years were modelled to values comparable to those in other assimilated years (Figure 5)."

Fixed (line 323-324).

Line 399: "Modelled nutrient stocks were above detect limits and indicated the lack of macronutrient limitations."

Fixed (line 447-448).

Fig. 3: The caption for panel a (2010 - 2013) is confusing. The simulations also cover 2014 and this caption gives the impression that the simulations went through 2010–2013 continuously, which is not what you did. You should come up with a better caption. Also, as mentioned above, a third panel showing results for different years with a single parameter set could be useful. The last sentence of the caption seems to make no sense, since the x-axes of both panels are the same.

Thanks. We have deleted the captions from both (a) and (b) in Figure 3 and instead explained those in the caption. Please see our response above about using the single year model parameter set in Taylor diagrams. We also have made both Taylor diagrams look less busy by labeling each data point number rather than text.

Fig. 4: The numbers and letters are very hard to read and often overlap. Maybe rearrange in 4 columns (2 for the means and 2 for the CVs)? The units in (b) are wrong, the CV is dimensionless.

Please see our updated Figure 4 if this addresses your concerns.

Fig. 5: The caption says that the panels show C stocks and flows in units of mmol C m-2 and mmol

C m-2 d-1 but the panels also show NH4, NO3, and PO4, so the associated numbers must have different units. The caption explains the numbers in the first rows and the numbers in parentheses but not the numbers in the second rows next to the arrows. Then it says that "N and P flows, as well as the flows smaller than 0.01 mmol C m-3 d-1, are omitted." but the panels show arrows from and to NH4 and to the inorganic nutrient compartments.

We have fixed the Figure 5 and Figure S9 captions as suggested.

Fig. 8 suffers from the same problems as Fig. 4 (% numbers are also dimensionless). In addition, the first rows in (b) should be left out as they are always 0 by definition.

We have fixed Figure 8 the same as Figure 4, but have kept the first row in both panels (a) and (b) to explicitly represent the baseline state and for consistency between the two panels. We also have corrected Figures S10-11.

---

## Author Response (AR3)

**Letter to Editor**

Dear Dr. Grégoire,

Thank you very much for handling our manuscript and having provided several important suggestions. In an effort of thoroughly addressing your and the two reviewers' concerns we also attach our response to your comments as follows.

1) The statement mentioned in the abstract (line 21-23) that ""High nucleic acid (HNA) bacteria show relatively high cell-SPECIFIC productivity, respiration, and utilisation of the semi-labile dissolved organic carbon pool compared to their low nucleic acid (LNA) bacteria counterparts." is highlighted as an important result of model simulations performed in this paper although it results from model parameterization (and hence assumptions). Reviewer #2 pointed out that this conclusion can be considered as an output from the model if it applies to the non-normalized quantity. However, as written, this refers to the biomass-normalized quantities. Please carefully check the detailed comment of reviewer #2 and provide a clear answer.

We agree with reviewer #2. The fact that cell-specific BP, respiration, SDOC uptake rates were significantly higher for HNA bacteria compared to those for their LNA counterparts (Section 3.2) is mainly because of the way the parameter optimization was conducted (now detailed in Text S3). The higher initial parameter values assigned for HNA bacterial growth, RDOC excretion, mortality, and respiration rates (Table 1) might drive not only their faster cell-specific growth rates but also their higher DOC uptake rates to coexist with LNA bacteria when the loss rates were relatively large for HNA bacteria. Though driven by the model assumptions, the important aspect of these results lies in the fact that the model can leverage such assumptions to examine the implications for the WAP food-web dynamics and biogeochemistry (line 356-362).

2) The model description needs to be revised and the number of state variables to be checked. At the time of the first review, the GMD paper was not available. Since it is now published, I suggest that you keep in the main text a summary of the model main characteristics and put in the appendix details on the formulation. These details have been requested by Reviewer #2 and this is important that it remains accessible in the manuscript but to improve the clarity of the paper and make it more accessible, transfer some parts in the supplementary section as suggested by reviewer#3 in his/her comment.

As suggested by both reviewers, we have greatly reduced and simplified the Methods section on the model processes, and much of the relevant sections are now redirected to Kim et al. (2021) or Supplementary Material of our study. As suggested, we also have removed the model equations previously in Appendix and only focused on the formulation of the newly added processes and equations for HNA and LNA bacteria in main text (line 78-107).

3) Both reviewers do not understand the concept of bacterial models and fmodes. This has to be absolutely clarified. To refer to Bowman et al., (2017) is not enough. Please clearly explain, in this paper, to what functions the fmodes refer. This comment has been mentioned many times by reviewer #2 and the third reviewer has exactly the same concern. Please carefully check and answer the detailed comments of Reviewers #2 and #3.

We completely agree with your assessment. Thus, after significant thought, we have decided that it would be best to completely remove the Methods section on the functional modes given that the functional modes did not have significant relationships with the modelled bacterial C stocks and rates and other ecosystem functions and were not presented in Results and Discussion to begin with. Thus, Removing this part does not change the findings and conclusions of our study. We also have made the Methods section on the taxonomic modes more accessible and in line with our mechanistic modeling perspectives, as well as moved the taxonomic mode figure from main text to Supplementary Material (line 108-121, Fig. S10).

4) Please explain why the variability in model simulations is lower compared to that from observation.

The lower variability in the simulations should be considered as the limitation of our model and we have included the relevant information in the revised manuscript. The model results have less variability than the observations likely because we do not have sufficient information in the model forcing to capture all of the small-scale and high-frequency sources of variability, such as local circulation and tidal flow near Palmer station in the 4 study years modelled in this study. By contrast, our model adequately captures seasonal variations in modelle ecosystem dynamics. This is likely because such high frequency processes do not strongly rectify into the seasonal cycles in the system (line 382-386).

**Response to Referee #1 (Reviewer #2)**

Third review of "Modelling polar marine ecosystem functions guided by bacterial physiological and taxonomic traits" by H. H. Kim et al.

I think this is the first time I am writing a third review for a manuscript. In their second revision, the authors have now clarified/corrected most of the technical and language problems of the previous versions. Nevertheless, substantial problems remain, as detailed below, so that I still can not recommend publication of the ms as is. While the amount of required changes is quite small, the confusion of assumptions and results is sufficiently severe, so that it should be considered a major revision.

Thank you for reviewing our manuscript again. We have addressed your concerns as follows.

My main problem now lies in the still unresolved problem that the authors present an immediate consequence of their model assumptions as a result of their study, namely that the "High nucleic acid (HNA) bacteria show relatively high cell-specific productivity, respiration, and utilisation of the semi-labile dissolved organic carbon pool compared to their low nucleic acid (LNA) bacteria counterparts." (Abstract, lines 21–23). The authors' response does not really address this problem as it only applies to the total, rather than the biomass-normalised, rates. Please note that I have no problem accepting as a result the finding that the total (not biomass-normalised) rates are higher for HNA than for LNA, and I also think that this would be actually much more relevant in terms of both ecology and biogeochemistry. I also do not question that the growth rate of the HNA may potentially be lower at low labile DOC concentration because the HNA have a higher half-saturation concentration. But this does not apply (at least not for the parameters shown in Tables S2–S6) for the labile DOC concentrations in this study (Fig. 8, LDOC). In addition, several (not optimised or constrained) loss-rate parameters (RDOC production, mortality, respiration) are higher for HNA, and these must be compensated by faster DOC uptake in order to allow coexistence of HNA and LNA. Clearly, therefore, the higher biomass-specific rates of HNA are imposed by the model assumptions and must not be presented as a result.

We agree with your comments. The fact that cell-specific BP, respiration, SDOC uptake rates significantly higher for HNA bacteria compared to those for their LNA counterparts (Section 3.2) is mainly because of the way the parameter optimization was conducted (now detailed in Text S3). The higher initial parameter values assigned for HNA bacteria's growth, RDOC excretion, mortality, and respiration rates (Table 1) might drive not only their faster cell-specific growth rates but also their higher DOC uptake rates to coexist with LNA bacteria when the loss rates were relatively large for HNA bacteria. Though driven by the model assumptions, the important aspect of these results lies in the fact that the model can leverage such assumptions to examine the implications for the WAP food-web dynamics and biogeochemistry (line 356-362).

The description of the state variables on lines 80–102 is still wrong. The statement that the model has 12 state variables (line 81) is simply not true. I do not understand the hesitation of the authors to correct this obvious mistake.

We have 12 model state variables as stated in the previous version of the manuscript. The confusion may have come from the fact that there are 14 boxes in the model schematic (Fig. 1). However, high level and RDOM are not state variables because their time derivitatives are not calculated in the model, even if there are C flows going into them as the sink terms of other model state variables. If we counted N and P components of the model compartments the number of state variables increases further. As the other reviewer suggested as well as we are not sure where this confusion on the number of state variables comes from, we have decided to omit this infomraiton in the revised version, given that the actual total number of state variables is not important.

The problems with the bacterial modes and fmodes largely remain. For example, I had asked what functions the functional modes refer to. The authors refer (also in their response letter) to Bowman et al. (2017) for details but all that Bowman et al. (2017) write about the fmodes is this: "Based on inspection of the within-cluster sum of squares plot, we identified […] eight modes based on inferred metabolic pathways (not shown)." I think it is impossible to judge the validity of the statements regarding the fmodes without concrete information about the associated actual functions (metabolic pathways). How can one know whether these functions are selected in a meaningful or useful manner if no information about them is provided? Regarding the bacterial modes, in their response letter, the authors write that they changed the example in Fig. 6 to Dokdonia. This does not address the problem I described. I never had any problem understanding the Dokdonia case. My problem is understanding how the assignment of the modes to species works and I explained the (apparent?) contradiction between the authors' definition and what is shown in Fig. 6 with the example of Candidatus Pelagibacter ubique and C. Thioglobus singularis. Using a set of species where this problem does not show up obviously does not help here. The unclear presentation of the mode concept and the above problem of the confusion of assumptions and result are the reasons why I grade the scientific significance and quality as poor. The presentation quality gets a fair grade mainly because of the extremely poor language quality. While I do appreciate the additional information about the model and the corrections in this latest revision, I do in fact expect that authors supply this kind of information already with the initial submission.

Thank you for your comments. we have decided that it would be best to completely remove the Methods section on the functional modes, given that the functional modes did not have significant relationships with the modelled bacterial C flows and other ecosystem functions and were not presented in Results and Discussion to begin with. Thus, please note that removing this part does not change any of the findings and conclusions in our study. The revised manuscript now only focuses on the taxonomic modes. We also have moved the taxonomic mode figure (previously Fig.

6) to Supplementary Material (now Fig. S10) to simplify the relevant Methods section and make it more accessible (line 108-121). The revised mansucirpt now briefly discusses what each mode is associated with regard to bacterial taxonomy in the Results section (line 264-267). We hope these efforts address your concerns.

The sentence on lines 447–448 is still unclear to me. "… the values above …" (above what?) This is followed by "… detect limits …" It appears that two sentences were merged and something was lost, e.g., "… the values above those required by …" "… this was used to detect the limits or indicate the lack of macronutrient limitation …"

Thank you – we fixed that sentence to: The modelled nutrient stocks were above the detection limits, indicating no evidence of macronutrient limitations at the study site (line 371-372).

**Response to Referee #3 (Reviewer #3)**

This paper uses data from a 1D ecosystem model alongside bacterial genomic data to give insight into the role of bacteria in ecosystem functioning at a site in the West Antarctic Peninsula. The ecosystem model, recently accpeted for publication in a separate paper, has been modified to include two functional types of bacteria, HNA and LNA, and it is able to assimilate biomass of these types using flow cytometry data. Outputs from the model are examined in association with bacterial groupings derived from previously published taxonomic analysis of genome data from the same location. This is interesting work, and I was pleased to see a study that combine ecosystem model and genome data. The paper has been much improved by the previous cycles of review. However, I found it difficult to read and I recommend a little further work before it is published, to make it accessible to a wider readership.

Thank you for taking time to review our manuscript and also for your positive feedback. Please see below how we have addressed your comments on the previous version of this manuscript.

First, now that the paper describing the ecosystem model has been accepted for publication (Kim et al., 2021) the methods section can be reduced and simplified. Section 2.1 just needs a summary of the model previously published: the key points are the main functional types, which elements are tracked (C,N,P) and whether there is flexible stoichiometry. Of course the differences from Kim et al. (2021) need to be covered, but this can be relatively brief. I would remove the equations and the references to them – I know that a previous reviewer requested them, but they are identical to the ones in Kim et al. (2021) and can now be read there; lines 80-117 will be much easier to read without the references. If the equations are kept the symbols need to be explained. The description of the data assimilation scheme in sections 2.3-2.5 is very similar to that in Kim et al. (2021) and does not need to be repeated in such detail. Only a summary is needed in this paper, with the emphasis on the differences compared to the first paper, e.g. the difference in calculation of the target error. Reduced in this way, the methods section would be easier to read and give more weight to the work on bacteria, which is the novel part of this paper.

Thank you for your suggestion. As you suggested, we have greatly reduced and simplified the Methods section by moving many of the details on parameter optimization processes and target error adjustment to Supplementary Material (now in Text S3, S5). However, we still provide a summary of the key model aspects and in particular what is new to this modified model compared to the original WAP-1D-VAR v1.0 (line 78-107, 129-132, 181-183). Throughout the manuscript, we now redirect the reader to Kim et al. (2021) for more through presentations of the details of the base model. As suggested, we also have removed all the model differential equations of other ecosystem varaibles previously presented in main text as well as from Appendix.

Second, I suggest giving a slightly fuller explanation of the genomic data, for readers like myself

who are less familiar with this than with the modelling. The terms mode, functional mode, closest estimated genomes and closest completed genomes all seem to be specific to Bowman et al. (2017) and as such need to be explained more fully (or omitted if they are not needed – I don't see that referring to closest estimated genomes and closest completed genomes is required for the discussion here). I understand that taxonomic modes are groupings based on 16S rRNA gene sequence, but I don't understand where the fmodes come from. Line 134 says that "functional modes (fmodes hereafter) were derived from predicted community metabolic structure" – what data was this based on? From reading Bowman et al., 2017, I think it is also the 16S data, but the phrasing here seems to imply that it comes from a different source. I'm also not clear how the distinction between modes and f-modes can be interpreted: in section 3.3 it is stated that "fmode did not have a significant relationship with any of the modelled ecosystem functions examined" – how can we interpret that in terms of the bacterial contribution to ecosystem functioning?

Thank you for your comments. We agree that our bacterial mode description was not as clear as it should have been and it caused lots of confusion to both you and the other reviewer. We have realized that presenting the details on genomic and statistical analyses for mode extraction would make our manuscript even more difficult to understand, so we have decided to greatly simplify the Methods section on bacterial modes. First, we have completely removed the Methods section on functional modes as they did not show any significant relationships with the modelled bacterial stock and flows or other key ecosystem functions. Second, we have shortened the Methods section on taxonomic modes and changed the wording to be more in line with our numerical modeling aspects (line 108-121). Third, we have moved the taxonomic mode figure to Supplementary Material (Fig. S10) and only briefly discuss how each mode represents its specific taxonomic attributes in the Results section (line 264-267).

Third, section 2.2 and Figure 2 led me to expect more integration of the genomic data into the model than I actually found. Figure 2 shows the data-driven part of the model, where the bacterial modes are used, but I could not find any description of how this is done in the methods. Line 133 refers to "the data-driven part representing how bacterial modes (Bowman et al. 2017) are compared to final model outputs based on optimized model parameters from the dynamic part". So is this just a comparison between model outputs and bacterial modes, as presented in sections 3.3 and 4.3? In what sense is the modelling framework data driven, i.e. what does the arrow in Figure 2 pointing from the bacterial modes to the model field represent?

Thank you for your comments. Your assessment is correct that we compare the optimized model outputs from the ecosystem model (Fig. 1) with the bacterial modes from Bowman et al. (2017). We refer to the bacterial mode part as "data-driven" because Bowman et al. (2017) uses a data-driven approach to derive the taxonomic modes; in particular, they used an unsupervised machine learning algorithm called Kohonen's self-organizing maps (Kohonen, 2001) to reduce the high dimensionality of 16S rRNA gene sequence abundance data to a single taxonomic mode. We have

elaborated the Methods section to make this aspect clear (line 108-121). We also removed the arrow in Fig. 2 from the "bacterial modes" in the data-driven part to the "model outputs" in the mechanistic part as the bacterial modes themselves are not directly incorporated to the ecosystem model in contrast to HNA and LNA C biomass data, which are assimilated into the simulation.

My main scientific concern is that in many cases the model values have a much lower standard deviation than the observations (Figure 3, Figure S2), and more so than in the previous model (Kim et al, 2021). So the model appears to be missing much of the variability observed in the field measurements. Do the authors have any comment on this?

Thank you for making this point. The model results have less variability than the observations likely because we do not have sufficient information in the model forcing to capture all of the small-scale and high-frequency sources of variability, such as local circulation and tidal flow near Palmer station in the 4 study years modelled in our study. By contrast, our model adequately captures seasonal variations in modelle ecosystem dynamics. This is likely because such high frequency processes do not strongly rectify into the seasonal cycles in the system (line 382-386).

With these modifications, I think the paper will be a useful contribution to the literature, with relevance beyond the particular study area.

A few specific points:
Abstract: the abbreviation WAP needs to be explained.
Line 61: r missing from Palmer
Line 81: I agree with the previous review comment that there are more than 12 state variables, and it is still not clear that only the carbon stocks are being considered. But why give a number? I don't see that this is important for the rest of the text.
Line 83: LDOC in the text is given as LDOM in Figure 1. Is it the same thing? Similarly for SDOC, RDOC.
Line 108: rpesent instead of present
Line 168: avialable instead of available
Line 291 and Figure 3: it is not specified how the standard deviation is normalized.
Tables S2-S6: it would be helpful to explain the abbreviations OP and CS in the legends.
Text S1 to S2: I think much of this is now part of Kim et al. (2021) and can be removed.

Thank you – all of these technical comments have now been fixed.

---

## Author Response (AR4)

**Response to reviewer**

Thank you for the opportunity to review the revised version of this paper. The authors have responded to the questions and feedback from the editor and reviewers and the paper is now much improved. The revised methods section is much easier to read and the removal of the fmodes is a helpful simplification. The fact that the fmodes did not show any significant relationships with key ecosystem functions may be interesting and worth investigating further, but I don't think it is ready for publication yet. The authors have adequately addressed the other points raised in review:

- The link between the higher HNA cell-specific rates and the assumptions built into the model parameters (lines 356-362 and the abstract);
- The reasons for, and the need to improve on, the low variability in the model outputs compared to observations (lines 382-386).

The paper has some novel aspects and its methods could be debated, but in my opinion it is now ready to be published and further discussion can happen through the normal scientific process.

> Thank you very much for your positive review.

There are just a few very minor points which I think should be addressed, as follows:

Line 205: I suggest adding a reference to Text S3 as well as Tables S2-S6.

> Fixed.

Line 232 "the model captured best the temporal and spatial (depth) variability of PP": I don't understand why this is "best". The skill for PP is relatively low, looking at Figure 3. Is the point that for PP the variability is captured better than the absolute value? I think this sentence needs revising.

> Revised (line 232-233).

Line 259 "There was little interannual variability in the average microzooplankton": the values in Figure 5 range from 0.39 to 0.76, which does not appear to me to be small variability – the highest value is nearly twice the lowest. This sentence needs to be rephrased, or the variability put into context to explain why this variability is little.

> Revised (line 259-260).

Figure S9 legend: I think the units are not correct – if the values are normalized by NPP they should not be in mmol C m-3 etc.

> You are correct as it should be unitless. We removed particle sinking flux values in Figure S9 as its unit becomes irrelevant and difficult to interpret if normalized by PP or PP in 1-day.

Figure S10a: The number 5 is almost invisible on the yellow hexagon – could it be changed to black?

> Fixed as suggested.

There are a few spelling mistakes in both the manuscript and the supplementary material. Use of English: I suggest the following changes for the authors' consideration. In my opinion they would improve the clarity of the manuscript.

> All of these spelling mistakes have been fixed following your suggestions below.

Line 128 "the results from 10 m are only presented in detail": I think this should be "only the results from 10 m are presented in detail".

Line 130 "yet to include the adequate number": change to "yet to include an adequate number"

Line 223: add "of" after "Because"

Line 256 "the variable for a single year's": I think this should be "the variable for which a single year's"

Line 336 "we assigned the identical initial parameter value": change to "we assigned an identical initial parameter value"

Line 347 "exhibited the intermediate levels": remove "the", i.e. "exhibited intermediate levels"

Line 348 "same number of the constrained parameters": remove 'the"

Line 350 "suggesting the connection": change to "suggesting a connection"

Line 366 "with the cell-specific growth rate": change to "with a cell-specific growth rate"

Line 374 "the strong interannual variability": remove "the"

Line 390 "showing the increased HNA growth rates": remove "the"

Line 406 "often observed during the Antarctic phytoplankton: remove "the"

Line 447 "was characterized by the negative temperature anomaly": change "the" to "a"

Line 448 "and the positive sea-ice anomaly": change "the" to "a"